# STRATEGIC EXPLORATION FOR INVERSE CONSTRAINT INFERENCE WITH EFFICIENCY GUARANTEE

## ABSTRACT

Optimizing objective functions under constraints is a fundamental problem in many real-world applications. However, constraints are often not explicitly provided and must be inferred from the observed behavior of expert agents. The problem is known as Inverse Constraint Inference (ICI). A common solver, Inverse Constrained Reinforcement Learning (ICRL) seeks to recover the optimal constraints in complex environments in a data-driven manner. Existing ICRL algorithms collect training samples from an interactive environment. However, the efficacy and efficiency of these sampling strategies remain unknown. To bridge this gap, we introduce a strategic exploration framework with guaranteed efficiency. Specifically, we define a feasible constraint set for ICRL problems and investigate how expert policy and environmental dynamics influence the optimality of constraints. Motivated by our findings, we propose two exploratory algorithms to achieve efficient constraint inference via 1) dynamically reducing the bounded aggregate error of cost estimation and 2) strategically constraining the exploration policy. Both algorithms are theoretically grounded with tractable sample complexity. We empirically demonstrate the performance of our algorithms under various environments.

## 1 INTRODUCTION

Constrained Reinforcement Learning (CRL) addresses sequential decision-making problems within safety constraints and achieves considerable success in various safety-critical applications (Gu et al., 2022). However, in many real-world environments, such as robot control (García & Shafie, 2020; Thomas et al., 2021) and autonomous driving (Krasowski et al., 2020), specifying the exact constraint that can consistently guarantee the safe control is challenging, which is further exacerbated when the ground-truth constraint is time-varying and context-dependent.

Instead of utilizing a pre-defined constraint, an alternative approach, Inverse Constrained Reinforcement Learning (ICRL) (Malik et al., 2021; Liu et al., 2024a), seeks to learn the constraint signals from the demonstrations of expert agents and imitate their behaviors by adopting the inferred constraint. ICRL effectively incorporates expert experience into the online CRL paradigm and thus better explains how expert agents optimize cumulative rewards under their empirical constraints. Under this framework, existing ICRL algorithms often assume the presence of a known dynamics model (Scobee & Sastry, 2020; McPherson et al., 2021), or a generative transition model that responds to queries for any state-action pair (Papadimitriou et al., 2023; Liu et al., 2023). However, this setting has a considerable gap with scenarios in practice where the transition models are often not available, or even time-varying, necessitating agents to physically navigate to new states to learn about them through exploration.

To mitigate the gap, some recent studies (Malik et al., 2021; Qiao et al., 2023; Baert et al., 2023) explicitly maximized the policy entropy throughout the learning process, yielding soft-optimal policy representations that favor less-selected actions. Unfortunately, such an uncertainty-driven exploration largely ignores the potential estimation errors in dynamic models or policies. To date, it still lacks a theoretical framework to demonstrate how well the maximum entropy approaches facilitate the accurate estimation of constraints.

In this paper, we introduce a strategic exploration framework to solve ICRL problems with guaranteed efficiency. Recognizing the inherent challenge in uniquely identifying the exact constraint from expert demonstration, the objective of our framework is to recover the *set of feasible constraints* where each

element can accurately align with expert preferences, rather than to identify an exact constraint. By explicitly representing these constraint sets with the reward advantages and the transition model, we manage to confine the constraint estimation error with the discrepancy by comparing the estimated environmental dynamics and expert policy with the ground-truth ones. This strategy provides a quantifiable measure of error for our constraint estimation, linking it directly to a computationally tractable upper bound.

Under our framework, we design two strategic exploration algorithms for solving ICRL problems: 1) A Bounded Error Aggregate Reduction (BEAR) strategy, which guides the exploration policy to minimize the upper bound of discounted cumulative constraint estimation error; and 2) Policy-Constrained Strategic Exploration (PCSE), which diminishes the estimation error by selecting an exploration policy from a predefined set of candidate policies. This collection of policies is rigorously established to encompass the optimal policy, thereby promising to accelerate the training process significantly. For both algorithms, we provide a rigorous sample complexity analysis, furnishing a deeper understanding of the training efficiency of these algorithms.

To empirically study how well our method captures the accurate constraint, we conduct evaluations under different environments. The experimental results show that PCSE significantly outperforms other exploration strategies and is applicable to continuous environments.

## 2 RELATED WORK

In this section, we introduce previous works that are most related to our algorithms. Additional discussions can be found in Appendix B.

**Exploration in Inverse Reinforcement Learning (IRL).** Compared with the exploration strategies in RL for forward control (Amin et al., 2021; Ladosz et al., 2022), the exploration algorithms in IRL have relatively limited studies. Balakrishnan et al. (2020) utilized Bayesian optimization to identify multiple IRL solutions by efficiently exploring the reward function space. To learn a transferable reward function, Metelli et al. (2021) introduced an active sampling methodology that is designed to target the most informative regions with a generative model to facilitate effective approximations of the transition model and the expert policy. A subsequent research (Lindner et al., 2022) expanded this concept to finite-horizon MDPs with non-stationary policies, crafting innovative strategies to accelerate the exploration process. To better quantify the precision of recovered feasible rewards, Metelli et al. (2023) recently provided a lower bound on the sample complexity for estimating the feasible reward set in the finite-horizon setting with a generative model. However, these methods study only reward functions under a regular MDP without considering the safety of control or the constraints in the environments.

**Inverse Constrained Reinforcement Learning (ICRL).** Unlike IRL which solely focuses on the recovery of reward functions, ICRL seeks to elucidate the preference of expert agents by inferring which constraints they follow. The majority of ICRL algorithms update the cost functions by maximizing the likelihood of generating the expert dataset under the maximum (causal) entropy framework (Scobee & Sastry, 2020). This method can be efficiently scaled to both discrete (McPherson et al., 2021) and continuous state-action space (Malik et al., 2021; Baert et al., 2023; Liu et al., 2023; Qiao et al., 2023; Xu & Liu, 2024). To improve training efficiency, recent studies combined ICRL with bi-level optimization techniques (Liu & Zhu, 2022; Gaurav et al., 2023). However, current ICRL methods have not explored exploration strategies or conducted theoretical studies about the sample complexity of their algorithms.

## 3 PRELIMINARIES

**Notation.** Let $\mathcal{X}$ and $\mathcal{Y}$ be two sets. $\mathcal{Y}^{\mathcal{X}}$ represents the set of functions $f : \mathcal{X} \to \mathcal{Y}$. Let $\Delta^{\mathcal{X}}$ denote the set of probability measures over $\mathcal{X}$. Let $\Delta_{\mathcal{Y}}^{\mathcal{X}}$ denote the set of functions: $\mathcal{Y} \to \Delta^{\mathcal{X}}$. We define the vector infinity norm as $||a||_\infty = \max_i |a_i|$ and the matrix infinity norm as $||A||_\infty = \max_i \sum_j |A_{ij}|$. We define $\min_{x \in \mathcal{X}}^+ f(x)$ to return the minimum positive value of $f$ over $\mathcal{X}$. The complete notation is given in Appendix A.

**Constrained Markov Decision Process (CMDP).** We model the environment as a stationary CMDP $\mathcal{M} \cup c := (\mathcal{S}, \mathcal{A}, P_\mathcal{T}, r, c, \epsilon, \mu_0, \gamma)$, where $\mathcal{S}$ and $\mathcal{A}$ are the finite state and action spaces, with the cardinality denoted as $S = |\mathcal{S}|$ and $A = |\mathcal{A}|$; $P_\mathcal{T}(s'|s, a) \in \Delta^\mathcal{S}_{\mathcal{S} \times \mathcal{A}}$ defines the transition distribution; $r(s, a) \in [0, R_{\max}]$ and $c(s, a) \in [0, C_{\max}]$ denote the reward and cost functions; $\epsilon$ defines the threshold (budget) of the constraint; $\mu_0 \in \Delta^\mathcal{S}$ denotes the initial state distribution; and $\gamma \in [0, 1)$ is the discount factor. $\mathcal{M}$ denotes the CMDP without cost (i.e., CMDP$\backslash c$). The agent's behavior is modeled by a policy $\pi \in \Delta^\mathcal{A}_\mathcal{S}$. $\Pi^*_{\mathcal{M} \cup c}$ denotes the set of all optimal policies for a CMDP. The expert policy $\pi^E$ is optimal, i.e., $\pi^E \in \Pi^*_{\mathcal{M} \cup c}$. Let $f \in \mathbb{R}^S$ and $g \in \mathbb{R}^{\mathcal{S} \times \mathcal{A}}$, we slightly abuse $P_\mathcal{T}$ and $\pi$ as operators: $(P_\mathcal{T} f)(s, a) = \sum_{s' \in \mathcal{S}} P_\mathcal{T}(s'|s, a) f(s')$ and $(\pi g)(s) = \sum_{a \in \mathcal{A}} \pi(a|s) g(s, a)$. Moreover, the expansion operator $(Ef)(s, a) = f(s)$. In our work, we assume a discrete finite state-action space within an infinite horizon setting.

Given the CMDP, we define the discounted normalized occupancy measure (Altman, 2021) as $\rho^\pi_\mathcal{M}(s, a) = (1 - \gamma) \sum_{t=0}^{\infty} \gamma^t \mathbb{P}^\pi_{\mu_0}(S_t = s, A_t = a)$ so that $(1 - \gamma) V^\pi(r, \mu_0) = \langle \rho^\pi_\mathcal{M}, r \rangle$ and $(1 - \gamma) V^\pi(c, \mu_0) = \langle \rho^\pi_\mathcal{M}, c \rangle$, where $(1 - \gamma)$ is the normalizer for $\rho^\pi_\mathcal{M}$ to be a probability measure and $V^\pi$ is a reward or cost state-value function under the policy $\pi$ and the initial distribution $\mu_0$.

**Constrained Reinforcement Learning (CRL).** Within a CMDP environment, CRL learns a policy $\pi$ that maximizes the cumulative rewards subject to a known constraint:

$$\arg \max_\pi \ \mathbb{E}_{\mu_0, \pi, p_\mathcal{T}} \left[ \sum_{t=0}^{\infty} \gamma^t r(s_t, a_t) \right] \quad \text{s.t.} \ \mathbb{E}_{\mu_0, \pi, p_\mathcal{T}} \left[ \sum_{t=0}^{\infty} \gamma^t c(s_t, a_t) \right] \le \epsilon. \tag{1}$$

In this paper, we primarily focus on the cumulative constraint as in (1) instead of instantaneous constraints due to its broader applications (Wachi et al., 2024). In particular, since $c \ge 0$, by setting $\epsilon > 0$, the constraint in (1) denotes a soft constraint, enabling its application to the environment with stochastic dynamics. On the other hand, we convert this constraint into a hard one when setting $\epsilon = 0$, which facilitates the enforcement of absolute constraints at each decision step.

**Value and advantage functions.** We define the reward action-value functions as $Q^{c; \pi}_\mathcal{M}$ and $Q^{r, \pi}_\mathcal{M}$. The superscript $r$ specifies the actual costs or rewards evaluated. The reward action-value function is $Q^{r, \pi}_\mathcal{M}(s, a) = \mathbb{E}_{\pi, P_\mathcal{T}} [\sum_{t=0}^{\infty} \gamma^t r(s_t, a_t)|s_0 = s, a_0 = a]$, and the reward advantage function follows $A^{r, \pi}_\mathcal{M}(s, a) = Q^{r, \pi}_\mathcal{M}(s, a) - V^{r, \pi}_\mathcal{M}(s)$, where the reward state-value function is $V^{r, \pi}_\mathcal{M}(s) = \mathbb{E}_\pi [Q^{r, \pi}_\mathcal{M}(s, a)]$. The subscript specifies the environment $\mathcal{M}$ that contains reward function $r$. The superscript specifies the actual rewards under evaluation. We define the cost action-value function as $Q^{c, \pi}_{\mathcal{M} \cup c}(s, a) = \mathbb{E}_{\pi, P_\mathcal{T}} [\sum_{t=0}^{\infty} \gamma^t c(s_t, a_t)|s_0 = s, a_0 = a]$. The subscript specifies the CMDP environment $\mathcal{M} \cup c$. The superscript specifies the actual costs under evaluation. The cost state-value function follows $V^{c, \pi}_{\mathcal{M} \cup c}(s) = \mathbb{E}_\pi [Q^{c, \pi}_{\mathcal{M} \cup c}(s, a)]$.

## 4 LEARNING FEASIBLE CONSTRAINTS

This section introduces the feasible cost set, essential for resolving the unidentifiability issue (Ng et al., 2000; Metelli et al., 2021) in formulating the ICRL problem. Furthermore, we outline how to quantify the accuracy of an estimated cost set, demonstrating how its estimation error can be bounded by imperfections in estimating environmental dynamics and the expert policy.

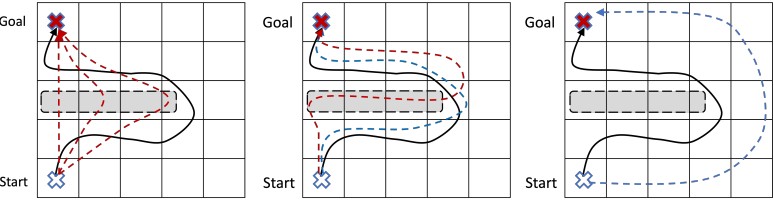

Figure 1: Illustrating the trajectories of the expert policy (black) and exploratory policies (red and blue) in the grid-worlds. The constraint set (gray) is not observable. In the left scenario, exploratory policies reach the goal in shorter paths and thus have larger rewards. In the middle scenario, the exploratory policies' rewards match the expert's. Their trajectories can overlap (red) or mismatch (blue). In the right scenario, exploratory policies result in longer paths that gain fewer rewards.

## 4.1 FEASIBLE COSTS IN CMDP

Since the expert policy satisfies constraints while achieving the highest cumulative rewards, we define feasible cost functions based on two intuitions: 1) if a policy achieves higher rewards than the expert policy (shorter path in Figure 1, left), the underlying constraints *must be violated*, and we can detect unsafe state-action pairs by examining these infeasible trajectories; 2) if a policy achieves the same or lower rewards than the expert policy (equal or longer path in Figure 1, middle & right), this suggests an absence of notable constraint-violating actions, implying that the underlying constraints *may or may not be violated.* To minimize the impact of constraints on the reward-maximizing policy, ICRL focuses on identifying the *minimal* set of constraints necessary to explain expert behaviors (Scobee & Sastry, 2020). In this sense, policies in case 2 are not employed to expand the cost set.

**Lemma 4.1.** *Suppose the expert policy $\pi^E$ of a CMDP $\mathcal{M} \cup c$ is known and the current state is $s$. Let $\mathfrak{A}^E(s)$ denote the set containing all expert actions at state $s$, i.e., $\mathfrak{A}^E(s) = \{a \in \mathcal{A} \mid \pi^E(a|s) > 0\}$. Then, at least one of the following two conditions must be satisfied: 1) The cost function ensures $\mathbb{E}_{\mu_0, \pi^E, P_\mathcal{T}} \left[ \sum_{t=0}^{\infty} \gamma^t c(s_t, a_t) \right] = \epsilon$; 2) $\forall a' \notin \mathfrak{A}^E(s), A_\mathcal{M}^{r,\pi^E}(s, a') \leq 0$.*

The above lemma shows that if there exists an action yielding greater rewards than the expert action, the expert policy's cumulative costs must *reach the threshold*. Thus, enforcing that any higher-reward action must incur greater costs than the expert action is sufficient to establish a constraint-violation condition (i.e., expected return of costs $> \epsilon$). Let $\mathcal{Q}_c = \{(s,a)|Q_{\mathcal{M}\cup c}^{c,\pi^E}(s,a) - V_{\mathcal{M}\cup c}^{c,\pi^E}(s) > 0\}$ denote the set of state-action pairs with higher costs than the expert, given a cost function $c$. In scenarios with hard constraints, it simplifies to: $\mathcal{Q}_c = \{(s,a)|c(s,a) > 0\}$. While capturing cost functions that align with the expert policy, ICRL minimizes $|\mathcal{Q}_c|$ by excluding state-action pairs from case 2 to derive a minimal set of constraints. We formally define the ICRL problem as follows.

**Definition 4.2.** (ICRL problem (Malik et al., 2021)). An ICRL problem is a pair $\mathfrak{P} = (\mathcal{M}, \pi^E)$. A cost representation $c \in [0, C_{\max}]^{\mathcal{S} \times \mathcal{A}}$ is feasible for $\mathfrak{P}$ if $\pi^E$ is an optimal policy for the CMDP $\mathcal{M} \cup c$, i.e., $\pi^E \in \Pi_{\mathcal{M}\cup c}^*$. Let $\mathcal{F}_\mathfrak{P} = \{c|\pi^E \in \Pi_{\mathcal{M}\cup c}^*\}$ denote a general set of feasible cost functions. We denote by $\mathcal{C}_\mathfrak{P}$ the minimal set of feasible cost functions for $\mathfrak{P}$, named feasible cost set that satisfies $\mathcal{C}_\mathfrak{P} = \{c^*|c^* = \arg\min_{c \in \mathcal{F}_\mathfrak{P}} |\mathcal{Q}_c|\}$.

Before formulating the cost function, we introduce the necessary assumptions for different constraints.

**Assumption 4.3.** *Either of the following two statements holds:*
*(i) The constraint in (1) is a hard constraint such that $\epsilon = 0$;*
*(ii) The constraint in (1) is a soft constraint such that $\epsilon > 0$, and the expert policy is deterministic.*

The rationale behind case (ii) is that when the expert policy $\pi^E$ is stochastic at state $s$, we only know $\mathbb{E}_{a' \sim \pi^E}[Q_{\mathcal{M}\cup c}^{c,\pi^E}(s,a')] = V_{\mathcal{M}\cup c}^{c,\pi^E}(s) \geq 0$. In order to determine the value of $Q_{\mathcal{M}\cup c}^{c,\pi^E}(s,a)$ for a specific expert action $a$, additional information is required, such as whether the budget is used up and reward signals of other expert actions. Furthermore, note that in some states, expert policy is not defined if all actions lead to constraint violation. Since feasible cost functions are defined to explain expert behaviors, we do not utilize them to explain the non-existing expert policy in such states. In this work, $\mathcal{S}$ denotes all the states where the expert policy is available. Based on these findings, we are ready to establish the implicit formulation of feasible cost sets.

**Lemma 4.4.** *(Feasible Cost Set Implicit). Under Assumption 4.3, let $\mathfrak{P} = (\mathcal{M}, \pi^E)$ be an ICRL problem. $c$ is a feasible cost function, i.e., $c \in \mathcal{C}_\mathfrak{P}$ if and only if $\forall (s,a) \in \mathcal{S} \times \mathcal{A}$:*

*(1) Expert Consistent $(s,a)$: If $\pi^E(a|s) > 0$, $Q_{\mathcal{M}\cup c}^{c,\pi^E}(s,a) - V_{\mathcal{M}\cup c}^{c,\pi^E}(s) = 0$;*

*(2) Constraint-Violating $(s,a)$: If $\pi^E(a|s) = 0$ and $A_\mathcal{M}^{r,\pi^E}(s,a) > 0$, $Q_{\mathcal{M}\cup c}^{c,\pi^E}(s,a) - V_{\mathcal{M}\cup c}^{c,\pi^E}(s) > 0$;*

*(3) Non-Critical $(s,a)$: If $\pi^E(a|s) = 0$ and $A_\mathcal{M}^{r,\pi^E}(s,a) \leq 0$, $Q_{\mathcal{M}\cup c}^{c,\pi^E}(s,a) - V_{\mathcal{M}\cup c}^{c,\pi^E}(s) \leq 0$.*

Case (1) in the above lemma justifies the rationale behind case (ii) in Assumption 4.3. We proceed to the explicit form of feasible cost sets.

**Lemma 4.5.** *(Feasible Cost Set Explicit). Let $\mathfrak{P} = (\mathcal{M}, \pi^E)$ be an ICRL problem. $c$ is a feasible cost, i.e., $c \in \mathcal{C}_\mathfrak{P}$ if and only if there exists $\zeta \in \mathbb{R}_{>0}^{\mathcal{S} \times \mathcal{A}}$ and $V^c \in \mathbb{R}_{\geq 0}^{\mathcal{S}}$, $\forall (s,a) \in \mathcal{S} \times \mathcal{A}$:*

$$c = A_\mathcal{M}^{r,\pi^E} \zeta + (E - \gamma P_\mathcal{T}) V^c, \tag{2}$$

*where the expansion operator $E : \mathbb{R}^{\mathcal{S}} \rightarrow \mathbb{R}^{\mathcal{S} \times \mathcal{A}}$ satisfies $(Ef)(s, a) = f(s)$. Furthermore,* $\|V^c(s)\|_\infty \leq C_{\max}/(1 - \gamma)$ *and* $\|\zeta\|_\infty \leq C_{\max}/\min_{(s,a)}^+ |A_{\mathcal{M}}^{r,\pi^E}|.$

Intuitively, the first term in (2) penalizes constraint-violating movements that not only deviate from the expert's preference but also have larger rewards (i.e., $A_{\mathcal{M}}^{r,\pi^E} > 0$). This penalty ensures the violation of constraint condition in (1), thereby prohibiting any policies following these movements. The second term $V^c \in \mathbb{R}^{\mathcal{S}}$ can be interpreted as a cost-shaping operator that depends on the CMDP but not on the expert policy. To represent hard constraints, $V^c$ is a zero matrix whose entries are all zeros, i.e., $V^c = \mathbf{0}^{\mathcal{S}}$. However, if the target constraint is soft, we must ensure that $V^c(s) = V_{\mathcal{M} \cup c}^{c,\pi^E}(s)$.

## 4.2 Error Propagation

Our primary objective is to minimize the estimation error of constraints (i.e., the feasible cost sets $\mathcal{C}_{\mathfrak{P}}$). To define this error, based on Lemma 4.5, we first bound the estimation error of the cost functions (i.e., elements in the set) with some theoretically manageable terms in the following.

**Lemma 4.6.** *(Error Propagation). Let $\mathfrak{P} = (\mathcal{M}, \pi^E)$ and $\widehat{\mathfrak{P}} = (\widehat{\mathcal{M}}, \widehat{\pi}^E)$ be two ICRL problems where $\widehat{\mathcal{M}} = (\mathcal{M} \backslash P_{\mathcal{T}}) \cup \widehat{P_{\mathcal{T}}}$. For any $c \in \mathcal{C}_{\mathfrak{P}}$ satisfying $c = A_{\mathcal{M}}^{r,\pi^E} \zeta + (E - \gamma P_{\mathcal{T}})V^c$ and $\|c\|_\infty \leq C_{\max}$ there exists $\widehat{c} \in \mathcal{C}_{\widehat{\mathfrak{P}}}$ satisfying $\|\widehat{c}\|_\infty \leq C_{\max}$:*

$$\widehat{c} = A_{\widehat{\mathcal{M}}}^{r,\widehat{\pi}^E} \frac{\zeta}{1 + \chi/C_{\max}} + (E - \gamma \widehat{P_{\mathcal{T}}}) \frac{V^c}{1 + \chi/C_{\max}}, \tag{3}$$

*where $\chi = \max_{(s,a) \in \mathcal{S} \times \mathcal{A}} \chi(s, a)$ with $\chi(s, a) = \gamma \left| (P_{\mathcal{T}} - \widehat{P_{\mathcal{T}}})V^c \right|(s, a) + \left| A_{\mathcal{M}}^{r,\pi^E} - A_{\widehat{\mathcal{M}}}^{r,\widehat{\pi}^E} \right| \zeta(s, a),$ such that element-wise it holds that:*

$$|c - \widehat{c}|(s, a) \leq \frac{2\chi}{1 + \chi/C_{\max}}. \tag{4}$$

This lemma states the existence of a cost $\widehat{c}$ in the estimated feasible set $\mathcal{C}_{\widehat{\mathfrak{P}}}$ fulfilling the bound composed by two terms. The first term concerns the estimation error of the transition model. The second term depends on both the expert policy approximation and the estimated MDP, which can be further decomposed as follows:

**Lemma 4.7.** *For a given policy $\pi$, let $A_{\mathcal{M}}^{r,\pi}$ denote the reward advantage function based on the original CMDP $\mathcal{M} \cup c$. For an estimated policy $\widehat{\pi}$, let $A_{\widehat{\mathcal{M}}}^{r,\widehat{\pi}}$ denote the reward advantage function based on the estimated MDP $\widehat{\mathcal{M}}$ and estimated cost function $\widehat{c}$. Then, we have*

$$\left| A_{\mathcal{M}}^{r,\pi} - A_{\widehat{\mathcal{M}}}^{r,\widehat{\pi}} \right| \leq \frac{2\gamma}{1 - \gamma} \left| (\widehat{P_{\mathcal{T}}} - P_{\mathcal{T}})V_{\widehat{\mathcal{M}}}^{r,\widehat{\pi}} \right| + \frac{\gamma(1 + \gamma)}{1 - \gamma} \left| (\pi - \widehat{\pi})P_{\mathcal{T}}V_{\mathcal{M}}^{r,\pi} \right|.$$

With the estimation error of cost functions bounded as in Lemma 4.6, we next analyze the estimation errors of optimal policies $\pi^*$ between CMDP with true cost and estimated cost, i.e., $\mathcal{M} \cup c$ and $\mathcal{M} \cup \widehat{c}$. This error quantifies the extent to which the estimated cost function captures expert behaviors.

**Lemma 4.8.** *For every given policy $\pi$, the first inequality below holds element-wise. For every optimal policies $\pi^* \in \Pi_{\mathcal{M} \cup c}^*$ and $\widehat{\pi}^* \in \Pi_{\widehat{\mathcal{M}} \cup \widehat{c}}^*$ of CMDPs $\mathcal{M} \cup c$ and $\widehat{\mathcal{M}} \cup \widehat{c}$ respectively, the second inequality below holds.*

$$\left| Q_{\mathcal{M} \cup c}^{c,\pi} - Q_{\mathcal{M} \cup \widehat{c}}^{c,\pi} \right| \leq \left| (I_{\mathcal{S} \times \mathcal{A}} - \gamma P_{\mathcal{T}} \pi)^{-1} |c - \widehat{c}| \right|,$$

$$\max_{\pi \in \{\widehat{\pi}^*, \pi^*\}} \left\| Q_{\mathcal{M} \cup c}^{c,\pi} - Q_{\mathcal{M} \cup \widehat{c}}^{c,\pi} \right\|_\infty \leq \frac{1}{1 - \gamma} \|c - \widehat{c}\|_\infty.$$

With the above results, we can define the *optimality* of the estimated cost sets based on the Probably Approximately Correct (PAC) condition (Haussler, 1992; Mohri et al., 2018). The estimated feasible set $\mathcal{C}_{\widehat{\mathfrak{P}}}$ is "close" to the exact feasible set $\mathcal{C}_{\mathfrak{P}}$, if for every cost $c \in \mathcal{C}_{\mathfrak{P}}$, there exists one estimated cost $\widehat{c} \in \mathcal{C}_{\widehat{\mathfrak{P}}}$ that is "close" to $c$, and vice versa.

**Definition 4.9.** (Optimality Criterion). Let $\mathcal{C}_{\mathfrak{P}}$ be the exact feasible set and $\mathcal{C}_{\widehat{\mathfrak{P}}}$ be the feasible set recovered after observing $n \geq 0$ samples collected in the source $\mathcal{M}$ and $\pi^E$. We say that an algorithm for ICRL is $(\varepsilon, \delta, n)$-correct if with probability at least $1 - \delta$, it holds that:

$$\inf_{\widehat{c} \in \mathcal{C}_{\widehat{\mathfrak{P}}}} \sup_{\pi^* \in \Pi^*_{\mathcal{M} \cup c}} \left| Q^{c, \pi^*}_{\mathcal{M} \cup c}(s, a) - Q^{c, \pi^*}_{\mathcal{M} \cup \widehat{c}}(s, a) \right| \leq \varepsilon, \forall c \in \mathcal{C}_{\mathfrak{P}},$$

$$\inf_{c \in \mathcal{C}_{\mathfrak{P}}} \sup_{\widehat{\pi}^* \in \Pi^*_{\widehat{\mathcal{M}} \cup \widehat{c}}} \left| Q^{c, \widehat{\pi}^*}_{\mathcal{M} \cup c}(s, a) - Q^{c, \widehat{\pi}^*}_{\mathcal{M} \cup \widehat{c}}(s, a) \right| \leq \varepsilon, \forall \widehat{c} \in \mathcal{C}_{\widehat{\mathfrak{P}}},$$

where $\pi^*$ is an optimal policy in $\mathcal{M} \cup c$ and $\widehat{\pi}^*$ is an optimal policy in $\widehat{\mathcal{M}} \cup \widehat{c}$.

The above definition aims to ensure the estimation error of cost does not compromise the optimality of the expert policy. The first condition manifests *completeness*, since the recovered feasible cost set needs to track every potential true cost function. The second condition expresses *accuracy* since any recovered cost function must be in close proximity to a viable true cost function, preventing an unnecessarily large recovered feasible set. The dual requirements are inspired by the PAC optimality criterion in (Metelli et al., 2021; Lindner et al., 2022).

## 5 EFFICIENT EXPLORATION FOR ICRL

In this section, we introduce algorithms for efficient exploration by leveraging the aforementioned cost set and estimation error. Our objective is to collect high-quality samples from interactions with the environment, thereby improving the accuracy of our cost set estimations. Unlike most existing ICRL works (Papadimitriou et al., 2023; Liu et al., 2022a) that rely on a generative model for collecting samples, our exploration strategy must determine *which* states require more frequent visits and *how* to traverse to them starting from the initial state $s_0$. To achieve this goal, we first define the estimated transition model and the expert policy (Section 5.1), based on which we develop a BEAR (Bounded Error Aggregate Reduction) strategy algorithm (Section 5.2) and a PCSE (Policy-Constrained Strategic Exploration) algorithm (Section 5.3) for solving ICRL problems, respectively.

### 5.1 ESTIMATING TRANSITION DYNAMICS AND EXPERT MODEL

We consider a model-based setting where the agent strategically explores the environment to learn transition dynamics and expert policy. These components are vital for bounding the estimation error of the feasible cost set (Lemma 4.6). To achieve this, we record the returns of a state-action pair $(s, a)$ by observing a next state $s' \sim P(\cdot|s, a)$, and the preference of expert agents $a_E \sim \pi^E(\cdot|s)$ in each visited state. For iteration $\forall k$, we denote by $n_k(s, a, s')$ the number of times we observe the transition $(s, a, s')$. Denote $n_k(s, a) = \sum_{s' \in \mathcal{S}} n_k(s, a, s')$ and $n_k(s) = \sum_{a \in \mathcal{A}} n_k(s, a)$. For the expert policy and the transition model estimation, we define the *cumulative* counts $N_k(s, a, s') = \sum_{j=1}^{k} n_j(s, a, s')$, $N_k(s, a) = \sum_{j=1}^{k} n_j(s, a)$ and $N_k(s) = \sum_{j=1}^{k} n_j(s)$. Accordingly, we can represent the estimated transition model and expert policy as:

$$\widehat{P}_{\mathcal{T}k}(s'|s, a) = \frac{N_k(s, a, s')}{N_k^+(s, a)}, \quad \widehat{\pi}_k^E(a|s) = \frac{N_k(s, a)}{N_k^+(s)}, \tag{5}$$

where $x^+ = \max\{1, x\}$. With these estimations, we derive the confidence intervals for the transition model and expert policy using the Hoeffding inequality (see Lemma C.5). We prove that the true transition model and the expert policy fall into these intervals with high probability. Based on these results, we derive an upper bound on the estimation error of feasible cost sets and prove that this upper bound can be guaranteed with high probability as follows:

**Lemma 5.1.** *Let $\delta \in (0, 1)$, with probability at least $1 - \delta$, for any pair of cost functions $c \in \mathcal{C}_{\mathfrak{P}}$ and $\widehat{c}_k \in \mathcal{C}_{\widehat{\mathfrak{P}}_k}$ at iteration $k$, we have*

$$|c(s, a) - \widehat{c}_k(s, a)| \leq \mathcal{C}_k(s, a), \quad \mathcal{C}_k(s, a) = \min \left\{ \frac{2\sigma \sqrt{\frac{\ell_k(s, a)}{2N_k^+(s, a)}}}{1 + \sigma/C_{\max} \sqrt{\frac{\ell_k(s, a)}{2N_k^+(s, a)}}}, C_{\max} \right\}. \tag{6}$$

*where $\sigma = \frac{\gamma C_{\max} \left( R_{\max}(3 + \gamma)/\min^+ \left| A_{\mathcal{M}}^{r, \pi^E} \right| + (1 - \gamma) \right)}{(1 - \gamma)^2}$ and $\ell_k(s, a) = \log \left( \frac{36SA(N_k^+(s, a))^2}{\delta} \right)$.*

It is worth noting that $\mathcal{C}_k(s,a)$ typically decreases after the number of samples collected for a specific $(s,a)$ pair reaches a peak. To efficiently allocate a fixed number of samples to meet the demand of Definition 4.9, we introduce the exploration strategy next.

## 5.2 EXPLORATION VIA REDUCING BOUNDED ERRORS

Based on the above upper bound, we are ready to design algorithms for efficiently solving the ICRL problem. Since our primary goal is to fulfill the PAC-condition in Definition 4.9, we begin by establishing an upper bound on the estimation error, which pertains to the disparity for the performance of optimal policy $\pi^*$ between CMDP with true cost and CMDP with estimated cost at iteration $k$, i.e., $\mathcal{M} \cup c$ and $\mathcal{M} \cup \widehat{c}_k$. Our key results are presented as follows:

**Lemma 5.2.** *At iteration $k$, let $e_k(s,a;\pi^*) = |Q_{\mathcal{M}\cup c}^{c,\pi^*}(s,a) - Q_{\mathcal{M}\cup\widehat{c}_k}^{c,\pi^*}(s,a)|$ defines the estimation error of discounted cumulative costs within the true CMDP\c $\mathcal{M}$. For any policy $\pi^* \in \Pi_{\mathcal{M}\cup c}^*$, we upper bound the above estimation error $e_k(\cdot)$ as follows:*

$$\|e_k(s,a;\pi^*)\|_\infty \leq \left\|\mu_0^T(I_{\mathcal{S}\times\mathcal{A}} - \gamma P_\mathcal{T}\pi)^{-1}\mathcal{C}_k\right\|_\infty. \tag{7}$$

To reduce this error bound, we introduce BEAR exploration strategy for ICRL in Algorithm 1 (represented in teal color), which explores to reduce the bounded error. This is equivalent to solving the RL problem defined by $\mathcal{M}^{\mathcal{C}_k} = (\mathcal{M}\backslash r) \cup \mathcal{C}_k$, where we replace the reward $r$ in MDP $\mathcal{M}$ with $\mathcal{C}_k$. We can use any RL solver to find the exploration policy in practice. We show in Corollary C.6 that the exploration algorithm converges (satisfies Definition 4.9) when either of the following statements is satisfied:

$$(i) \quad \frac{1}{1-\gamma} \max_{(s,a)\in\mathcal{S}\times\mathcal{A}} \mathcal{C}_k(s,a) \leq \varepsilon, \qquad (ii) \quad \left\|\mu_0^T(I_{\mathcal{S}\times\mathcal{A}} - \gamma P_\mathcal{T}\pi)^{-1}\mathcal{C}_k\right\|_\infty \leq \varepsilon. \tag{8}$$

**Sample Complexity.** Next, we analyze the sample complexity of Algorithm BEAR. The updated accuracy $\varepsilon_k$ in Algorithm 1 equals to $(i)$ of (8). Let $\eta_k^h(s,a|s_0), h \in [n_{\max}]$ be the probability of state-action pair $(s,a)$ reached in the $h$-th step following a policy $\pi_k \in \Pi_{\mathcal{M}^{c_k}}$ starting in state $s_0$. We can compute it recursively:

$$\eta_k^0(s,a|s_0) := \pi_k(a|s)\mathbb{1}_{\{s=s_0\}}, \quad \eta_k^{h+1}(s,a|s_0) := \sum_{a',s'} \pi_k(a|s)P_\mathcal{T}(s|s',a')\eta_k^h(s',a'|s_0),$$

where $\pi_k$ is the exploration policy in iteration $k$. We then define the pseudo-counts that are crucial to deal with the uncertainty of the transition dynamics in our analysis.

**Definition 5.3.** (*Pseudo-counts*) We introduce the pseudo-counts of visiting a specific state-action pair $(s,a)$ in the $h$-th step within the first $k$ iterations as:

$$\bar{N}_k(s,a) = \mu_0 \sum_{h=1}^{n_{\max}} \sum_{i=1}^k \eta_i^h(s,a|s_0).$$

Similar to (5), we define $\bar{N}_k^+(s,a) = \max\{0, \bar{N}_k(s,a)\}$. The following lemma upper bounds the estimation error of feasible costs with the pseudo-counts under a certain confidence interval.

**Lemma 5.4.** *With probability at least $1 - \delta/2$, $\forall s,a,h,k \in \mathcal{S}\times\mathcal{A}\times[0,n_{\max}]\times\mathbb{N}^+$, we have:*

$$\min\left\{\sigma\sqrt{\frac{\ell_k(s,a)}{2N_k^+(s,a)}}, C_{\max}\right\} \leq \check{\sigma}\sqrt{\frac{2\bar{\ell}_k(s,a)}{\bar{N}_k^+(s,a)}}, \tag{9}$$

*where $\bar{\ell}_k(s,a) = \log(36SA(\bar{N}_k^+(s,a))^2/\delta)$ and $\check{\sigma} = \max\{\sigma, \sqrt{2}C_{\max}\}$.*

Subsequently, the sample complexity of Algorithm 1 is presented as follows:

**Theorem 5.5.** (*Sample Complexity of BEAR*). *If Algorithm BEAR terminates at iteration $K$ with the updated accuracy $\varepsilon_K$, then with probability at least $1 - \delta$, it fulfills Definition 4.9 with a number of samples upper bounded by*

$$n \leq \widetilde{\mathcal{O}}\left(\frac{\check{\sigma}^2 SA}{(1-\gamma)^2\varepsilon_K^2}\right).$$

The above theorem has taken into account the sample complexity of the RL phase. In fact, further improvements can be made to enhance the algorithm's performance.

## 5.3 Exploration via Constraining Candidate Policies

The above exploration strategy has limitations, as it explores to minimize uncertainty across *all policies*, which is not aligned with our primary focus of reducing uncertainty for *potentially optimal policies*. As a result, this approach places an additional burden on sample efficiency. To address these limitations, we propose PCSE for ICRL in Algorithm 1 (represented in purple color). Specifically,

---

**Algorithm 1** BEAR and PCSE for ICRL in an unknown environment

**Input:** significance $\delta \in (0, 1)$, target accuracy $\varepsilon$, maximum number of samples per iteration $n_{\max}$;
Initialize $k \leftarrow 0$, $\varepsilon_0 = \frac{1}{1-\gamma}$;
**while** $\varepsilon_k > \varepsilon$ **do**
    Solve RL problem defined by $\mathcal{M}^{\mathcal{C}_k}$ to obtain the exploration policy $\pi_k$;
    Solve optimization problem in (10) to obtain the exploration policy $\pi_k$;
    Explore with $\pi_k$ for $n_e$ episodes;
    For each episode, collect $n_{\max}$ samples from $(s, a)$;
    Update accuracy $\varepsilon_{k+1} = \max_{(s,a) \in \mathcal{S} \times \mathcal{A}} \mathcal{C}_{k+1}(s, a)/(1 - \gamma)$;
    Update accuracy $\varepsilon_{k+1} = \|\mu_0^T (I_{\mathcal{S} \times \mathcal{A}} - \gamma P_{\mathcal{T}} \pi)^{-1} \mathcal{C}_k\|_\infty$;
    Update $\widehat{\pi}_{k+1}^E$ and $\widehat{P_{\mathcal{T}}}_{k+1}$ in (5);
    $k \leftarrow k + 1$.
**end while**

---

we intentionally constrain the search for policies to those yielding a value function at iteration $k$ close to the estimated optimal one. Thus we focus only on the plausibly optimal policies and formulate the optimization problem as:

$$\varepsilon_{k+1} = \sup_{\substack{\mu_0 \in \Delta^{\mathcal{S}} \\ \pi \in \Pi_k}} \mu_0^T (I_{\mathcal{S} \times \mathcal{A}} - \gamma P_{\mathcal{T}} \pi) \mathcal{C}_{k+1}, \quad \text{s.t.} \quad \Pi_k = \Pi_k^c \cap \Pi_k^r, \tag{10}$$

$$\Pi_k^c = \left\{ \pi \in \Delta_{\mathcal{S}}^{\mathcal{A}} : \sup_{\mu_0 \in \Delta^{\mathcal{S}}} \mu_0^T \left( V_{\widehat{\mathcal{M}}_k \cup \widehat{c}_k}^{c, \pi} - V_{\widehat{\mathcal{M}}_k \cup \widehat{c}_k}^{c, *} \right) \leq 4\varepsilon_k + \epsilon \right\},$$

$$\Pi_k^r = \left\{ \pi \in \Delta_{\mathcal{S}}^{\mathcal{A}} : \inf_{\mu_0 \in \Delta^{\mathcal{S}}} \mu_0^T \left( V_{\widehat{\mathcal{M}}_k}^{r, \pi} - V_{\widehat{\mathcal{M}}_k}^{r, \widehat{\pi}_k^*} \right) \geq \mathfrak{R}_k \right\},$$

where $\mathfrak{R}_k = \frac{2\gamma R_{\max}}{(1-\gamma)^2} \|P_{\mathcal{T}} - \widehat{P_{\mathcal{T}}}_k\|_\infty + \frac{\gamma R_{\max}}{(1-\gamma)^2} \|(\pi^* - \widehat{\pi}_k^*)\|_\infty$.

The rationale in $\Pi_k$ can be attributed to the intersection of two aspects: 1) $\Pi_k^c$ constrains exploration policies to visit states within an additional budget, thereby ensuring *resilience* to estimation error when searching for optimal policies; 2) $\Pi_k^r$ states that exploration policies should focus on states with potentially higher cumulative rewards, where possible constraints lie. As the estimation error decreases, the gap (i.e., $\mathfrak{R}_k$) also diminishes, eventually converging to zero, which ensures the *optimality* of constrained policies. We have shown in Appendix C.12 that optimality policies can be captured by subsequent $\Pi_k$.

To solve the optimization problem (10), we represent its Lagrangian objective as $L(\rho_{\mathcal{M}}^\pi, \lambda) =$

$$-\langle \rho_{\mathcal{M}}^\pi, \mathcal{C}_{k+1} \rangle + \lambda_2 \left( (1-\gamma)(V_{\widehat{\mathcal{M}}_k}^{\widehat{\pi}_k^*} + \mathfrak{R}_k) - \langle \rho_{\mathcal{M}}^\pi, r \rangle \right) + \lambda_1 \left( -(1-\gamma)(V_{\widehat{\mathcal{M}}_k \cup \widehat{c}_k}^{c, *} + 4\varepsilon_k + 2\epsilon) + \langle \rho_{\mathcal{M}}^\pi, \widehat{c}_k \rangle \right),$$

where $\lambda = [\lambda_1, \lambda_2]^T$ records two Lagrangian multipliers. The dual problem of (10) can be defined as

$$\min_{\rho_{\mathcal{M}}^\pi} \max_{\lambda \geq 0} L(\rho_{\mathcal{M}}^\pi, \lambda). \tag{11}$$

To solve this dual problem, we assume that Slater's condition is fulfilled and we follow the two-timescale stochastic approximation (Borkar & Konda, 1997; Konda & Tsitsiklis, 1999). The following two gradient steps are alternately conducted until convergence,

$$\rho_{\mathcal{M}, k+1}^\pi = \rho_{\mathcal{M}, k}^\pi - a_k (L_\rho'(\rho_{\mathcal{M}, k}^\pi, \lambda_k) + W_k), \quad \lambda_{k+1} = \lambda_k + b_k (L_\lambda'(\rho_{\mathcal{M}, k}^\pi, \lambda_k) + U_k),$$

where coefficients $a_k \ll b_k$, satisfying $\sum_k a_k = \sum b_k = \infty$, $\sum a_k^2 < \infty$ and $\sum b_k^2 < \infty$. $W_k$ and $U_k$ are two zero-mean noise sequences. Under this condition, the convergence is guaranteed

in the limit (Borkar, 2009). At each time step $k$, the exploration policy is calculated as: $\pi_k(a|s) = \rho^\pi_{\mathcal{M},k}(s,a)/\sum_a \rho^\pi_{\mathcal{M},k}(s,a)$.

**Sample Complexity.** In the following theorem, we prove that PCSE for ICRL fulfills the PAC-condition in Definition 4.9 and we show its sample complexity. To present this result, we define the cost advantage function $A^{c,*}_{\widehat{\mathcal{M}}\cup\tilde{c}}(s,a) = Q^{c,*}_{\widehat{\mathcal{M}}\cup\tilde{c}}(s,a) - V^{*,c}_{\widehat{\mathcal{M}}\cup\tilde{c}}(s)$, in which $\tilde{c} \in \arg\min_{c\in\mathcal{C}_{\mathfrak{P}}} \max_{(s,a)\in\mathcal{S}\times\mathcal{A}} |c(s,a) - \widehat{c}_K(s,a)|$ is the cost function in the exact cost feasible set $\mathcal{C}_{\mathfrak{P}}$ closest to the estimated cost function $\widehat{c}_K(s,a)$ at the terminating iteration $K$.

**Theorem 5.6.** *(Sample Complexity of PCSE). If Algorithm PCSE terminates at iteration $K$ with accuracy $\varepsilon_K$ and the accuracy of previous iteration is $\varepsilon_{K-1}$, then with probability at least $1-\delta$, it fulfills Definition 4.9 with a number of samples upper bounded by*

$$n \le \widetilde{\mathcal{O}}\left(\min\left\{\frac{\check{\sigma}^2 SA}{(1-\gamma)^2\varepsilon_K^2}, \frac{\sigma^2(6\varepsilon_{K-1}+\epsilon)^2 SA}{\min_{(s,a)}\left(A^{c,*}_{\widehat{\mathcal{M}}\cup\tilde{c}}(s,a)\right)^2\varepsilon_K^2}\right\}\right).$$

The first term matches the sample complexity of the BEAR strategy since both strategies explore for the same purpose. The second term depends on the ratio $(6\varepsilon_{K-1}+\epsilon)/\varepsilon_K$ and the minimum cost advantage function $\min_{(s,a)} A^{c,*}_{\widehat{\mathcal{M}}\cup\tilde{c}}$. The ratio depends on both $n_{\max}$ and $n_e$. If the two values are high, the ratio is high and the algorithm tends to uniformly sample every state-action pair. Otherwise, the ratio is small due to the fact that $\varepsilon_{K-1}$ is an accumulation of $\mathcal{C}_{K-1}$ (generally larger than $\mathcal{C}_K$). A smaller $\epsilon$, namely a tighter constraint, benefits the sample efficiency. The cost advantage function $\min_{(s,a)} A^{c,*}_{\widehat{\mathcal{M}}\cup\tilde{c}}$ shows that the larger the suboptimality gap, the easier to infer the constraint.

## 6 EMPIRICAL EVALUATION

We empirically compare our algorithms against other methods across both discrete and continuous environments, where the agent aims to navigate from a starting location to a target location (where it receives a positive reward) while satisfying the constraint condition.

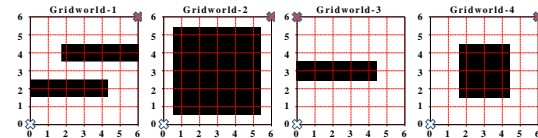

Figure 2: Four different Gridworld environments.

**Experiment Settings.** The evaluation metrics include: 1) *discounted cumulative rewards*, which measure the optimality of the learned policy; 2) *discounted cumulative costs*, which assess the safety of the learned policy; and 3) *Weighted Generalized Intersection over Union (WGIoU)* (see Appendix D.2), which evaluates the similarity between inferred constraints and ground-truth constraints.

**Comparison Methods.** We compare our exploration algorithms, i.e., BEAR and PCSE, with four other exploration strategies. Results of two baselines: random exploration and $\epsilon$-greedy exploration are demonstrated in Figure 3. Results of two other baselines: maximum-entropy exploration and upper confidence bound exploration are shown in Appendix Figure 5.

### 6.1 EVALUATION UNDER DISCRETE ENVIRONMENTS

Figure 2 illustrates four discrete testing environments, each characterized by distinct constraints. The white, red, and black markers indicate the starting, target, and constrained locations, respectively. The expert policy is trained under ground-truth constraints, while the ICRL algorithms are examined when these constraints are not available. Note that these environments are stochastic so that the environment executes a randomized sampled action with a specific probability ($p = 0.05$). Figure 3 shows the training process of three metrics for six exploration strategies in four Gridworld environments, along with the performance of expert policy (represented by the grey line). It can be shown that the performance of the optimal policy in $\mathcal{M} \cup \widehat{c}$ gradually converges to the performance of the optimal policy in $\mathcal{M} \cup c$. Also, we find that PCSE (represented by the red curve) exhibits the highest sample efficiency while achieving similar performance among the six exploration strategies. In Gridworld-2 and Gridworld-4, WGIoU converges to a degree of similarity less than 1 (ground-truth). This is because ICRL emphasizes the identification of the *minimal* set of constraints necessary to explain expert behaviors. We demonstrate the learned constraints in the rightmost column of Figure 8 and 10. The learned constraints are captured because visiting these states leads to higher cumulative rewards,

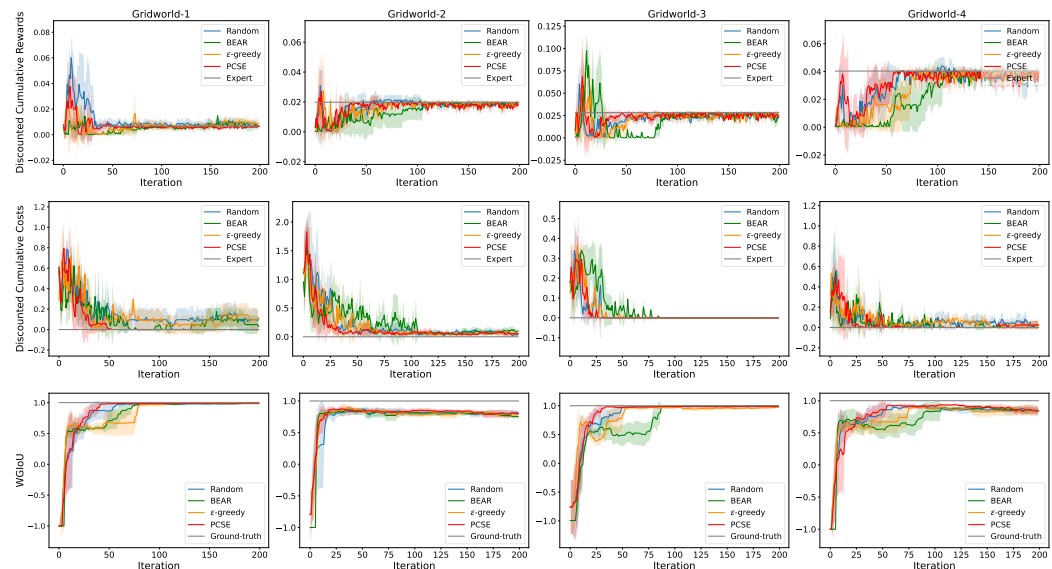

Figure 3: Training curves of discounted cumulative rewards (top), costs (middle), and WGIoU (bottom) for four exploration strategies in four Gridworld environments.

whereas other uncaptured ground-truth constraints do not influence the optimality of expert behavior. Constraint learning processes of six strategies are demonstrated in Figure 7 to 10 in Appendix E.1.

## 6.2 EVALUATION UNDER CONTINUOUS ENVIRONMENTS

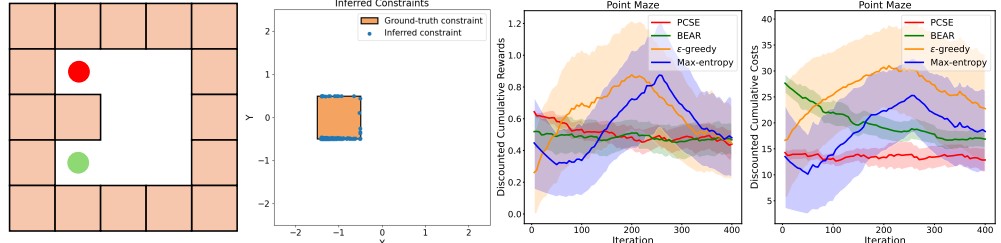

Figure 4: Point Maze environment, inferred constraints, discounted cumulative rewards and costs.

Figure 4 (leftmost) illustrates the continuous Point Maze environment, where the green agent has a continuous state space. The agent's goal is to reach the red ball inside the maze with pink walls. The environment is stochastic due to the noises imposed on the observed states. Figure 4 (middle left) demonstrates the inferred constraints (represented by blue dots) obtained through PCSE, with the center of the maze designated at $(0, 0)$. Figure 4 (middle right and rightmost) reports the discounted cumulative rewards and costs during training. Check Appendix E.2 for more experimental details.

## 7 CONCLUSIONS

This paper introduces a strategically efficient exploration framework for ICRL problems. We conduct theoretical analysis to investigate the influence of estimation errors in expert policy and environmental dynamics on the estimation of constraints. Building upon this, we propose two exploration strategies, namely BEAR and PCSE. Both algorithms actively explore the environment to minimize the aggregated bounded error of cost estimation. Moreover, PCSE goes a step further by constraining the exploration policies to plausibly optimal ones, thus enhancing the overall efficiency. We provide tractable sample complexity analyses for both algorithms. To validate the effectiveness of our method, we perform empirical evaluations in various environments. Several future research directions deserve attention to address the limitations of this paper: 1) extending this work to finite-horizon settings and deriving lower bounds for sample complexities, and 2) analyzing the transferability of constraint information.

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

# APPENDIX

## Table of Contents

## A NOTATION AND SYMBOLS

In Table 1, we report the explicit definition of notation and symbols applied in our paper.

Table 1: Overview of notation and symbols

| Symbol | Name | Signature |
|--------|------|-----------|
| $\mathcal{M}$ | CMDP without knowing the cost (CMDP$\setminus c$) | $(\mathcal{S}, \mathcal{A}, P_{\mathcal{T}}, r, \epsilon, \mu_0, \gamma)$ |
| $\mathcal{M} \cup c$ | CMDP | $(\mathcal{S}, \mathcal{A}, P_{\mathcal{T}}, r, c, \epsilon, \mu_0, \gamma)$ |
| $\mathcal{S}$ | State space | / |
| $\mathcal{A}$ | Action space | / |
| $P_{\mathcal{T}}$ | Transition model | $\Delta_{\mathcal{S} \times \mathcal{A}}^{\mathcal{S}}$ |
| $s_0$ | Initial state | $\mathcal{S}$ |
| $\pi$ | Policy | $\Delta_{\mathcal{S}}^{\mathcal{A}}$ |
| $\pi^E$ | Expert policy | $\Delta_{\mathcal{S}}^{\mathcal{A}}$ |
| $r$ | Reward function | $[0, R_{\max}]^{\mathcal{S} \times \mathcal{A}}$ |
| $c$ | Cost function | $[0, C_{\max}]^{\mathcal{S} \times \mathcal{A}}$ |
| $\epsilon$ | Threshold of constraint | $\mathbb{R}^{\mathcal{S}}$ |
| $V_{\mathcal{M}}^{r,\pi}$ | Reward state-value function of $\pi$ in $\mathcal{M}$ | $\mathbb{R}^{\mathcal{S}}$ |
| $Q_{\mathcal{M}}^{r,\pi}$ | Reward action-value function of $\pi$ in $\mathcal{M}$ | $\mathbb{R}^{\mathcal{S} \times \mathcal{A}}$ |
| $A_{\mathcal{M}}^{r,\pi}$ | Reward advantage function of $\pi$ in $\mathcal{M}$ | $\mathbb{R}^{\mathcal{S} \times \mathcal{A}}$ |
| $V_{\mathcal{M} \cup c}^{c,\pi}$ | Cost state-value function of $\pi$ in $\mathcal{M} \cup c$ | $\mathbb{R}^{\mathcal{S}}$ |
| $Q_{\mathcal{M} \cup c}^{c,\pi}$ | Cost action-value function of $\pi$ in $\mathcal{M} \cup c$ | $\mathbb{R}^{\mathcal{S} \times \mathcal{A}}$ |
| $A_{\mathcal{M} \cup c}^{c,\pi}$ | Cost advantage function of $\pi$ in $\mathcal{M} \cup c$ | $\mathbb{R}^{\mathcal{S} \times \mathcal{A}}$ |
| $\mathcal{C}_{\mathfrak{P}}$ | Exact feasible set | / |
| $\mathcal{C}_{\widehat{\mathfrak{P}}}$ | Recovered feasible set | / |
| $\eta_k^h(s, a \mid s_0)$ | State action pair visitation frequencies | $\Delta^{\mathcal{S} \times \mathcal{A}}$ |
| $\rho_{\mathcal{M}}^{\pi}$ | Occupancy measure of $\pi$ in $\mathcal{M}$ | $\Delta^{\mathcal{S} \times \mathcal{A}}$ |
| $\varepsilon$ | Target accuracy | $\mathbb{R}^+$ |
| $\delta$ | Significancy | $(0, 1)$ |
| $n_e$ | Number of exploration episodes | $\mathbb{N}^+$ |
| $E$ | Expansion operator | $\mathbb{R}^{\mathcal{S}} \to \mathbb{R}^{\mathcal{S} \times \mathcal{A}}$ |
| $I_{\mathcal{S} \times \mathcal{A}}$ | Identity matrix on $\mathcal{S} \times \mathcal{A}$ | / |
| $I_{\mathcal{S}}$ | Identity matrix on $\mathcal{S}$ | / |
| $[a]$ | Set that contains integers from 0 to $a$ | $\{0, 1, \dots, a\}, a \in \mathbb{N}$ |

## B ADDITIONAL RELATED WORKS

**Sample Efficiency**. Sample-efficient algorithms have been explored across various RL directions, yielding significant advancements. To find the minimal structural assumptions that empower sample-efficient learning, Jin et al. (2021) introduced the Bellman Eluder (BE) dimension and proposed a sample-efficient algorithm for problems with low BE dimension. Liu et al. (2024b) introduced a sample-efficient RL framework called Maximize to Explore (MEX), which reduces computational cost and enhances compatibility. In the field of imitation learning, Liu et al. (2022b) addressed both online and offline settings, proposing optimistic and pessimistic generative adversarial policy imitation algorithms with tractable regret bounds. In the realm of model-free RL, Jin et al. (2018) developed a Q-learning algorithm with Upper Confidence Bound (UCB) exploration, achieving a regret bound of $\sqrt{T}$ in episodic MDPs. Wachi et al. (2018) modeled state safety values using a Gaussian Process (GP) and proposed a more efficient approach to balance the trade-off between exploring the safety function, exploring the reward function, and exploiting knowledge to maximize rewards. In the context of constrained reinforcement learning (CRL), Miryoosefi & Jin (2022) bridged reward-free RL and CRL, providing sharp sample complexity results for CRL in tabular Markov Decision Processes (MDPs). Focusing on episodic finite-horizon Constrained MDPs (CMDPs), Kalagarla et al. (2021) established a probably approximately correct (PAC) guarantee on the number of episodes required to find a near-optimal policy, with a linear dependence on the state and action

spaces and a quadratic dependence on the time horizon. From a meta-learning perspective, Liu & Zhu (2023) framed the problem of learning an expert's reward function and constraints from few demonstrations as a bi-level optimization, introducing a provably efficient algorithm to learn a meta-prior over reward functions and constraints. In terms of sample efficiency in IRL, (Lazzati et al., 2024a) redefines offline IRL by introducing the feasible reward set to address limited data coverage, proposing approaches to ensure inclusion monotonicity through pessimism. (Lazzati & Metelli, 2024) extends IRL to Utility Learning (UL), introducing a framework for capturing agents' risk attitudes via utility functions. (Lazzati et al., 2024b) tackles scalability in online IRL by introducing reward compatibility and a state-space-independent algorithm for Linear MDPs, bridging IRL and Reward-Free Exploration (RFE). For misspecification in IRL, (Skalse & Abate, 2023) provides a framework and tools to evaluate the robustness of standard IRL models (e.g., optimality, Boltzmann rationality) to misspecification, ensuring reliable inferences from real-world data. (Skalse & Abate, 2024) quantifies IRL's sensitivity to behavioral model inaccuracies, showing that even small misspecifications can result in significant errors in inferred reward functions.

**Constraint Inference.** Constraint learning in reinforcement learning has advanced significantly to address shared safety requirements and improve scalability and efficiency. Chou et al. (2018) introduced a method to infer shared constraints across tasks using safe and unsafe trajectories, leveraging hit-and-run sampling and integer programming with theoretical guarantees. Kim & Oh (2022) proposed Off-Policy TRC, a sample-efficient RL method with CVaR constraints that addresses distributional shift via surrogate functions and trust-region constraints, achieving high returns and safety in complex tasks. To ensure stable convergence, Moskovitz et al. (2023) developed ReLOAD, which guarantees last-iterate convergence and overcomes limitations of gradient-based methods in CRL. For scenarios with unknown rewards and dynamics, Lindner et al. (2024) introduced a CMDP method that constructs a convex safe set from safe demonstrations, enabling task transferability and outperforming IRL-based approaches. Kim et al. (2024) extended IRL framework to infer tighter safety constraints from diverse expert demonstrations, addressing the ill-posed nature of constraint learning and enhancing multi-task generalization. Our approach infers a feasible cost set encompassing all cost functions consistent with the provided demonstrations, eliminating reliance on additional information to address the inherent ill-posedness of inverse problems. In contrast, prior works either require multiple demonstrations across diverse environments or rely on additional settings to ensure the uniqueness of the recovered constraints. This feasible set approach can focus on analyzing the intrinsic complexity of the ICRL problem only, without being obfuscated by other factors, resulting in solid theoretical guarantees (Lazzati et al., 2024b).

## C PROOFS OF THEORETICAL RESULTS IN THE MAIN PAPER

In this section, we provide detailed proofs of theoretical results in the main paper.

### C.1 PROOF OF LEMMA 4.1

*Proof.* If neither case happens, i.e., $\mathbb{E}_{\mu_0, \pi^E, P_\mathcal{T}}\left[\sum_{t=0}^\infty \gamma^t c(s_t, a_t)\right] < \epsilon$ and $\exists\, a' \in \mathcal{A}$ that satisfies both $a' \notin \mathfrak{A}^E(s)$ and $A_\mathcal{M}^{r, \pi^E}(s, a') > 0$, we can always construct a new policy, which only differs from the expert policy $\pi^E$ in state $s$, as $\pi'(a|s) = \begin{cases} \theta & , a = a' \\ 1 - \theta, & a \sim \pi^E \end{cases}$. There must $\exists\, \theta \in (0, 1]$ that uses some (or all) of the leftover budget $\tau = \epsilon - \mathbb{E}_{\mu_0, \pi^E, P_\mathcal{T}}\left[\sum_{t=0}^\infty \gamma^t c(s_t, a_t)\right]$ while having a larger cumulative reward, which makes $\pi^E$ not an optimal policy. This makes a contradiction.

The existence of such $\theta$ can be proved as follows. By recursively using the Bellman Equation, we can obtain

$$\mathbb{E}_{\mu_0}\left[V_{\mathcal{M} \cup c}^{c, \pi^E}(s_0)\right] = \alpha(P_\mathcal{T}, \pi^E, \gamma, c) + \beta(P_\mathcal{T}, \pi^E, \gamma, c) \cdot \mathbb{E}_{\pi^E}\left[Q_{\mathcal{M} \cup c}^{c, \pi^E}(s, a^E)\right]. \tag{12}$$

where coefficients $\alpha \geq 0$, $\beta > 0$. $\beta$ can not equal to 0, since state $s$ has to be visited with at least some probability. Otherwise, we do not need to explain $\pi^E(s)$. Note that if $Q_{\mathcal{M} \cup c}^{c, \pi^E}(s, a') \leq \mathbb{E}_{\pi^E}\left[Q_{\mathcal{M} \cup c}^{c, \pi^E}(s, a^E)\right]$, $\pi'$ is a strictly better policy than the expert policy for any

$\theta \in (0, 1]$ (larger rewards with equal or less costs). This clearly makes a contradiction. Hence, we focus on $Q_{\mathcal{M}\cup c}^{c,\pi^E}(s, a') > \mathbb{E}_{\pi^E}\left[Q_{\mathcal{M}\cup c}^{c,\pi^E}(s, a^E)\right]$. In this case, we can always obtain a $\theta > 0$, by letting

$$\mathbb{E}_{\mu_0}\left[V_{\mathcal{M}\cup c}^{c,\pi^E}(s_0)\right] + \tau' = \alpha(P_{\mathcal{T}}, \pi^E, \gamma, c) + \beta(P_{\mathcal{T}}, \pi^E, \gamma, c) \cdot \left[(1-\theta)\mathbb{E}_{\pi^E}\left[Q_{\mathcal{M}\cup c}^{c,\pi^E}(s, a^E)\right] + \theta Q_{\mathcal{M}\cup c}^{c,\pi^E}(s, a')\right], \tag{13}$$

where $\tau' \in [0, \tau)$ denotes the leftover budget after applying $\pi'$. By subtracting Eq. (12) from Eq. (13), we have $\forall\, Q_{\mathcal{M}\cup c}^{c,\pi^E}(s, a') > \mathbb{E}_{\pi^E}\left[Q_{\mathcal{M}\cup c}^{c,\pi^E}(s, a^E)\right]$,

$$\theta = \frac{\tau'}{\beta(P_{\mathcal{T}}, \pi^E, \gamma, c)\left[Q_{\mathcal{M}\cup c}^{c,\pi^E}(s, a') - \mathbb{E}_{\pi^E}\left[Q_{\mathcal{M}\cup c}^{c,\pi^E}(s', a^E)\right]\right]}. \tag{14}$$

With this analysis, if $A_{\mathcal{M}}^{r,\pi^E}(s, a') > 0$, which indicates 2) of Lemma 4.1 is not satisfied so 1) must be satisfied, $Q_{\mathcal{M}\cup c}^{c,\pi^E}(s, a') > \mathbb{E}_{\pi^E}\left[Q_{\mathcal{M}\cup c}^{c,\pi^E}(s, a^E)\right] = V_{\mathcal{M}\cup c}^{c,\pi^E}(s)$ suffices to let $\mathbb{E}_{\mu_0, \pi'', P_{\mathcal{T}}}\left[\sum_{t=0}^{\infty} \gamma^t c(s_t, a_t)\right] > \epsilon$ with $\pi''$ only differs from $\pi^E$ at state $s$ where $\pi''(s) = a'$, which is a constraint-violating condition. $\square$

## C.2 Proof of Lemma 4.4

*Proof.* In this proof, we distinguish two cases as in Assumption 4.3.
In the first case, the constraint n (1) is hard, i.e., $\epsilon = 0$.

- (i) By definition of expert policy $\pi^E$, we have $V_{\mathcal{M}\cup c}^{c,\pi^E}(s) = 0$. On one hand, if $c$ is feasible, $V_{\mathcal{M}\cup c}^{c,\pi^E} = \mathbb{E}_{\pi^E}[Q_{\mathcal{M}\cup c}^{c,\pi^E}] = 0$. Also, since $c \in [0, C_{\max}]^{\mathcal{S}\times\mathcal{A}}$, $Q_{\mathcal{M}\cup c}^{c,\pi^E} \geq 0$. As a result, $Q_{\mathcal{M}\cup c}^{c,\pi^E} = 0 = V_{\mathcal{M}\cup c}^{c,\pi^E}$. On the other hand, any $c \in [0, C_{\max}]^{\mathcal{S}\times\mathcal{A}}$ that satisfies $Q_{\mathcal{M}\cup c}^{c,\pi^E} = V_{\mathcal{M}\cup c}^{c,\pi^E} = 0$ makes $\pi^E$ an optimal policy under this condition.
- (ii) By definition of expert policy $\pi^E$, we have $V_{\mathcal{M}\cup c}^{c,\pi^E}(s) = 0$. On one hand, since $A_{\mathcal{M}}^{r,\pi^E}(s, a) > 0$, if $c$ is feasible, $Q_{\mathcal{M}\cup c}^{c,\pi^E}(s, a) > 0$, otherwise $\pi^E$ is not optimal. On the other hand, any cost function $c \in [0, C_{\max}]^{\mathcal{S}\times\mathcal{A}}$ that satisfies $Q_{\mathcal{M}\cup c}^{c,\pi^E}(s, a) > 0 = V_{\mathcal{M}\cup c}^{c,\pi^E}(s)$ ensures action $a$ violates the constraint, and makes $\pi^E$ an optimal policy under this condition.
- (iii) By definition of expert policy $\pi^E$, we have $V_{\mathcal{M}\cup c}^{c,\pi^E}(s) = 0$. On one hand, since $A_{\mathcal{M}}^{r,\pi^E}(s, a) \leq 0$, any $c \in [0, C_{\max}]^{\mathcal{S}\times\mathcal{A}}$ ensures the expert's optimality. However, in terms of the minimal set $\mathcal{C}_{\mathfrak{P}}$ in Definition 4.2, $c(s, a) = 0$ and $Q_{\mathcal{M}\cup c}^{c,\pi^E}(s, a) = 0 = V_{\mathcal{M}\cup c}^{c,\pi^E}(s)$. On the other hand, any $c(s, a) \in [0, C_{\max}]^{\mathcal{S}\times\mathcal{A}}$ that satisfies $Q_{\mathcal{M}\cup c}^{c,\pi^E}(s, a) = 0 = V_{\mathcal{M}\cup c}^{c,\pi^E}(s)$ ensures $\pi^E$ an optimal policy under this condition.

In the second case, the constraint in (1) is soft, i.e., $\epsilon > 0$, and the expert policy is deterministic.

- (i) Since the expert policy $\pi^E$ is deterministic, we have $Q_{\mathcal{M}\cup c}^{c,\pi^E}(s, a) = V_{\mathcal{M}\cup c}^{c,\pi^E}(s)$. On one hand, if $c$ is feasible, $Q_{\mathcal{M}\cup c}^{c,\pi^E}(s, a) = V_{\mathcal{M}\cup c}^{c,\pi^E}(s)$. On the other hand, any $c \in [0, C_{\max}]^{\mathcal{S}\times\mathcal{A}}$ that satisfies $Q_{\mathcal{M}\cup c}^{c,\pi^E}(s, a) = V_{\mathcal{M}\cup c}^{c,\pi^E}(s)$ makes $\pi^E$ an optimal policy under this condition.
- (ii) In this case, since $A_{\mathcal{M}}^{r,\pi^E}(s, a) > 0$, situation 2) of Lemma 4.1 is not satisfied. As a result, 1) of Lemma 4.1 must be satisfied. On one hand, if $c$ is feasible, $Q_{\mathcal{M}\cup c}^{c,\pi^E}(s, a) > Q_{\mathcal{M}\cup c}^{c,\pi^E}(s, a^E) = V_{\mathcal{M}\cup c}^{c,\pi^E}(s)$ suffices to let $\mathbb{E}_{\mu_0, \pi^E, P_{\mathcal{T}}}\left[\sum_{t=0}^{\infty} \gamma^t c(s_t, a_t)\right] > \epsilon$. On the other hand, any cost function $c \in [0, C_{\max}]^{\mathcal{S}\times\mathcal{A}}$ that satisfies $Q_{\mathcal{M}\cup c}^{c,\pi^E}(s, a) > V_{\mathcal{M}\cup c}^{c,\pi^E}(s)$ ensures action $a$ violates the constraint, and makes $\pi^E$ an optimal policy under this condition.
- (iii) On one hand, since $A_{\mathcal{M}}^{r,\pi^E}(s, a) \leq 0$, any relationship between $Q_{\mathcal{M}\cup c}^{c,\pi^E}(s, a)$ and $V_{\mathcal{M}\cup c}^{c,\pi^E}(s)$ ensures the expert's optimality. However, in terms of the minimal set $\mathcal{C}_{\mathfrak{P}}$ in Definition 4.2, $Q_{\mathcal{M}\cup c}^{c,\pi^E}(s, a) \leq V_{\mathcal{M}\cup c}^{c,\pi^E}(s)$. On the other hand, any $c \in [0, C_{\max}]^{\mathcal{S}\times\mathcal{A}}$ that satisfies $Q_{\mathcal{M}\cup c}^{c,\pi^E}(s, a) \leq V_{\mathcal{M}\cup c}^{c,\pi^E}(s)$ ensures $\pi^E$ an optimal policy under this condition.

$\square$

### C.3 PROOF OF LEMMA 4.5

**Lemma C.1.** *Let $\mathfrak{P} = (\mathcal{M}, \pi^E)$ be an ICRL problem. A Q-function satisfies the condition of Lemma 4.4 if and only if there exist $\zeta \in \mathbb{R}_{>0}^{\mathcal{S} \times \mathcal{A}}$ and $V^c \in \mathbb{R}_{\geq 0}^{\mathcal{S}}$ such that:*

$$Q_{\mathcal{M} \cup c}^c = A_{\mathcal{M}}^{r, \pi^E} \zeta + EV^c, \tag{15}$$

*where the expansion operator $E$ satisfies $(Ef)(s, a) = f(s)$.*

Here, the term $\zeta$ ensures 1) (when $A_{\mathcal{M}}^{r, \pi^E} > 0$) the constraint condition in (1) is violated at $(s, a)$ pairs that achieve larger rewards than the expert policy, and 2) (when $A_{\mathcal{M}}^{r, \pi^E} \leq 0$) only necessary cost functions are captured by feasible cost set $\mathcal{C}_{\mathfrak{P}}$.

*Proof.* We prove both the 'if' and 'only if' sides.

To demonstrate the "if" side, we can easily see that all the Q-functions of the form $Q_{\mathcal{M} \cup c}^c(s, a) = A_{\mathcal{M}}^{r, \pi^E}(s, a)\zeta(s, a) + EV^c(s)$ satisfies the conditions of Lemma 4.4 in the following:

1) Let $s \in \mathcal{S}$ and $a \in \mathcal{A}$ such that $\pi^E(a|s) > 0$, then we have $Q_{\mathcal{M} \cup c}^c(s, a) = V^c(s) = V_{\mathcal{M} \cup c}^c(s)$. This is the condition (i) in Lemma 4.4. Note that $V^c(s) = V_{\mathcal{M} \cup c}^c(s)$ holds true for the following two cases since each state $s \in \mathcal{S}$ has an expert policy such that $\pi^E(a|s) > 0$.

2) Let $s \in \mathcal{S}$ and $a \in \mathcal{A}$ such that $\pi^E(a|s) = 0$ and $Q_{\mathcal{M}}^{r, \pi^E}(s, a) > V_{\mathcal{M}}^{r, \pi^E}(s)$, then we have $Q_{\mathcal{M} \cup c}^c(s, a) = A_{\mathcal{M}}^{r, \pi^E}(s, a)\zeta(s, a) + V^c(s) = A_{\mathcal{M}}^{r, \pi^E}(s, a)\zeta(s, a) + V_{\mathcal{M} \cup c}^c(s) > V_{\mathcal{M} \cup c}^c(s)$. This is the case (ii) in Lemma 4.4.

3) Let $s \in \mathcal{S}$ and $a \in \mathcal{A}$ such that $\pi^E(a|s) = 0$ and $Q_{\mathcal{M}}^{r, \pi^E}(s, a) \leq V_{\mathcal{M}}^{r, \pi^E}(s)$, then we have $Q_{\mathcal{M} \cup c}^c(s, a) = A_{\mathcal{M}}^{r, \pi^E}(s, a)\zeta + V^c(s) = A_{\mathcal{M}}^{r, \pi^E}(s, a)\zeta(s, a) + V_{\mathcal{M} \cup c}^c(s) \leq V_{\mathcal{M} \cup c}^c(s)$. This is the case (iii) in Lemma 4.4.

To demonstrate the "only if" side, suppose that $Q_{\mathcal{M} \cup c}^c$ satisfies conditions of Lemma 4.4, we take $V^c(s) = V_{\mathcal{M} \cup c}^c(s)$ since we are proving the existence of $V^c \in \mathbb{R}_{\geq 0}^{\mathcal{S}}$.

1) In the critical region and follows the expert policy, where $Q_{\mathcal{M}}^{r, \pi^E}(s, a) = V_{\mathcal{M}}^{r, \pi^E}(s)$, $0\zeta(s, a) = Q_{\mathcal{M} \cup c}^c - EV_{\mathcal{M} \cup c}^c = 0$. Hence, there definitely exists $\zeta(s, a) > 0$.

2) In the constraint-violating region with more rewards, where $Q_{\mathcal{M}}^{r, \pi^E}(s, a) > V_{\mathcal{M}}^{r, \pi^E}(s)$, $A_{\mathcal{M}}^{r, \pi^E}(s, a)\zeta(s, a) = Q_{\mathcal{M} \cup c}^c - EV_{\mathcal{M} \cup c}^c > 0$. Hence, there definitely exists $\zeta(s, a) > 0$.

3) In the non-critical region with less rewards, where $Q_{\mathcal{M}}^{r, \pi^E}(s, a) \leq V_{\mathcal{M}}^{r, \pi^E}(s)$, $A_{\mathcal{M}}^{r, \pi^E}(s, a)\zeta(s, a) = Q_{\mathcal{M} \cup c}^c - EV_{\mathcal{M} \cup c}^c \leq 0$. Hence, there definitely exists $\zeta(s, a) > 0$. $\square$

**Proof of Lemma 4.5**

*Proof.* Recall that $Q_{\mathcal{M} \cup c}^c = (I_{\mathcal{S} \times \mathcal{A}} - \gamma P_{\mathcal{T}} \pi^E)^{-1} c$ and based on Lemma C.1, we can show that:

$$c = \left(I_{\mathcal{S} \times \mathcal{A}} - \gamma P_{\mathcal{T}} \pi^E\right)\left(A_{\mathcal{M}}^{r, \pi^E} \zeta + EV^c\right)$$

$$= A_{\mathcal{M}}^{r, \pi^E} \zeta + EV^c - \gamma P_{\mathcal{T}} \pi^E A_{\mathcal{M}}^{r, \pi^E} \zeta - \gamma P_{\mathcal{T}} \pi^E EV^c$$

Since $\pi^E A_{\mathcal{M}}^{r, \pi^E} = \mathbf{0}_{\mathcal{S}}$ and $\pi^E E = I_{\mathcal{S}}$,

$$c = A_{\mathcal{M}}^{r, \pi^E} \zeta + (E - \gamma P_{\mathcal{T}})V^c$$

We now bound the infinity norm of $\zeta$ and $V^c$. First, from Eq. (15), we know that $EV^c(s) = Q_{\mathcal{M} \cup c}^c(s, a^E)$. Hence, intuitively $\|V^c(s)\|_\infty \leq \frac{C_{\max}}{1-\gamma}$. Second, from Eq. (2),

$c(s,a) = A_{\mathcal{M}}^{r,\pi^E}(s,a)\zeta(s,a) + (E - \gamma P_{\mathcal{T}})V^c(s)$. 1) When $A_{\mathcal{M}}^{r,\pi^E} > 0$, $\zeta = (c(s,a) - (E - \gamma P_{\mathcal{T}})V^c(s))/A_{\mathcal{M}}^{r,\pi^E}(s,a) \leq C_{\max}/\min_{(s,a)}^+ A_{\mathcal{M}}^{r,\pi^E}(s,a)$. 2) When $A_{\mathcal{M}}^{r,\pi^E} < 0$, $\zeta = (-c(s,a) + (E - \gamma P_{\mathcal{T}})V^c(s))/(-A_{\mathcal{M}}^{r,\pi^E}(s,a))$. Since $(E - \gamma P_{\mathcal{T}})V^c(s) = c(s,a^E) \leq C_{\max}$, $\zeta \leq C_{\max}/\left(-\max_{(s,a)}^+ A_{\mathcal{M}}^{r,\pi^E}(s,a)\right)$. 3) When $A_{\mathcal{M}}^{r,\pi^E} = 0$, we define $\zeta(s,a) = 0$. To combine all the three conditions, $\|\zeta\|_\infty \leq C_{\max}/\min_{(s,a)}^+ |A_{\mathcal{M}}^{r,\pi^E}|$. $\qquad\square$

## C.4   PROOF OF LEMMA 4.6

*Proof.* From Lemma 4.5, $\forall(s,a) \in \mathcal{S} \times \mathcal{A}$, we can express the cost functions belonging to $\mathcal{C}_{\mathfrak{P}}$ and $\mathcal{C}_{\widehat{\mathfrak{P}}}$ as:

$$c(s,a) = A_{\mathcal{M}}^{r,\pi^E}\zeta(s,a) + (E - \gamma P_{\mathcal{T}})V^c(s,a)$$

$$\widehat{c}(s,a) = A_{\widehat{\mathcal{M}}}^{r,\widehat{\pi}^E}\widehat{\zeta}(s,a) + (E - \gamma\widehat{P_{\mathcal{T}}})\widehat{V}^c(s,a)$$

where $(\zeta, \widehat{\zeta}) \in \mathbb{R}_{>0}^{\mathcal{S} \times \mathcal{A}}$ and $V, \widehat{V} \in \mathbb{R}_{\geq 0}^{\mathcal{S}}$. Since we look for the existence of $\widehat{c} \in \mathcal{C}_{\widehat{\mathfrak{P}}}$ satisfying $\|\widehat{c}\|_\infty \leq C_{\max}$, we provide a specific choice of $\widehat{V}$ and $\widehat{\zeta}$: $\widehat{\zeta}(s,a) = \frac{\zeta(s,a)}{1+\chi/C_{\max}}$, $\widehat{V}^c(s,a) = \frac{V^c(s,a)}{1+\chi/C_{\max}}$, where $\chi = \max_{(s,a)\in\mathcal{S}\times\mathcal{A}}\chi(s,a)$ with $\chi(s,a) = \gamma\left|(P_{\mathcal{T}} - \widehat{P_{\mathcal{T}}})V^c\right|(s,a) + \left|A_{\mathcal{M}}^{r,\pi^E} - A_{\widehat{\mathcal{M}}}^{r,\widehat{\pi}^E}\right|\zeta(s,a)$. Next, we prove $\|\widehat{c}\|_\infty \leq C_{\max}$. Let $\widetilde{c} = (1 + \chi/C_{\max})\widehat{c}$, $\forall(s,a) \in \mathcal{S} \times \mathcal{A}$ we have:

$$|\widetilde{c}(s,a)| \leq |c(s,a)| + |\widetilde{c}(s,a) - c(s,a)|$$

$$\leq C_{\max} + \gamma\left|(P_{\mathcal{T}} - \widehat{P_{\mathcal{T}}})V^c\right|(s,a) + \left|A_{\mathcal{M}}^{r,\pi^E} - A_{\widehat{\mathcal{M}}}^{r,\widehat{\pi}^E}\right|\zeta(s,a)$$

$$\leq C_{\max} + |\chi(s,a)|$$

$$\leq C_{\max} + \chi \qquad\qquad (16)$$

As a result, $\|\widehat{c}\|_\infty = \|\widetilde{c}\|_\infty/(1 + \chi/C_{\max}) \leq C_{\max}$. Thus, we have:

$$|c(s,a) - \widehat{c}(s,a)| = |c(s,a) - \frac{\widetilde{c}(s,a)}{1 + \chi/C_{\max}}|$$

$$\leq \frac{1}{1 + \chi/C_{\max}}\left(|c(s,a) - \widetilde{c}(s,a)| + \chi/C_{\max}|c(s,a)|\right)$$

$$\leq \frac{2\chi}{1 + \chi/C_{\max}}. \qquad\qquad (17)$$

$\qquad\square$

## C.5   PROOF OF LEMMA 4.7

**Lemma C.2.** *(Simulation Lemma for action-value function.) Let $\mathcal{M} = (\mathcal{S}, \mathcal{A}, P_{\mathcal{T}}, r, \mu_0, \gamma)$ and $\widehat{\mathcal{M}} = (\mathcal{S}, \mathcal{A}, \widehat{P_{\mathcal{T}}}, r, \mu_0, \gamma)$ be two MDPs. Let $\widehat{\pi} \in \Delta_{\mathcal{S}}^{\mathcal{A}}$ be a policy. The following equality holds element-wise:*

$$Q_{\mathcal{M}}^{r,\widehat{\pi}} - Q_{\widehat{\mathcal{M}}}^{r,\widehat{\pi}} = \gamma(I_{\mathcal{S}\times\mathcal{A}} - \gamma P_{\mathcal{T}}\widehat{\pi})^{-1}(P_{\mathcal{T}} - \widehat{P_{\mathcal{T}}})V_{\widehat{\mathcal{M}}}^{r,\widehat{\pi}} \qquad (18)$$

*Proof.* The proof can be shown as follows:

$$Q_{\mathcal{M}}^{r,\widehat{\pi}} - Q_{\widehat{\mathcal{M}}}^{r,\widehat{\pi}} = (I_{\mathcal{S}\times\mathcal{A}} - \gamma P_{\mathcal{T}}\widehat{\pi})^{-1}r - (I_{\mathcal{S}\times\mathcal{A}} - \gamma P_{\mathcal{T}}\widehat{\pi})^{-1}(I_{\mathcal{S}\times\mathcal{A}} - \gamma P_{\mathcal{T}}\widehat{\pi})Q_{\widehat{\mathcal{M}}}^{r,\widehat{\pi}}$$

$$= (I_{\mathcal{S}\times\mathcal{A}} - \gamma P_{\mathcal{T}}\widehat{\pi})^{-1}(I_{\mathcal{S}\times\mathcal{A}} - \gamma\widehat{P_{\mathcal{T}}}\widehat{\pi})Q_{\widehat{\mathcal{M}}}^{r,\widehat{\pi}} - (I_{\mathcal{S}\times\mathcal{A}} - \gamma P_{\mathcal{T}}\widehat{\pi})^{-1}(I_{\mathcal{S}\times\mathcal{A}} - \gamma P_{\mathcal{T}}\widehat{\pi})Q_{\widehat{\mathcal{M}}}^{r,\widehat{\pi}}$$

$$= \gamma(I_{\mathcal{S}\times\mathcal{A}} - \gamma P_{\mathcal{T}}\widehat{\pi})^{-1}(P_{\mathcal{T}} - \widehat{P_{\mathcal{T}}})\widehat{\pi}Q_{\widehat{\mathcal{M}}}^{r,\widehat{\pi}}$$

$$= \gamma(I_{\mathcal{S}\times\mathcal{A}} - \gamma P_{\mathcal{T}}\widehat{\pi})^{-1}(P_{\mathcal{T}} - \widehat{P_{\mathcal{T}}})V_{\widehat{\mathcal{M}}}^{r,\widehat{\pi}}$$

$\qquad\square$

**Lemma C.3.** *(Simulation Lemma for state-value function.) Let $\mathcal{M} = (\mathcal{S}, \mathcal{A}, P_{\mathcal{T}}, r, \mu_0, \gamma)$ and $\widehat{\mathcal{M}} = (\mathcal{S}, \mathcal{A}, \widehat{P_{\mathcal{T}}}, r, \mu_0, \gamma)$ be two MDPs. Let $\widehat{\pi} \in \Delta_{\mathcal{S}}^{\mathcal{A}}$ be a policy. The following equality holds element-wise:*

$$V_{\mathcal{M}}^{r,\widehat{\pi}} - V_{\widehat{\mathcal{M}}}^{r,\widehat{\pi}} = \gamma(I_{\mathcal{S}} - \gamma\widehat{\pi}P_{\mathcal{T}})^{-1}\widehat{\pi}(\widehat{P_{\mathcal{T}}} - P_{\mathcal{T}})V_{\widehat{\mathcal{M}}}^{r,\widehat{\pi}} \tag{19}$$

*Proof.* The proof can be shown as follows:

$$\begin{aligned}
V_{\mathcal{M}}^{r,\widehat{\pi}} - V_{\widehat{\mathcal{M}}}^{r,\widehat{\pi}} &= (I_{\mathcal{S}} - \gamma\widehat{\pi}P_{\mathcal{T}})^{-1}r - (I_{\mathcal{S}} - \gamma\widehat{\pi}P_{\mathcal{T}})^{-1}(I_{\mathcal{S}} - \gamma\widehat{\pi}P_{\mathcal{T}})V_{\widehat{\mathcal{M}}}^{r,\widehat{\pi}} \\
&= (I_{\mathcal{S}} - \gamma\widehat{\pi}P_{\mathcal{T}})^{-1}(I_{\mathcal{S}} - \gamma\widehat{\pi}\widehat{P_{\mathcal{T}}})V_{\widehat{\mathcal{M}}}^{r,\widehat{\pi}} - (I_{\mathcal{S}} - \gamma\widehat{\pi}P_{\mathcal{T}})^{-1}(I_{\mathcal{S}} - \gamma\widehat{\pi}P_{\mathcal{T}})V_{\widehat{\mathcal{M}}}^{r,\widehat{\pi}} \\
&= \gamma(I_{\mathcal{S}} - \gamma\widehat{\pi}P_{\mathcal{T}})^{-1}\widehat{\pi}(P_{\mathcal{T}} - \widehat{P_{\mathcal{T}}})V_{\widehat{\mathcal{M}}}^{r,\widehat{\pi}} \\
&= \gamma(I_{\mathcal{S}} - \gamma\widehat{\pi}P_{\mathcal{T}})^{-1}\widehat{\pi}(P_{\mathcal{T}} - \widehat{P_{\mathcal{T}}})V_{\widehat{\mathcal{M}}}^{r,\widehat{\pi}}
\end{aligned}$$

$\square$

**Lemma C.4.** *(Policy Mismatch Lemma.) Let $\mathcal{M} = (\mathcal{S}, \mathcal{A}, P_{\mathcal{T}}, r, \mu_0, \gamma)$ be an MDP. Let $\pi, \widehat{\pi} \in \Delta_{\mathcal{S}}^{\mathcal{A}}$ be two policies. The following equality holds element-wise:*

$$V_{\mathcal{M}}^{r,\pi} - V_{\mathcal{M}}^{r,\widehat{\pi}} = \gamma(I_{\mathcal{S}} - \gamma\widehat{\pi}P_{\mathcal{T}})^{-1}(\pi - \widehat{\pi})P_{\mathcal{T}}V_{\mathcal{M}}^{r,\pi}$$

*Proof.* The proof can be shown as follows:

$$\begin{aligned}
V_{\mathcal{M}}^{r,\pi} - V_{\mathcal{M}}^{r,\widehat{\pi}} &= (I_{\mathcal{S}} - \gamma\widehat{\pi}P_{\mathcal{T}})^{-1}(I_{\mathcal{S}} - \gamma\widehat{\pi}P_{\mathcal{T}})V_{\mathcal{M}}^{r,\pi} - (I_{\mathcal{S}} - \gamma\widehat{\pi}P_{\mathcal{T}})^{-1}r \\
&= (I_{\mathcal{S}} - \gamma\widehat{\pi}P_{\mathcal{T}})^{-1}(I_{\mathcal{S}} - \gamma\widehat{\pi}P_{\mathcal{T}})V_{\mathcal{M}}^{r,\pi} - (I_{\mathcal{S}} - \gamma\widehat{\pi}P_{\mathcal{T}})^{-1}(I_{\mathcal{S}} - \gamma\pi P_{\mathcal{T}})V_{\mathcal{M}}^{r,\pi} \\
&= \gamma(I_{\mathcal{S}} - \gamma\widehat{\pi}P_{\mathcal{T}})^{-1}(\pi - \widehat{\pi})P_{\mathcal{T}}V_{\mathcal{M}}^{r,\pi}
\end{aligned}$$

$\square$

**Proof of Lemma 4.7**

*Proof.* By utilizing the triangular inequality of norms, we can obtain:

$$\begin{aligned}
\left|A_{\mathcal{M}}^{r,\pi} - A_{\widehat{\mathcal{M}}}^{r,\widehat{\pi}}\right| &\le \left|A_{\mathcal{M}}^{r,\widehat{\pi}} - A_{\widehat{\mathcal{M}}}^{r,\widehat{\pi}}\right| + \left|A_{\mathcal{M}}^{r,\pi} - A_{\mathcal{M}}^{r,\widehat{\pi}}\right| \\
&\overset{I,II}{\le} \frac{2\gamma}{1-\gamma}\left|(\widehat{P_{\mathcal{T}}} - P_{\mathcal{T}})V_{\widehat{\mathcal{M}}}^{r,\widehat{\pi}}\right| + \frac{\gamma(1+\gamma)}{1-\gamma}|(\pi - \widehat{\pi})P_{\mathcal{T}}V_{\mathcal{M}}^{r,\pi}|,
\end{aligned} \tag{20}$$

where the second inequality is derived by the following two parts.

**Part I.** Let's consider the first part.

$$\begin{aligned}
\left|A_{\mathcal{M}}^{r,\widehat{\pi}} - A_{\widehat{\mathcal{M}}}^{r,\widehat{\pi}}\right| &\overset{(i)}{=} \left|\left(Q_{\mathcal{M}}^{r,\widehat{\pi}} - Q_{\widehat{\mathcal{M}}}^{r,\widehat{\pi}}\right) - E\left(V_{\mathcal{M}}^{r,\widehat{\pi}} - V_{\widehat{\mathcal{M}}}^{r,\widehat{\pi}}\right)\right| \\
&\overset{(ii)}{\le} \left|\left(Q_{\mathcal{M}}^{r,\widehat{\pi}} - Q_{\widehat{\mathcal{M}}}^{r,\widehat{\pi}}\right)\right| + \left|E\left(V_{\mathcal{M}}^{r,\widehat{\pi}} - V_{\widehat{\mathcal{M}}}^{r,\widehat{\pi}}\right)\right| \\
&\overset{(iii)}{=} \gamma\left|(I_{\mathcal{S}\times\mathcal{A}} - \gamma P_{\mathcal{T}}\widehat{\pi})^{-1}(\widehat{P_{\mathcal{T}}} - P_{\mathcal{T}})V_{\widehat{\mathcal{M}}}^{r,\widehat{\pi}}\right| + \gamma\left|(I_{\mathcal{S}} - \gamma\widehat{\pi}P_{\mathcal{T}})^{-1}\widehat{\pi}(\widehat{P_{\mathcal{T}}} - P_{\mathcal{T}})V_{\widehat{\mathcal{M}}}^{r,\widehat{\pi}}\right| \\
&\overset{(iv)}{=} \gamma\left\|(I_{\mathcal{S}\times\mathcal{A}} - \gamma P_{\mathcal{T}}\widehat{\pi})^{-1}\right\|_{\infty}\left|(\widehat{P_{\mathcal{T}}} - P_{\mathcal{T}})V_{\widehat{\mathcal{M}}}^{r,\widehat{\pi}}\right| + \gamma\left\|(I_{\mathcal{S}} - \gamma\widehat{\pi}P_{\mathcal{T}})^{-1}\right\|_{\infty}\|\widehat{\pi}\|_{\infty}\left|(\widehat{P_{\mathcal{T}}} - P_{\mathcal{T}})V_{\widehat{\mathcal{M}}}^{r,\widehat{\pi}}\right| \\
&\overset{(v)}{\le} \frac{2\gamma}{1-\gamma}\left|(\widehat{P_{\mathcal{T}}} - P_{\mathcal{T}})V_{\widehat{\mathcal{M}}}^{r,\widehat{\pi}}\right|
\end{aligned}$$

- (i) exploits the definition of advantage function.

- (ii) applies the triangular inequality.

- (iii) applies the simulation Lemma for action-value function in Lemma C.2 (a variant of (Agarwal et al., 2019, Lemma 2.2)) and the simulation Lemma for state-value function in Lemma C.3.

- (iv) exploits Holder's inequality and the theorem of matrix infinity norm inequalities that $\|AB\|_\infty \leq \|A\|_\infty \|B\|_\infty$.

- (v) exploits the fact that $\|(I_{\mathcal{S} \times \mathcal{A}} - \gamma P_{\mathcal{T}} \widehat{\pi})^{-1}\|_\infty \leq \frac{1}{1-\gamma}$, $\|(I_{\mathcal{S}} - \gamma \widehat{\pi} P_{\mathcal{T}})^{-1}\|_\infty \leq \frac{1}{1-\gamma}$, and $\|\pi\|_\infty \leq 1$.

**Part II.** Let's consider the second part:

$$
\begin{aligned}
\left| A_{\mathcal{M}}^{r,\pi} - A_{\mathcal{M}}^{r,\widehat{\pi}} \right| &= \left| \left( Q_{\mathcal{M}}^{r,\pi} - Q_{\mathcal{M}}^{r,\widehat{\pi}} \right) - E\left( V_{\mathcal{M}}^{r,\pi} - V_{\mathcal{M}}^{r,\widehat{\pi}} \right) \right| \\
&\overset{(i)}{=} \left| \gamma\left( P_{\mathcal{T}} V_{\mathcal{M}}^{r,\pi} - P_{\mathcal{T}} V_{\mathcal{M}}^{r,\widehat{\pi}} \right) - E\left( V_{\mathcal{M}}^{r,\pi} - V_{\mathcal{M}}^{r,\widehat{\pi}} \right) \right| \\
&\overset{(ii)}{=} \gamma \left| P_{\mathcal{T}} \left( V_{\mathcal{M}}^{r,\pi} - V_{\mathcal{M}}^{r,\widehat{\pi}} \right) \right| + \left| E\left( V_{\mathcal{M}}^{r,\pi} - V_{\mathcal{M}}^{r,\widehat{\pi}} \right) \right| \\
&\overset{(iii)}{\leq} (1+\gamma) \left| E\left( V_{\mathcal{M}}^{r,\pi} - V_{\mathcal{M}}^{r,\widehat{\pi}} \right) \right| \\
&\overset{(iv)}{\leq} \gamma(1+\gamma) \left| (I_{\mathcal{S}} - \gamma \widehat{\pi} P_{\mathcal{T}})^{-1} (\pi - \widehat{\pi}) P_{\mathcal{T}} V_{\mathcal{M}}^{r,\pi} \right| \\
&\leq \gamma(1+\gamma) \left\| (I_{\mathcal{S}} - \gamma \widehat{\pi} P_{\mathcal{T}})^{-1} \right\|_\infty \left| (\pi - \widehat{\pi}) P_{\mathcal{T}} V_{\mathcal{M}}^{r,\pi} \right| \\
&\overset{(v)}{\leq} \frac{\gamma(1+\gamma)}{1-\gamma} \left| (\pi - \widehat{\pi}) P_{\mathcal{T}} V_{\mathcal{M}}^{r,\pi} \right|
\end{aligned}
$$

- (i) applies the Bellman equation $Q = r + \gamma P_{\mathcal{T}} V$.

- (ii) applies the triangular inequality.

- (iii) holds since $\|P_{\mathcal{T}}\|_\infty \leq 1$.

- (iv) applies the policy mismatch Lemma for state-value function in Lemma C.4.

- (v) exploits the fact that $\|(I_{\mathcal{S}} - \gamma \widehat{\pi} P_{\mathcal{T}})^{-1}\|_\infty \leq \frac{1}{1-\gamma}$

$\square$

## C.6 Proof of Lemma 4.8

*Proof.* We can show that:

$$
\begin{aligned}
\left| Q_{\mathcal{M} \cup c}^{c,\pi^*} - Q_{\mathcal{M} \cup c}^{c,\widehat{\pi}^*} \right| &\overset{(a)}{=} \left| (I_{\mathcal{S} \times \mathcal{A}} - \gamma P_{\mathcal{T}} \pi)^{-1} c - (I_{\mathcal{S} \times \mathcal{A}} - \gamma P_{\mathcal{T}} \pi)^{-1} \widehat{c} \right| \\
&= \left| (I_{\mathcal{S} \times \mathcal{A}} - \gamma P_{\mathcal{T}} \pi)^{-1} |c - \widehat{c}| \right|
\end{aligned}
\tag{21}
$$

- (a) results from the matrix representation of Bellman equation, i.e., $Q_{\mathcal{M} \cup c}^{c,\pi} = (I_{\mathcal{S} \times \mathcal{A}} - \gamma P_{\mathcal{T}} \pi)^{-1} c$.

By definition of infinity norm, we have

$$
|Q_{\mathcal{M} \cup c}^{c,\pi} - Q_{\mathcal{M} \cup \widehat{c}}^{c,\widehat{\pi}}| \leq \|Q_{\mathcal{M} \cup c}^{c,\pi} - Q_{\mathcal{M} \cup \widehat{c}}^{c,\pi}\|_\infty.
\tag{22}
$$

Further, we derive the error upper bound of the action-value function by that of cost.

$$
\begin{aligned}
\|Q_{\mathcal{M} \cup c}^{c,\pi} - Q_{\mathcal{M} \cup \widehat{c}}^{c,\pi}\|_\infty &\overset{(b)}{=} \left\| (I_{\mathcal{S} \times \mathcal{A}} - \gamma P_{\mathcal{T}} \pi)^{-1} |c - \widehat{c}| \right\|_\infty \\
&\overset{(c)}{=} \left\| (I_{\mathcal{S} \times \mathcal{A}} - \gamma P_{\mathcal{T}} \pi)^{-1} \right\|_\infty \|c - \widehat{c}\|_\infty \\
&\overset{(d)}{\leq} \frac{1}{1-\gamma} \|c - \widehat{c}\|_\infty
\end{aligned}
$$

- (b) uses Eq. (21)

- (c) exploits the theorem of matrix infinity norm inequalities that $\|AB\|_\infty \le \|A\|_\infty \|B\|_\infty$

- (d) results from $\|(I_{\mathcal{S}\times\mathcal{A}} - \gamma\pi P_{\mathcal{T}})^{-1}\|_\infty \le \frac{1}{1-\gamma}$.

□

### C.7   PROOF OF LEMMA 5.1

**Lemma C.5.** *(Good Event). Let $\delta \in (0,1)$, define the good event $\mathcal{E}_k$ as the event at iteration $k$ such that the following inequalities hold simultaneously for all $(s,a) \in \mathcal{S} \times \mathcal{A}$ and $k \ge 1$:*

$$\left|(\widehat{P_{\mathcal{T}}}_k - P_{\mathcal{T}})V_{\widehat{\mathcal{M}}_k}^{r,\widehat{\pi}_k^E}\right|(s,a) \le \frac{R_{\max}}{1-\gamma}\sqrt{\frac{\ell_k(s,a)}{2N_k^+(s,a)}},$$

$$\left|(P_{\mathcal{T}} - \widehat{P_{\mathcal{T}}}_k)V_{\mathcal{M}}^{r,\pi^E}\right|(s,a) \le \frac{R_{\max}}{1-\gamma}\sqrt{\frac{\ell_k(s,a)}{2N_k^+(s,a)}},$$

$$\left|(\pi - \widehat{\pi}_k^E)P_{\mathcal{T}}V_{\mathcal{M}}^{r,\pi^E}\right|(s,a) \le \frac{R_{\max}}{1-\gamma}\sqrt{\frac{\ell_k(s,a)}{2N_k^+(s,a)}},$$

$$\left|(\widehat{\pi}_k^E - \pi^E)\widehat{P_{\mathcal{T}}}_k V_{\widehat{\mathcal{M}}_k}^{r,\widehat{\pi}_k^E}\right|(s,a) \le \frac{R_{\max}}{1-\gamma}\sqrt{\frac{\ell_k(s,a)}{2N_k^+(s,a)}},$$

$$\left|(P_{\mathcal{T}} - \widehat{P_{\mathcal{T}}}_k)V^c\right|(s,a) \le \frac{C_{\max}}{1-\gamma}\sqrt{\frac{\ell_k(s,a)}{2N_k^+(s,a)}},$$

$$\left|(P_{\mathcal{T}} - \widehat{P_{\mathcal{T}}}_k)\widehat{V}_k^c\right|(s,a) \le \frac{C_{\max}}{1-\gamma}\sqrt{\frac{\ell_k(s,a)}{2N_k^+(s,a)}},$$

*where $V_{\widehat{\mathcal{M}}_k}^{r,\widehat{\pi}^E}$, $V_{\mathcal{M}}^{r,\pi^E}$, $V^c$ and $\widehat{V}_k^c$ are defined in Lemma 4.6 and Lemma 4.7. $\ell_k(s,a) = \log(36SA(N_k^+(s,a))^2/\delta)$. Then, $\Pr(\mathcal{E}_k) \ge 1 - \delta$.*

*Proof.* We show that each statement does not hold with probability less than $\delta/6$. Let us denote $\beta_{N_k^+(s,a)}^3(s,a) = \frac{C_{\max}}{1-\gamma}\sqrt{\frac{\ell_k(s,a)}{2N_k^+(s,a)}}$ and $\beta_m^3(s,a) = \frac{C_{\max}}{1-\gamma}\sqrt{\frac{\ell_k(s,a)}{2m}}$. Consider the second to last inequality. The probability that it does not hold is:

$$\Pr\left[\exists k \ge 1, \exists(s,a) \in \mathcal{S} \times \mathcal{A} : \left|(P_{\mathcal{T}} - \widehat{P_{\mathcal{T}}}_k)V^c\right|(s,a) > \beta_{N_k^+(s,a)}^3(s,a)\right]$$

$$\overset{(a)}{\le} \sum_{(s,a)} \Pr\left[\exists k \ge 1 : \left|(P_{\mathcal{T}} - \widehat{P_{\mathcal{T}}}_k)V^c\right|(s,a) > \beta_{N_k^+(s,a)}^3(s,a)\right]$$

$$\overset{(b)}{=} \sum_{(s,a)} \Pr\left[\exists m \ge 0 : \left|(P_{\mathcal{T}} - \widehat{P_{\mathcal{T}}}_k)V^c\right|(s,a) > \beta_m^3(s,a)\right]$$

$$\overset{(c)}{\le} \sum_m \sum_{(s,a)} \Pr\left[\left|(P_{\mathcal{T}} - \widehat{P_{\mathcal{T}}}_k)V^c\right|(s,a) > \beta_m^3(s,a)\right]$$

$$\overset{(d)}{\le} \sum_m \sum_{(s,a)} 2\exp\left(\frac{-2(\beta_m^3(s,a))^2 m^2}{m\left(\frac{C_{\max}}{1-\gamma}\right)^2}\right)$$

$$= \sum_m \sum_{(s,a)} 2\exp\left(-\ell_k(s,a)\right)$$

$$= \sum_m \sum_{(s,a)} \frac{2\delta}{36 SA(m^+)^2}$$

$$= \frac{\delta}{18}(1 + \frac{\pi^2}{6}) \leq \frac{\delta}{6} \tag{23}$$

- (a) and (c) use union bound inequalities over $(s, a)$ and $m$.

- (b) assumes that we visit a state-action pair $(s, a)$ for $m$ times, and only focus on these $m$ times that the transition model is updated.

- (d) applies the Hoeffding's inequality and $\|V^c\|_\infty \leq C_{\max}/(1 - \gamma)$ in Lemma 4.6. The factor $m^2$ in the numerator results from dividing by $1/m$ to average over samples, and the factor $m$ in the denominator results from the sum over $m$ in the denominator of Hoeffding's bound.

Similarly, we have $\beta^{1,2}_{N_k^+(s,a)}(s,a) = \frac{R_{\max}}{1-\gamma}\sqrt{\frac{\ell_k(s,a)}{2N_k^+(s,a)}}$ and $\beta^{1,2}_m(s,a) = \frac{R_{\max}}{1-\gamma}\sqrt{\frac{\ell_k(s,a)}{2m}}$ for Lemma's first and second, third and fourth inequalities, respectively. Lemma's last inequality employs $\beta^3_{N_k^+(s,a)}(s,a)$ and $\beta^3_m(s,a)$ again. A union bound over the six probabilities results in $\Pr(\bar{\mathcal{E}}_k) \leq (\delta/6 + \delta/6 + \delta/6 + \delta/6 + \delta/6 + \delta/6) = \delta$. Thus, $\Pr(\mathcal{E}_k) = 1 - \Pr(\bar{\mathcal{E}}_k) \geq 1 - \delta$. □

**Proof of Lemma 5.1**

*Proof.*

$$\chi(s,a) \overset{(a)}{\leq} \gamma \left| (P_{\mathcal{T}} - \widehat{P_{\mathcal{T}}})V^c \right| + \left| A_{\mathcal{M}}^{r,\pi^E} - A_{\widehat{\mathcal{M}}}^{r,\widehat{\pi}^E} \right| \zeta$$

$$\overset{(b)}{\leq} \frac{\gamma \left( R_{\max}(3+\gamma)\zeta(s,a) + C_{\max}(1-\gamma) \right)}{(1-\gamma)^2} \sqrt{\frac{\ell_k(s,a)}{2N_k^+(s,a)}}$$

$$\leq \frac{\gamma \left( R_{\max}(3+\gamma)\|\zeta\|_\infty + C_{\max}(1-\gamma) \right)}{(1-\gamma)^2} \sqrt{\frac{\ell_k(s,a)}{2N_k^+(s,a)}}$$

$$\leq \frac{\gamma C_{\max} \left( R_{\max}(3+\gamma)/\min_{(s,a)}^+ |A_{\mathcal{M}}^{r,\pi^E}| + (1-\gamma) \right)}{(1-\gamma)^2} \sqrt{\frac{\ell_k(s,a)}{2N_k^+(s,a)}} \tag{24}$$

$$= \sigma \sqrt{\frac{\ell_k(s,a)}{2N_k^+(s,a)}} \tag{25}$$

where, for concision, we denote $\sigma = \frac{\gamma C_{\max}\left( R_{\max}(3+\gamma)/\min_{(s,a)}^+ |A_{\mathcal{M}}^{r,\pi^E}| + (1-\gamma) \right)}{(1-\gamma)^2}$.

- (a) uses Lemma 4.6 and the triangular inequality.

- (b) uses Lemma 4.7 and Lemma C.5.

From Lemma 4.6, since $\frac{2\chi}{1+\chi/C_{\max}}$ increases monotonically with $\chi$, we have

$$|c(s,a) - \widehat{c}_k(s,a)| \leq \frac{2\chi}{1 + \chi/C_{\max}} = \max_{(s,a)\in\mathcal{S}\times\mathcal{A}} \frac{2\sigma\sqrt{\frac{\ell_k(s,a)}{2N_k^+(s,a)}}}{1 + \sigma/C_{\max}\sqrt{\frac{\ell_k(s,a)}{2N_k^+(s,a)}}}. \tag{26}$$

Also, note that

$$|c(s,a) - \widehat{c}_k(s,a)| \leq \max\{c(s,a), \widehat{c}_k(s,a)\} \leq C_{\max} \tag{27}$$

Thus, the following formula holds true,

$$|c(s,a) - \widehat{c}_k(s,a)| \leq \mathcal{C}_k(s,a), \forall\, (s,a) \in \mathcal{S} \times \mathcal{A}, \tag{28}$$

$$\mathcal{C}_k(s,a) = \min\left\{ \max_{(s,a)\in\mathcal{S}\times\mathcal{A}} \frac{2\sigma\sqrt{\frac{\ell_k(s,a)}{2N_k^+(s,a)}}}{1 + \sigma/C_{\max}\sqrt{\frac{\ell_k(s,a)}{2N_k^+(s,a)}}}, C_{\max} \right\}, \tag{29}$$

Taking the supremum of Eq. (28) over all $(s,a)$ pairs, we obtain

$$\|c(s,a) - \widehat{c}_k(s,a)\|_\infty \leq \max_{(s,a)\in\mathcal{S}\times\mathcal{A}} \mathcal{C}_k(s,a). \tag{30}$$

Note that, since

$$\max_{(s,a)\in\mathcal{S}\times\mathcal{A}} \min\left\{ \max_{(s,a)\in\mathcal{S}\times\mathcal{A}} \frac{2\sigma\sqrt{\frac{\ell_k(s,a)}{2N_k^+(s,a)}}}{1 + \sigma/C_{\max}\sqrt{\frac{\ell_k(s,a)}{2N_k^+(s,a)}}}, C_{\max} \right\} = \max_{(s,a)\in\mathcal{S}\times\mathcal{A}} \min\left\{ \frac{2\sigma\sqrt{\frac{\ell_k(s,a)}{2N_k^+(s,a)}}}{1 + \sigma/C_{\max}\sqrt{\frac{\ell_k(s,a)}{2N_k^+(s,a)}}}, C_{\max} \right\}, \tag{31}$$

we can further simplify $\mathcal{C}_k$ as

$$\mathcal{C}_k(s,a) = \min\left\{ \frac{2\sigma\sqrt{\frac{\ell_k(s,a)}{2N_k^+(s,a)}}}{1 + \sigma/C_{\max}\sqrt{\frac{\ell_k(s,a)}{2N_k^+(s,a)}}}, C_{\max} \right\}. \tag{32}$$

$\square$

## C.8 Uniform Sampling Strategy for ICRL with a Generative Model

**Corollary C.6.** *Let $\mathcal{C}_{\mathfrak{P}}$ be the exact feasible set and $\mathcal{C}_{\widehat{\mathfrak{P}}_k}$ be the feasible set recovered after $k$ iterations. The conditions of Definition 4.9 are satisfied, if either of the following statements are satisfied:*

(1) $\quad \dfrac{1}{1-\gamma} \max\limits_{(s,a)\in\mathcal{S}\times\mathcal{A}} \mathcal{C}_k(s,a) \leq \varepsilon;$

(2) $\quad \max\limits_{\pi\in\Pi^\dagger} \max\limits_{\mu_0\in\Delta^\mathcal{S}} \left| \mu_0^T (I_{\mathcal{S}\times\mathcal{A}} - \gamma P_\mathcal{T}\pi)^{-1} \mathcal{C}_k \right| \leq \varepsilon,\ \Pi^\dagger = \left( \bigcap\limits_{c\in\mathcal{C}_\mathfrak{P}} \Pi^*_{\mathcal{M}\cup c} \right) \cup \left( \bigcap\limits_{\widehat{c}\in\mathcal{C}_{\widehat{\mathfrak{P}}_k}} \Pi^*_{\widehat{\mathcal{M}}_k\cup\widehat{c}_k} \right).$

*Proof.* For statement (1),

$$\inf_{\widehat{c}_k\in\mathcal{C}_{\widehat{\mathfrak{P}}_k}} \sup_{\pi^*\in\Pi^*_{\mathcal{M}\cup c}} \left| Q^{c,\pi^*}_{\mathcal{M}\cup c}(s,a) - Q^{c,\pi^*}_{\mathcal{M}\cup\widehat{c}_k}(s,a) \right| \leq \inf_{\widehat{c}_k\in\mathcal{C}_{\widehat{\mathfrak{P}}_k}} \sup_{\pi^*\in\Pi^*_{\mathcal{M}\cup c}} \|Q^{c,\pi^*}_{\mathcal{M}\cup c}(s,a) - Q^{c,\pi^*}_{\mathcal{M}\cup\widehat{c}_k}(s,a)\|_\infty$$

$$\overset{(a)}{\leq} \inf_{\widehat{c}_k\in\mathcal{C}_{\widehat{\mathfrak{P}}_k}} \frac{1}{1-\gamma} \|c(s,a) - \widehat{c}_k(s,a)\|_\infty$$

$$\overset{(b)}{=} \frac{1}{1-\gamma} \max_{(s,a)\in\mathcal{S}\times\mathcal{A}} \mathcal{C}_k(s,a) \leq \varepsilon,$$

$$\inf_{c\in\mathcal{C}_\mathfrak{P}} \sup_{\widehat{\pi}^*_k\in\Pi^*_{\widehat{\mathcal{M}}_k\cup\widehat{c}_k}} \left| Q^{c,\widehat{\pi}^*_k}_{\mathcal{M}\cup c}(s,a) - Q^{c,\widehat{\pi}^*_k}_{\mathcal{M}\cup\widehat{c}_k}(s,a) \right| \leq \inf_{c\in\mathcal{C}_\mathfrak{P}} \sup_{\widehat{\pi}^*_k\in\Pi^*_{\widehat{\mathcal{M}}_k\cup\widehat{c}_k}} \|Q^{c,\widehat{\pi}^*_k}_{\mathcal{M}\cup c}(s,a) - Q^{c,\widehat{\pi}^*_k}_{\mathcal{M}\cup\widehat{c}_k}(s,a)\|_\infty$$

$$\overset{(c)}{\leq} \inf_{c\in\mathcal{C}_\mathfrak{P}} \frac{1}{1-\gamma} \|c(s,a) - \widehat{c}_k(s,a)\|_\infty$$

$$\overset{(d)}{=} \frac{1}{1-\gamma} \max_{(s,a)\in\mathcal{S}\times\mathcal{A}} \mathcal{C}_k(s,a) \leq \varepsilon,$$

where step (a) and (c) use Lemma 4.8, step (b) and (d) use Lemma 5.1.

For statement (2),

$$\inf_{\widehat{c}_k \in \mathcal{C}_{\widehat{\mathfrak{P}}_k}} \sup_{\pi^* \in \Pi^*_{\mathcal{M} \cup c}} \left| Q^{c,\pi^*}_{\mathcal{M} \cup c}(s,a) - Q^{c,\pi^*}_{\mathcal{M} \cup \widehat{c}_k}(s,a) \right| \overset{(e)}{\leq} \inf_{\widehat{c}_k \in \mathcal{C}_{\widehat{\mathfrak{P}}_k}} \max_{\pi \in \Pi^\dagger} \left| (I_{\mathcal{S} \times \mathcal{A}} - \gamma P_{\mathcal{T}} \pi)^{-1} |c - \widehat{c}| \right|$$

$$\overset{(f)}{\leq} \max_{\pi \in \Pi^\dagger} \left| (I_{\mathcal{S} \times \mathcal{A}} - \gamma P_{\mathcal{T}} \pi)^{-1} \mathcal{C}_k \right|$$

$$\leq \max_{\pi \in \Pi^\dagger} \max_{\mu_0 \in \Delta^{\mathcal{S}}} \left| \mu_0^T (I_{\mathcal{S} \times \mathcal{A}} - \gamma P_{\mathcal{T}} \pi)^{-1} \mathcal{C}_k \right| \leq \varepsilon,$$

$$\inf_{c \in \mathcal{C}_{\mathfrak{P}}} \sup_{\widehat{\pi}^*_k \in \Pi^*_{\widehat{\mathcal{M}}_k \cup \widehat{c}_k}} \left| Q^{c,\widehat{\pi}^*_k}_{\mathcal{M} \cup c}(s,a) - Q^{c,\widehat{\pi}^*_k}_{\mathcal{M} \cup \widehat{c}_k}(s,a) \right| \overset{(g)}{\leq} \inf_{c \in \mathcal{C}_{\mathfrak{P}}} \max_{\pi \in \Pi^\dagger} \left| (I_{\mathcal{S} \times \mathcal{A}} - \gamma P_{\mathcal{T}} \pi)^{-1} |c - \widehat{c}| \right|$$

$$\overset{(h)}{\leq} \max_{\pi \in \Pi^\dagger} \left| (I_{\mathcal{S} \times \mathcal{A}} - \gamma P_{\mathcal{T}} \pi)^{-1} \mathcal{C}_k \right|$$

$$\leq \max_{\pi \in \Pi^\dagger} \max_{\mu_0 \in \Delta^{\mathcal{S}}} \left| \mu_0^T (I_{\mathcal{S} \times \mathcal{A}} - \gamma P_{\mathcal{T}} \pi)^{-1} \mathcal{C}_k \right| \leq \varepsilon,$$

where step (e) and (g) use Eq. (21), step (f) and (h) use Lemma 5.1. $\square$

### Uniform Sampling Strategy for ICRL with a Generative Model

In this part, we additionally consider the problem setting where the agent does not employ any exploration strategy to acquire desired information, but utilizes uniform sampling strategy to query a generative model. The problem setting is based on the following assumption, which is stronger than the assumption in the main paper.

**Assumption C.7.** The following statements hold:
(i). The agent have access to the *generative model* of $\mathcal{M}$;
(ii). The agent can query the expert's policy $\pi^E$ in *any* state $s \in \mathcal{S}$.

More specifically, the agent can always query a generative model about a state-action pair $(s,a)$ to receive a next state $s' \sim P(\cdot|s,a)$ and about a state $s$ to receive an expert action $a_E \sim \pi^E(\cdot|s)$. We first present Alg. 2 for uniform sampling strategy with the generative model and study the sample complexity of this algorithm in Theorem C.9.

---

**Algorithm 2** Uniform Sampling Strategy for ICRL

**Input:** significance $\delta \in (0,1)$, target accuracy $\varepsilon$, maximum number of samples per iteration $n_{\max}$
Initialize $k \leftarrow 0$, $\varepsilon_0 = \frac{1}{1-\gamma}$
**while** $\varepsilon_k > \varepsilon$ **do**
    Collect $\lceil \frac{n_{\max}}{SA} \rceil$ samples from each $(s,a) \in \mathcal{S} \times \mathcal{A}$
    Update accuracy $\varepsilon_{k+1} = \frac{1}{1-\gamma} \max_{(s,a) \in \mathcal{S} \times \mathcal{A}} \mathcal{C}_{k+1}(s,a)$
    Update $\widehat{\pi}^E_{k+1}(a|s)$ and $\widehat{P}_{\mathcal{T}k+1}(s'|s,a)$ in (5)
    $k \leftarrow k + 1$
**end while**

---

**Lemma C.8.** (*Metelli et al., 2021, Lemma B.8*). *Let* $a, b \geq 0$ *such that* $2a\sqrt{b} > e$. *Then, the inequality* $x \geq a\log(bx^2)$ *is satisfied for all* $x \geq -2aW_{-1}\left(-\frac{1}{2a\sqrt{b}}\right)$, *where* $W_{-1}$ *is the secondary component of the Lambert W function. Moreover,* $-2aW_{-1}\left(-\frac{1}{2a\sqrt{b}}\right) \leq 4a\log(2a\sqrt{b})$.

**Theorem C.9.** (*Sample Complexity of Uniform Sampling Strategy*). *If Algorithm 2 stops at iteration* $K$ *with accuracy* $\varepsilon_K$, *then with probability at least* $1 - \delta$, *it fulfills Definition 4.9 with a number of samples upper bounded by,*

$$n \leq \widetilde{\mathcal{O}}\left(\frac{\sigma^2 SA}{(1-\gamma)^2 \varepsilon_K^2}\right), \tag{33}$$

*where* $\sigma = \frac{\gamma C_{\max}\left(R_{\max}(3+\gamma)/\min^+_{(s,a)} |A^{r,\pi^E}_{\mathcal{M}}| + (1-\gamma)\right)}{(1-\gamma)^2}$ *and* $\widetilde{\mathcal{O}}$ *notation suppresses logarithmic terms.*

*Proof.* We start from Corollary C.6. We further bound:

$$\frac{1}{1-\gamma} \max_{(s,a) \in \mathcal{S} \times \mathcal{A}} \mathcal{C}_k(s,a) = \frac{1}{1-\gamma} \max_{(s,a) \in \mathcal{S} \times \mathcal{A}} \sigma \sqrt{\frac{\ell_k(s,a)}{2N_k^+(s,a)}}$$

After $K$ iterations, based on uniform sampling strategy, we know that $N_K \geq 1$ for any $(s,a) \in \mathcal{S} \times \mathcal{A}$. To terminate at iteration $K$, it suffices to enforce every $(s,a) \in \mathcal{S} \times \mathcal{A}$:

$$\frac{\gamma C_{\max} \left( R_{\max}(3+\gamma)/\min_{(s,a)}^+ |A_{\mathcal{M}}^{r,\pi^E}| + (1-\gamma) \right)}{(1-\gamma)^3} \sqrt{\frac{\ell_k(s,a)}{2N_k^+(s,a)}} = \varepsilon_K$$

$$\implies N_K(s,a) = \frac{\gamma^2 C_{\max}^2 \left( R_{\max}(3+\gamma)/\min_{(s,a)}^+ |A_{\mathcal{M}}^{r,\pi^E}| + (1-\gamma) \right)^2 \ell_k(s,a)}{2(1-\gamma)^6 \varepsilon_K^2}$$

From Lemma C.8, we derive

$$N_K(s,a)$$

$$= -\frac{\gamma^2 \left( R_{\max}(3+\gamma)\|\zeta\|_\infty + C_{\max}(1-\gamma) \right)^2}{(1-\gamma)^6 \varepsilon_K^2} W_{-1} \left( -\frac{2(1-\gamma)^6 \varepsilon_K^2}{\gamma^2 \left( R_{\max}(3+\gamma)\|\zeta\|_\infty + C_{\max}(1-\gamma) \right)^2} \sqrt{\frac{\delta}{36SA}} \right)$$

$$\leq \frac{2\gamma^2 \left( R_{\max}(3+\gamma)\|\zeta\|_\infty + C_{\max}(1-\gamma) \right)^2}{(1-\gamma)^6 \varepsilon_K^2} \log \left( \frac{\gamma^2 \left( R_{\max}(3+\gamma)\|\zeta\|_\infty + C_{\max}(1-\gamma) \right)^2}{(1-\gamma)^6 \varepsilon_K^2} \sqrt{\frac{36SA}{\delta}} \right)$$

$$= \widetilde{\mathcal{O}} \left( \frac{\gamma^2 \left( R_{\max}(3+\gamma)\|\zeta\|_\infty + C_{\max}(1-\gamma) \right)^2}{(1-\gamma)^6 \varepsilon_K^2} \right)$$

$$= \widetilde{\mathcal{O}} \left( \frac{\gamma^2 C_{\max}^2 \left( R_{\max}(3+\gamma)/\min_{(s,a)}^+ |A_{\mathcal{M}}^{r,\pi^E}| + (1-\gamma) \right)^2}{(1-\gamma)^6 \varepsilon_K^2} \right) \tag{34}$$

By summing $n = \sum_{(s,a) \in \mathcal{S} \times \mathcal{A}} N_K(s,a)$, we obtain the upper bound.

$$n \leq \widetilde{\mathcal{O}} \left( \frac{\gamma^2 C_{\max}^2 \left( R_{\max}(3+\gamma)/\min_{(s,a)}^+ |A_{\mathcal{M}}^{r,\pi^E}| + (1-\gamma) \right)^2 SA}{(1-\gamma)^6 \varepsilon_K^2} \right) \tag{35}$$

Since $\sigma = \frac{\gamma C_{\max} \left( R_{\max}(3+\gamma)/\min_{(s,a)}^+ |A_{\mathcal{M}}^{r,\pi^E}| + (1-\gamma) \right)}{(1-\gamma)^2}$, we have

$$n \leq \widetilde{\mathcal{O}} \left( \frac{\sigma^2 SA}{(1-\gamma)^2 \varepsilon_K^2} \right). \tag{36}$$

Regarding the sample complexity in the RL phase, since the reward function is known, by Corollary 2.7 in Section 2.3.1 from book 'Reinforcement Learning: Theory and Algorithms' (Agarwal et al., 2019), the sample complexity of obtaining a $\varepsilon$-optimal policy is $O(SA/(1-\gamma)^3 \varepsilon^2)$, which is dominated by the sample complexity in Theorem 5.5. Note that $\sigma$ also contains $1/(1-\gamma)$. As a result, Eq. (36) still holds true, after taking the sample complexity of this RL phase into account. $\square$

## C.9 PROOF OF LEMMA 5.2

*Proof.*

$$\|e_k(s,a;\pi^*)\|_\infty \overset{(a)}{\leq} \left\| (I_{\mathcal{S} \times \mathcal{A}} - \gamma P_{\mathcal{T}} \pi^*)^{-1} |c - \widehat{c}_k| \right\|_\infty \overset{(b)}{\leq} \left\| \mu_0^T (I_{\mathcal{S} \times \mathcal{A}} - \gamma P_{\mathcal{T}} \pi)^{-1} \mathcal{C}_k \right\|_\infty. \tag{37}$$

- (a) follows Lemma 4.8 (treat $\pi = \pi^*$ and $\widehat{c} = \widehat{c}_k$).
- (b) follows Lemma 5.1.

$\square$

## C.10 PROOF OF LEMMA 5.4

*Proof.* This results generalizes (Kaufmann et al., 2021, Lemma 7) to our setting. We define event $\mathcal{G}^{\mathrm{cnt}}$ as:

$$\mathcal{G}^{\mathrm{cnt}} = \left\{ \forall k \in \mathbb{N}^\star, \forall (s,a) \in \mathcal{S} \times \mathcal{A} : N_k(s,a) \geq \frac{1}{2}\bar{N}_k(s,a) - \log\left(\frac{2SA}{\delta}\right) \right\}. \quad (38)$$

We calculate the probability of the complement of event $\mathcal{G}^{\mathrm{cnt}}$.

$$\mathbb{P}\left(\left(\mathcal{G}^{\mathrm{cnt}}\right)^c\right)$$

$$\stackrel{(a)}{\leq} \sum_{(s,a)\in\mathcal{S}\times\mathcal{A}} \mathbb{P}\left(\exists k \in \mathbb{N} : N_k(s,a) \leq \frac{1}{2}\bar{N}_k(s,a) - \log\left(\frac{2SA}{\delta}\right)\right)$$

$$\stackrel{(b)}{\leq} \sum_{(s,a)\in\mathcal{S}\times\mathcal{A}} \mathbb{P}\left(\exists k \in \mathbb{N} : \sum_{h=1}^{n_{\max}}\sum_{i=1}^{k} \mathbb{1}\left((s_i^h, a_i^h) = (s,a)\right) \leq \frac{1}{2}\sum_{s_0}\sum_{h=1}^{n_{\max}}\sum_{i=1}^{k} \mu_0(s_0)\eta_i^h(s,a|s_0) - \log\left(\frac{2SA}{\delta}\right)\right)$$

$$\stackrel{(c)}{\leq} \sum_{(s,a)\in\mathcal{S}\times\mathcal{A}} \frac{\delta}{2SA} = \frac{\delta}{2}, \quad (39)$$

- (a) results from a union bound over $(s,a)$.

- (b) results from Definition 5.3.

- (c) results from (Kaufmann et al., 2021, Lemma 9).

As a result, we have with probability at least $1 - \delta/2$:

$$N_k(s,a) \geq \frac{1}{2}\bar{N}_k(s,a) - \beta_{\mathrm{cnt}}(\delta), \quad (40)$$

where $\beta_{\mathrm{cnt}}(\delta) = \log(2SA/\delta)$.

The following proof adapts from (Lindner et al., 2022, Lemma B.18). Distinguish two cases. First, let $\beta_{\mathrm{cnt}}(\delta) \leq \frac{1}{4}\bar{N}_k(s,a)$. Then $N_k(s,a) \geq \frac{1}{4}\bar{N}_k(s,a)$, and

$$\min\left\{\sigma\sqrt{\frac{\ell_k(s,a)}{2N_k^+(s,a)}}, C_{\max}\right\} \leq \sigma\sqrt{\frac{\ell_k(s,a)}{2N_k^+(s,a)}} = \sigma\sqrt{\frac{\log(36SA(N_k^+(s,a))^2/\delta)}{2N_k^+(s,a)}}$$

$$\leq \sigma\sqrt{\frac{\log(36SA(\bar{N}_k^+(s,a)/4)^2/\delta)}{\bar{N}_k^+(s,a)/2}} \leq \sigma\sqrt{\frac{2\bar{\ell}_k(s,a)}{\bar{N}_k^+(s,a)}}, \quad (41)$$

where we use that $\log(36SAx^2/\delta)/x$ is non-increasing for $x > \mathrm{e}\sqrt{\frac{\delta}{36SA}}$, where $\mathrm{e}$ is Euler's number.

For the second case, let $\beta_{\mathrm{cnt}}(\delta) > \frac{1}{4}\bar{N}_k(s,a)$. Then,

$$\min\left\{\sigma\sqrt{\frac{\ell_k(s,a)}{2N_k^+(s,a)}}, C_{\max}\right\} \leq C_{\max} < C_{\max}\sqrt{\frac{4\beta_{\mathrm{cnt}}(\delta)}{\bar{N}_k^+(s,a)}} \leq C_{\max}\sqrt{\frac{4\bar{\ell}_k(s,a)}{\bar{N}_k^+(s,a)}}, \quad (42)$$

where we use $\bar{\ell}_k(s,a) = \log\left(36SA(\bar{N}_k^+(s,a))^2/\delta\right) = \beta_{\mathrm{cnt}}(\delta) + \log\left(18(\bar{N}_k^+(s,a))^2\right) \geq \beta_{\mathrm{cnt}}(\delta)$. By combining the two cases, we obtain

$$\min\left\{\sigma\sqrt{\frac{\ell_k(s,a)}{2N_k^+(s,a)}}, C_{\max}\right\} \leq \max\{\sigma, \sqrt{2}C_{\max}\}\sqrt{\frac{2\bar{\ell}_k(s,a)}{\bar{N}_k^+(s,a)}} = \check{\sigma}\sqrt{\frac{2\bar{\ell}_k(s,a)}{\bar{N}_k^+(s,a)}}, \quad (43)$$

where we denote $\check{\sigma} = \max\{\sigma, \sqrt{2}C_{\max}\}$. $\square$

## C.11    PROOF OF THEOREM 5.5

*Proof.* We assume BEAR exploration strategy terminates with $\tau$ iterations, then

$$
\frac{1}{1-\gamma} \max_{(s,a)} \mathcal{C}_\tau(s,a) \stackrel{(a)}{=} \frac{1}{1-\gamma} \max_{(s,a)} \min \left\{ \sigma\sqrt{\frac{\ell_\tau(s,a)}{2N_\tau^+(s,a)}}, C_{\max} \right\}
$$

$$
\stackrel{(b)}{\leq} \frac{1}{1-\gamma} \max_{(s,a)} \check{\sigma}\sqrt{\frac{2\bar{\ell}_\tau(s,a)}{\bar{N}_\tau^+(s,a)}}
$$

$$
= \varepsilon_\tau, \tag{44}
$$

where step (a) follows Lemma 5.1 and step (b) results from Lemma 5.4. Hence, we obtain,

$$
\varepsilon_\tau = \frac{\check{\sigma}}{1-\gamma} \max_{(s,a)} \sqrt{\frac{2\bar{\ell}_\tau(s,a)}{\bar{N}_\tau^+(s,a)}} = \frac{\check{\sigma}}{1-\gamma} \max_{(s,a)} \sqrt{\frac{2\log(36SA(\bar{N}_\tau^+(s,a))^2/\delta)}{\bar{N}_\tau^+(s,a)}}
$$

$$
\geq \frac{\check{\sigma}}{1-\gamma} \sqrt{\frac{2\log(36SA(\bar{N}_\tau^+(s,a))^2/\delta)}{\bar{N}_\tau^+(s,a)}}
$$

Thus,

$$
\bar{N}_\tau^+(s,a) \geq \frac{2\check{\sigma}^2\log(36SA(\bar{N}_\tau^+(s,a))^2/\delta)}{(1-\gamma)^2\varepsilon_\tau^2}
$$

From Lemma C.8, we have

$$
\bar{N}_\tau^+(s,a) = -\frac{4\check{\sigma}^2}{(1-\gamma)^2\varepsilon_\tau^2} W_{-1}\left( -\frac{(1-\gamma)^2\varepsilon_\tau^2}{4\gamma^2\check{\sigma}^2}\sqrt{\frac{\delta}{36SA}} \right)
$$

$$
\leq \frac{8\check{\sigma}^2}{(1-\gamma)^2\varepsilon_\tau^2}\log\left( \frac{4\gamma^2\check{\sigma}^2}{(1-\gamma)^2\varepsilon_\tau^2}\sqrt{\frac{36SA}{\delta}} \right)
$$

$$
= \widetilde{\mathcal{O}}\left( \frac{\check{\sigma}^2}{(1-\gamma)^2\varepsilon_\tau^2} \right) \tag{45}
$$

By summing over $n = \sum_{(s,a)\in\mathcal{S}\times\mathcal{A}} \bar{N}_\tau^+(s,a)$, we obtain the upper bound.

$$
n \leq \widetilde{\mathcal{O}}\left( \frac{\check{\sigma}^2 SA}{(1-\gamma)^2\varepsilon_\tau^2} \right), \tag{46}
$$

where $\check{\sigma} = \max\{\sigma, \sqrt{2}C_{\max}\}$

For consistency with the sample complexity of uniform sampling strategy, we replace $\tau$ with $K$, and obtain

$$
n \leq \widetilde{\mathcal{O}}\left( \frac{\check{\sigma}^2 SA}{(1-\gamma)^2\varepsilon_K^2} \right). \tag{47}
$$

$\square$

## C.12    THEORETICAL RESULTS ON POLICY-CONSTRAINED STRATEGIC EXPLORATION (PCSE)

**Definition C.10.** We define the optimal policy w.r.t. cost, reward, and safety as follows:

- The cost minimization policy: $\pi^{c,*} = \arg\min_{\pi\in\Pi} \mathbb{E}[\sum_t \gamma^t c(s_t,a_t)]$.

- The reward maximization policy: $\pi^{r,*} = \arg\max_{\pi\in\Pi} \mathbb{E}[\sum_t \gamma^t r(s_t,a_t)]$.

- The optimal safe policy: $\pi^* = \arg\max_{\pi\in\Pi_{safe}} \mathbb{E}[\sum_t \gamma^t r(s_t,a_t)]$ where $\Pi_{safe} = \{\pi : \mathbb{E}[\sum_t \gamma^t c(s_t,a_t)] \leq \epsilon\}$

Accordingly, we can have the following relations:

- $\mathbb{E}_{\mu_0}[V^{c,\pi^{c,*}}(s_0)] \le \mathbb{E}_{\mu_0}[V^{c,\pi^*}(s_0)] \le \mathbb{E}_{\mu_0}[V^{c,\pi^{c,*}}(s_0)] + \epsilon$ where the equality normally holds that $V^{c,\pi^{c,*}}(s_0) = 0$.

- $\mathbb{E}_{\mu_0}[V^{r,\pi^*}(s_0)] \le \mathbb{E}_{\mu_0}[V^{r,\pi^{r,*}}(s_0)]$.

Let's define the following symbols:

- $\varepsilon_0 = \frac{1}{4(1-\gamma)}$.

- $\varepsilon_k^\pi = \sup_{\mu_0 \in \Delta^{S \times A}} \mu_0^T (I_{S \times A} - \gamma P_{\mathcal{T}} \pi) \mathcal{C}_k$

- $\varepsilon_k = \max_{\pi \in \Pi_{k-1}} \varepsilon_k^\pi$

We can construct a set of plausibly optimal policies as

$$\Pi_k = \Pi_k^c \cap \Pi_k^r$$

$$\Pi_k^c = \left\{ \pi \in \Delta_S^A : \sup_{\mu_0 \in \Delta^S} \mu_0^T (V_{\widehat{\mathcal{M}} \cup \widehat{c}_k}^{c,\pi} - V_{\widehat{\mathcal{M}} \cup \widehat{c}_k}^{c,*}) \le 4\varepsilon_k + 2\epsilon \right\}$$

$$\Pi_k^r = \left\{ \pi \in \Delta_S^A : \inf_{\mu_0 \in \Delta^S} \mu_0^T \left( V_{\widehat{\mathcal{M}}}^{r,\pi} - V_{\widehat{\mathcal{M}}}^{r,\widehat{\pi}^*} \right) \ge \mathfrak{R}_k \right\},$$

where $\mathfrak{R}_k = \frac{2\gamma R_{\max}}{(1-\gamma)^2} \| P_{\mathcal{T}} - \widehat{P_{\mathcal{T}}} \|_\infty + \frac{\gamma R_{\max}}{(1-\gamma)^2} \| (\pi^* - \widehat{\pi}^*) \|_\infty$.

**Lemma C.11.** *($\pi^*$ propagation). Under the good event $\mathcal{E}_k$, if $\pi^*, \widehat{\pi}_k^* \in \Pi_{k-1}^c$ then $\pi^* \in \Pi_k^c$*

*Proof.* Given a $c \in \mathcal{C}_\mathfrak{P}$, we can show:

$$\sup_{\mu_0 \in \Delta^S} \mu_0^T \left( V_{\widehat{\mathcal{M}} \cup \widehat{c}_k}^{c,\pi^*} - V_{\widehat{\mathcal{M}} \cup \widehat{c}_k}^{c,*} \right) = \sup_{\mu_0 \in \Delta^S} \mu_0^T \left( V_{\widehat{\mathcal{M}} \cup \widehat{c}_k}^{c,\pi^*} - V_{\widehat{\mathcal{M}} \cup c}^{c,\pi^*} + V_{\widehat{\mathcal{M}} \cup c}^{c,\pi^*} - V_{\widehat{\mathcal{M}} \cup \widehat{c}_k}^{c,*} \right) \qquad (48)$$

$$\overset{(i)}{\le} \sup_{\mu_0 \in \Delta^S} \mu_0^T \left( \varepsilon_k + V_{\widehat{\mathcal{M}} \cup c}^{c,\pi^*} - V_{\widehat{\mathcal{M}} \cup \widehat{c}_k}^{c,*} \right)$$

$$\overset{(ii)}{\le} 2\varepsilon_k + 2\epsilon,$$

which demonstrates that $\pi^* \in \Pi_k^c$.

- (i) holds since

$$|V_{\widehat{\mathcal{M}} \cup \widehat{c}_k}^{c,\pi^*} - V_{\widehat{\mathcal{M}} \cup c}^{c,\pi^*}| \le (I_S - \gamma \pi^* P_{\mathcal{T}})^{-1} \pi^* |\widehat{c}_k - c|$$

$$\le (I_S - \gamma \pi^* P_{\mathcal{T}})^{-1} \pi^* \mathcal{C}_k,$$

where

  - The first inequality follows (Metelli et al., 2021, Lemma B.2) (treat $\widehat{r}_k = -\widehat{c}_k$ and $r = -c$).
  - The second inequality is due to the good event definition in Lemma C.5.

As a result:

$$\sup_{\mu_0 \in \Delta^S} \mu_0^T \left( V_{\widehat{\mathcal{M}} \cup \widehat{c}_k}^{c,\pi^*} - V_{\widehat{\mathcal{M}} \cup c}^{c,\pi^*} \right) = \varepsilon_k^{\pi^*} \le \max_{\pi \in \Pi_{k-1}^c} \varepsilon_k^\pi = \varepsilon_k \qquad (49)$$

- (ii) holds since

$$
\begin{aligned}
V^{c,\pi^*}_{\widehat{\mathcal{M}}\cup c} - V^{c,*}_{\widehat{\mathcal{M}}\cup \widehat{c}_k} &= V^{c,\pi^*}_{\widehat{\mathcal{M}}\cup c} - V^{c,\widehat{\pi}_k^{c,*}}_{\widehat{\mathcal{M}}\cup \widehat{c}_k} \\
&\le V^{c,\pi^{c,*}}_{\widehat{\mathcal{M}}\cup c} - V^{c,\widehat{\pi}_k^{c,*}}_{\widehat{\mathcal{M}}\cup \widehat{c}_k} + \epsilon \\
&= \min_\pi V^{c,\pi}_{\widehat{\mathcal{M}}\cup c} - \min_\pi V^{c,\pi}_{\widehat{\mathcal{M}}\cup \widehat{c}_k} + \epsilon \\
&\le \min_{\pi' \in \Pi^c_{k-1}} V^{c,\pi'}_{\widehat{\mathcal{M}}\cup c} - \min_{\pi' \in \Pi^c_{k-1}} V^{c,\pi'}_{\widehat{\mathcal{M}}\cup \widehat{c}_k} + 2\epsilon \\
&\le \max_{\pi \in \Pi^c_{k-1}} \left| V^{c,\pi}_{\widehat{\mathcal{M}}\cup \widehat{c}_k} - V^{c,\pi}_{\widehat{\mathcal{M}}\cup c} \right| + 2\epsilon,
\end{aligned}
$$

where

- The first inequality utilizes $\mathbb{E}_{\mu_0}[V^{c,\pi^{c,*}}(s_0)] + \epsilon \ge \mathbb{E}_{\mu_0}[V^{c,\pi^*}(s_0)]$.
- The second inequality utilizes $\forall c, \mathbb{E}_{\mu_0}[V^{c,\pi^{c,*}}(s_0)] \le \mathbb{E}_{\mu_0}[V^{c,\pi^*}(s_0)] \le \mathbb{E}_{\mu_0}[V^{c,\pi^{c,*}}(s_0)] + \epsilon$ for $\epsilon > 0$ and the assumption that $\pi^*, \widehat{\pi}_k^* \in \Pi^c_{k-1}$.
- The third inequality results from Lemma C.12.

By following the inequality (49), we have:

$$
\max_{\pi \in \Pi^c_{k-1}} \sup_{\mu_0 \in \Delta^S} \mu_0^T \left( V^{c,\pi}_{\widehat{\mathcal{M}}\cup \widehat{c}_k} - V^{c,\pi}_{\widehat{\mathcal{M}}\cup c} \right) + 2\epsilon = \varepsilon_k + 2\epsilon
$$

$\square$

**Lemma C.12.**

$$
\max_x f(x) - \max_x g(x) \le \max_x (f(x) - g(x))
$$
$$
\min_x f(x) - \min_x g(x) \le \max_x (f(x) - g(x))
$$

*Proof.* For the first inequality, suppose $x_1 = \arg\max f(x)$ and $x_2 = \arg\max g(x)$, then we have,

$$
\max_x f(x) - \max_x g(x) = f(x_1) - g(x_2) \le f(x_1) - g(x_1) \le \max_x (f(x) - g(x))
$$

For the second inequality, suppose $x_3 = \arg\min f(x)$ and $x_4 = \arg\min g(x)$, then we have,

$$
\min_x f(x) - \min_x g(x) = f(x_3) - g(x_4) \le f(x_4) - g(x_4) \le \max_x (f(x) - g(x))
$$

$\square$

**Lemma C.13.** *Under the good event $\mathcal{E}_k$, if $\widehat{\pi}_k^*, \xi \in \Pi^c_{k-1}$ and $\xi \notin \Pi^c_k$, then $\xi$ is suboptimal for some cost $\widehat{c}_{k'} \in \mathcal{R}_{\widehat{\mathfrak{P}}_{k'}}$ for all $k' \ge k$.*

*Proof.* Let's consider the following decomposition:

$$
\begin{aligned}
V^{c,\xi}_{\widehat{\mathcal{M}}\cup \widehat{c}_{k'}} - V^{c,*}_{\widehat{\mathcal{M}}\cup \widehat{c}_{k'}} &\overset{(i)}{\ge} V^{c,\xi}_{\widehat{\mathcal{M}}\cup \widehat{c}_{k'}} - V^{c,\widehat{\pi}_k^{c,*}}_{\widehat{\mathcal{M}}\cup \widehat{c}_{k'}} \\
&= V^{c,\xi}_{\widehat{\mathcal{M}}\cup \widehat{c}_{k'}} - V^{c,\xi}_{\widehat{\mathcal{M}}\cup \widehat{c}_k} + V^{c,\xi}_{\widehat{\mathcal{M}}\cup \widehat{c}_k} - V^{c,\widehat{\pi}_k^{c,*}}_{\widehat{\mathcal{M}}\cup \widehat{c}_k} + V^{c,\widehat{\pi}_k^{c,*}}_{\widehat{\mathcal{M}}\cup \widehat{c}_k} - V^{c,\widehat{\pi}_k^{c,*}}_{\widehat{\mathcal{M}}\cup \widehat{c}_{k'}} \\
&\overset{(ii)}{\ge} -4\varepsilon_k + V^{c,\xi}_{\widehat{\mathcal{M}}\cup \widehat{c}_k} - V^{c,\widehat{\pi}_k^{c,*}}_{\widehat{\mathcal{M}}\cup \widehat{c}_k} \\
&\overset{(iii)}{>} 2\epsilon
\end{aligned}
$$

which indicates that $\xi$ cannot be optimal for $k' \ge k$.

- (i) holds since $V^{c,\widehat{\pi}_k^{c,*}}_{\widehat{\mathcal{M}}\cup \widehat{c}_{k'}} \ge V^{c,\widehat{\pi}_{k'}^{c,*}}_{\widehat{\mathcal{M}}\cup \widehat{c}_{k'}} = V^{c,*}_{\widehat{\mathcal{M}}\cup \widehat{c}_{k'}}$,

- (ii) holds by following (Metelli et al., 2021, Lemma B.5) (treat $\pi = \widehat{\pi}_k^{c,*}$ and $\pi = \xi$ respectively, while $\widehat{r}_k = \widehat{c}_k$ and $\widehat{r}_{k'} = \widehat{c}_{k'}$)

$$\sup_{\mu_0 \in \Delta^S} \mu_0^T \left( V_{\widehat{\mathcal{M}} \cup \widehat{c}_{k'}}^{c,\widehat{\pi}_k^{c,*}} - V_{\widehat{\mathcal{M}} \cup \widehat{c}_k}^{c,\widehat{\pi}_k^{c,*}} \right) \leq 2\varepsilon_k^{\widehat{\pi}_k^{c,*}} \leq 2\varepsilon_k$$

$$\sup_{\mu_0 \in \Delta^S} \mu_0^T \left( V_{\widehat{\mathcal{M}} \cup \widehat{c}_k}^{c,\xi} - V_{\widehat{\mathcal{M}} \cup \widehat{c}_{k'}}^{c,\xi} \right) \leq 2\varepsilon^\xi \leq 2\varepsilon_k$$

- (iii) holds since according to the definition of $\Pi_k^c$ and considering our assumption that $\xi \notin \Pi_k^c$, we have:

$$\sup_{\mu_0 \in \Delta^S} \mu_0^T \left( V_{\widehat{\mathcal{M}} \cup \widehat{c}_k}^{c,\xi} - V_{\widehat{\mathcal{M}} \cup \widehat{c}_k}^{c,*} \right) > 4\varepsilon_k + 2\epsilon$$

$\square$

**Lemma C.14.** *If $\varepsilon_0 = \frac{1}{4(1-\gamma)}$, then for every $k \geq 0$, it holds that $\pi^*, \widehat{\pi}_{k+1}^* \in \Pi_k^c$.*

*Proof.* We prove the result by induction on $k$. For $k = 0$, for every policy $\pi \in \Delta_S^A$, we have $\sup_{\mu_0 \in \Delta^S} \mu_0^T \left( V_{\widehat{\mathcal{M}} \cup \widehat{c}_0}^{c,\pi} - V_{\widehat{\mathcal{M}} \cup \widehat{c}_0}^{c,*} \right) \leq \frac{1}{1-\gamma} \leq 4\varepsilon_0 \leq 4\varepsilon_0 + \epsilon$. Thus, $\Pi_0^c$ contains all the policies, i.e., $\Pi_0^c = \Delta_S^A$, and in particular $\pi^*, \widehat{\pi}_1^* \in \Pi_0^c$. Suppose that for every $1 \leq k' < k$ the statement holds, we aim to prove that the statement also holds for $k$. Let $k' = k - 1$, from the inductive hypothesis we have that $\pi^*, \widehat{\pi}_k^* \in \Pi_{k-1}^c$. Then, from Lemma C.11, it holds that $\pi^* \in \Pi_k^c$. If $\widehat{\pi}_{k+1}^* \in \Pi_k^c$, the proof is finished. If $\widehat{\pi}_{k+1}^* \notin \Pi_k^c$, we prove by contradiction. Let $1 \leq j \leq k$ be the iteration such that $\widehat{\pi}_{k+1}^* \in \Pi_{j-1}^c$ and $\widehat{\pi}_{k+1}^* \notin \Pi_j^c$. Note that this assumption always holds, since $\Pi_0^c$ contains all policies. Recalling the inductive hypothesis, we have that $\widehat{\pi}_j^* \in \Pi_{j-1}^c$. Thus, from Lemma C.13, it must be that $\widehat{\pi}_{k+1}^*$ is suboptimal for all $j' \geqslant j$, in particular for $j' = k + 1$, which brings about a contradiction. $\square$

**Lemma C.15.** *It holds that $\pi^* \in \Pi_k^r$, where:*

$$\Pi_k^r = \left\{ \pi \in \Delta_S^A : \inf_{\mu_0 \in \Delta^S} \mu_0^T \left( V_{\widehat{\mathcal{M}}}^{r,\pi} - V_{\widehat{\mathcal{M}}}^{r,\widehat{\pi}^*} \right) \geq \mathfrak{R}_k \right\} \text{ where}$$

$$\mathfrak{R}_k = \frac{2\gamma R_{\max}}{(1-\gamma)^2} \|P_{\mathcal{T}} - \widehat{P_{\mathcal{T}}}\|_\infty + \frac{\gamma R_{\max}}{(1-\gamma)^2} \|(\pi^* - \widehat{\pi}^*)\|_\infty$$

*Proof.* We should show if $\pi \in \Pi_k^r$, we will have $V_{\mathcal{M}}^{r,\pi} \geq V_{\mathcal{M}}^{r,\pi^*}$.

$$V_{\widehat{\mathcal{M}}}^{r,\pi} - V_{\widehat{\mathcal{M}}}^{r,\widehat{\pi}^*} = V_{\widehat{\mathcal{M}}}^{r,\pi} - V_{\mathcal{M}}^{r,\pi} + V_{\mathcal{M}}^{r,\pi} - V_{\mathcal{M}}^{r,\pi^*} + V_{\mathcal{M}}^{r,\pi^*} - V_{\mathcal{M}}^{r,\widehat{\pi}^*} + V_{\mathcal{M}}^{r,\widehat{\pi}^*} - V_{\widehat{\mathcal{M}}}^{r,\widehat{\pi}^*}$$

$$\overset{(i,ii,iii)}{\leq} \frac{2\gamma R_{\max}}{(1-\gamma)^2} \|P_{\mathcal{T}} - \widehat{P_{\mathcal{T}}}\|_\infty + \frac{\gamma R_{\max}}{(1-\gamma)^2} \|(\pi^* - \widehat{\pi}^*)\|_\infty + V_{\mathcal{M}}^{r,\pi} - V_{\mathcal{M}}^{r,\pi^*}$$

$$= \mathfrak{R}_k + V_{\mathcal{M}}^{r,\pi} - V_{\mathcal{M}}^{r,\pi^*}$$

Since $\inf_{\mu_0 \in \Delta^S} \mu_0^T \left( V_{\widehat{\mathcal{M}}}^{r,\pi} - V_{\widehat{\mathcal{M}}}^{r,\widehat{\pi}^*} \right) \geq \mathfrak{R}_k$, it must hold that $\inf_{\mu_0 \in \Delta^S} \mu_0^T \left( V_{\mathcal{M}}^{r,\pi} - V_{\mathcal{M}}^{r,\pi^*} \right) \geq 0$

- To show (i), we first follows the simulation Lemma for the state-value function:

$$V_{\widehat{\mathcal{M}}}^{r,\pi} - V_{\mathcal{M}}^{r,\pi} = \gamma(I_S - \gamma\pi\widehat{P_{\mathcal{T}}})^{-1}\pi(\widehat{P_{\mathcal{T}}} - P_{\mathcal{T}})V_{\mathcal{M}}^{r,\pi}$$

Then we derive an upper bound for the difference of these state-values as follows:

$$V_{\widehat{\mathcal{M}}}^{r,\pi} - V_{\mathcal{M}}^{r,\pi} \leq \frac{\gamma}{1-\gamma} \|\pi(\widehat{P_{\mathcal{T}}} - P_{\mathcal{T}})V_{\mathcal{M}}^{r,\pi}\|_\infty$$

$$\leq \frac{\gamma R_{\max}}{(1-\gamma)^2} \|\pi(\widehat{P_{\mathcal{T}}} - P_{\mathcal{T}})\|_\infty$$

$$\leq \frac{\gamma R_{\max}}{(1-\gamma)^2} \|\widehat{P_{\mathcal{T}}} - P_{\mathcal{T}}\|_\infty$$

- (ii) holds due to the policy mismatch Lemma C.4:

$$V_{\mathcal{M}}^{r,\pi^*} - V_{\mathcal{M}}^{r,\widehat{\pi}^*} = \gamma(I_{\mathcal{S}} - \gamma\widehat{\pi}^* P_{\mathcal{T}})^{-1}(\pi^* - \widehat{\pi}^*)P_{\mathcal{T}}V_{\mathcal{M}}^{r,\pi^*}$$

Then we derive an upper bound for the difference of these state-values as follows:

$$V_{\mathcal{M}}^{r,\pi^*} - V_{\mathcal{M}}^{r,\widehat{\pi}^*} \leq \frac{\gamma}{1-\gamma}\|(\pi^* - \widehat{\pi}^*)P_{\mathcal{T}}V_{\mathcal{M}}^{r,\pi^*}\|_\infty$$

$$\leq \frac{\gamma R_{\max}}{(1-\gamma)^2}\|(\pi^* - \widehat{\pi}^*)P_{\mathcal{T}}\|_\infty$$

$$\leq \frac{\gamma R_{\max}}{(1-\gamma)^2}\|(\pi^* - \widehat{\pi}^*)\|_\infty$$

- (iii) holds due to the derivation to (i):

$$V_{\mathcal{M}}^{r,\widehat{\pi}^*} - V_{\widehat{\mathcal{M}}}^{r,\widehat{\pi}^*} \leq \frac{\gamma R_{\max}}{(1-\gamma)^2}\|P_{\mathcal{T}} - \widehat{P_{\mathcal{T}}}\|_\infty$$

Since we can guarantee $V_{\mathcal{M}}^\pi \geq V_{\mathcal{M}}^{\pi^*}$, we know $\pi^* \in \{\pi | V_{\mathcal{M}}^\pi \geq V_{\mathcal{M}}^{\pi^*}\}$. Subsequently, according to our Lemma 4.4, to find the feasible constraint set, the exploration policy should follow the $\pi$ that visits states with larger cumulative rewards. $\qquad\square$

**Lemma C.16.** *Under the good event $\mathcal{E}_k$, let $\tilde{c} \in \arg\min_{c\in\mathcal{C}_{\mathfrak{P}}} \max_{(s,a)\in\mathcal{S}\times\mathcal{A}} |c(s,a) - \widehat{c}_k(s,a)|$. If $\pi \in \Pi_k$ and $\pi^* \in \Pi_{k-1}$, then $\sup_{\mu_0\in\Delta^{\mathcal{S}}} \mu_0^T \left(V_{\widehat{\mathcal{M}}\cup\tilde{c}}^{c,\pi} - V_{\widehat{\mathcal{M}}\cup\tilde{c}}^{c,*}\right) \leq 6\varepsilon_k + \epsilon$.*

*Proof.*

$$\sup_{\mu_0\in\Delta^{\mathcal{S}}} \mu_0^T \left(V_{\widehat{\mathcal{M}}\cup\tilde{c}}^{c,\pi} - V_{\widehat{\mathcal{M}}\cup\tilde{c}}^{c,*}\right)$$

$$\leq \underbrace{\sup_{\mu_0\in\Delta^{\mathcal{S}}} \mu_0^T \left(V_{\widehat{\mathcal{M}}\cup\tilde{c}}^{c,\pi} - V_{\widehat{\mathcal{M}}\cup\widehat{c}_k}^{c,\pi}\right)}_{(a)} + \underbrace{\sup_{\mu_0\in\Delta^{\mathcal{S}}} \mu_0^T \left(V_{\widehat{\mathcal{M}}\cup\widehat{c}_k}^{c,\pi} - V_{\widehat{\mathcal{M}}\cup\widehat{c}_k}^{c,*}\right)}_{(b)} + \underbrace{\sup_{\mu_0\in\Delta^{\mathcal{S}}} \mu_0^T \left(V_{\widehat{\mathcal{M}}\cup\widehat{c}_k}^{c,*} - V_{\widehat{\mathcal{M}}\cup\tilde{c}}^{c,*}\right)}_{(c)},$$

$$\leq (\varepsilon_k) + (4\varepsilon_k + \epsilon) + (\varepsilon_k)$$

$$= 6\varepsilon_k + \epsilon$$

where

- (a) holds due to $\sup_{\mu_0\in\Delta^{\mathcal{S}}} \mu_0^T \left(V_{\widehat{\mathcal{M}}\cup\tilde{c}}^{c,\pi} - V_{\widehat{\mathcal{M}}\cup\widehat{c}_k}^{c,\pi}\right) \leq \varepsilon_k^\pi \leq \varepsilon_k$.

- (b) results from $\sup_{\mu_0\in\Delta^{\mathcal{S}}} \mu_0^T \left(V_{\widehat{\mathcal{M}}\cup\widehat{c}_k}^{c,\pi} - V_{\widehat{\mathcal{M}}\cup\widehat{c}_k}^{c,*}\right) \leq 4\varepsilon_k + \epsilon$, since $\pi \in \Pi_k$.

- (c) follows Eq. (49), recalling the definition of $\tilde{c}$.

$\qquad\square$

## C.13 PROOF OF THEOREM 5.6

*Proof.* First of all, note that PCSE for ICRL is optimizing a tighter bound (Corollary C.6 (2)), compared with that of BEAR exploration strategy (Corollary C.6 (1)). Thus, results of Theorem 5.5, namely sample complexity of BEAR strategy, still apply to PCSE for ICRL, serving as the sample complexity in the worst case. Let's begin the problem-dependent analysis. Recall the definition of advantage function $A_{\widehat{\mathcal{M}}\cup\tilde{c}}^{c,*}(s,a) = Q_{\widehat{\mathcal{M}}\cup\tilde{c}}^{c,*}(s,a) - V_{\widehat{\mathcal{M}}\cup\tilde{c}}^{c,*}(s)$. Suppose we have derived a value of $\bar{N}_K(s,a)$ so that for all $(s,a) \in \mathcal{S} \times \mathcal{A}$, it holds that:

$$\mathcal{C}_K(s,a) = \min\left\{\sigma\sqrt{\frac{\ell_K(s,a)}{2N_K^+(s,a)}}, C_{\max}\right\} \leq \check{\sigma}\sqrt{\frac{2\bar{\ell}_K(s,a)}{\bar{N}_K^+(s,a)}} \leq \frac{-\min_{a'\in\mathcal{A}} A_{\widehat{\mathcal{M}}\cup\tilde{c}}^{c,*}(s,a')\varepsilon_K}{6\varepsilon_{K-1} + \epsilon}. \quad (50)$$

From Lemma C.8, we obtain

$$
\begin{aligned}
\bar{N}_k^+(s,a) &= \frac{2\check{\sigma}^2(6\varepsilon_{K-1}+\epsilon)^2 \bar{\ell}_K(s,a)}{(\min_{a'\in\mathcal{A}} A_{\widehat{\mathcal{M}}\cup\tilde{c}}^{c,*}(s,a'))^2 \varepsilon_K^2} \\
&= -\frac{4\sigma^2(6\varepsilon_{K-1}+\epsilon)^2}{(\min_{a'\in\mathcal{A}} A_{\widehat{\mathcal{M}}\cup\tilde{c}}^{c,*}(s,a'))^2 \varepsilon_K^2} W_{-1}\left( \frac{(\min_{a'\in\mathcal{A}} A_{\widehat{\mathcal{M}}\cup\tilde{c}}^{c,*}(s,a'))^2 \varepsilon_K^2}{4\sigma^2(6\varepsilon_{K-1}+\epsilon)^2} \sqrt{\frac{\delta}{36SA}} \right) \\
&= \frac{8\sigma^2(6\varepsilon_{K-1}+\epsilon)^2}{(\min_{a'\in\mathcal{A}} A_{\widehat{\mathcal{M}}\cup\tilde{c}}^{c,*}(s,a'))^2 \varepsilon_K^2} \log\left( \frac{4\sigma^2(6\varepsilon_{K-1}+\epsilon)^2}{(\min_{a'\in\mathcal{A}} A_{\widehat{\mathcal{M}}\cup\tilde{c}}^{c,*}(s,a'))^2 \varepsilon_K^2} \sqrt{\frac{36SA}{\delta}} \right) \\
&= \widetilde{\mathcal{O}}\left( \frac{\sigma^2(6\varepsilon_{K-1}+\epsilon)^2}{(\min_{a'\in\mathcal{A}} A_{\widehat{\mathcal{M}}\cup\tilde{c}}^{c,*}(s,a'))^2 \varepsilon_K^2} \right).
\end{aligned}
\tag{51}
$$

Summing over $n = \sum_{(s,a)\in\mathcal{S}\times\mathcal{A}} \bar{N}_k^+(s,a)$ and recalling the sample complexity of BEAR exploration strategy in Theorem 5.5, we obtain

$$
n \leq \widetilde{\mathcal{O}}\left( \min\left\{ \frac{\gamma^2\check{\sigma}^2 SA}{(1-\gamma)^2\varepsilon_K^2}, \frac{\sigma^2(6\varepsilon_{K-1}+\epsilon)^2 SA}{(\min_{(s,a)} A_{\widehat{\mathcal{M}}\cup\tilde{c}}^{c,*}(s,a))^2\varepsilon_K^2} \right\} \right).
\tag{52}
$$

Next, we explain the rationale for assumption in Eq. (50). We have for every $\pi \in \Pi_k$,

$$
\begin{aligned}
\|e_k(s,a;\pi^*,\widehat{\pi}^*)\|_\infty &\overset{(a)}{\leq} \gamma \left\| \max_{\pi\in\{\widehat{\pi}^*,\pi^*\}} \widetilde{V}_{\widehat{\mathcal{M}}\cup|c-\widehat{c}_k|}^{|c-\widehat{c}_k|,\pi} \right\|_\infty \\
&\overset{(b)}{\leq} \gamma\mu_0^T(I_\mathcal{S}-\gamma\pi\widehat{P_\mathcal{T}})^{-1}\pi\mathcal{C}_k \\
&\overset{(c)}{\leq} \frac{\gamma\varepsilon_K}{6\varepsilon_K+\epsilon}\mu_0^T(I_\mathcal{S}-\gamma\pi\widehat{P_\mathcal{T}})^{-1}\pi\left(-A_{\widehat{\mathcal{M}}\cup\tilde{c}}^{c,*}\right) \\
&\overset{(d)}{=} \frac{\gamma\varepsilon_K}{6\varepsilon_{K-1}+\epsilon}\mu_0^T\left(V_{\widehat{\mathcal{M}}\cup\tilde{c}}^{c,\pi}-V_{\widehat{\mathcal{M}}\cup\tilde{c}}^{c,*}\right) \overset{(e)}{\leq} \varepsilon_K,
\end{aligned}
\tag{53}
$$

- (a) follows the step (h) in Lemma 4.8.

- (b) follows the matrix form Bellman equation for value function.

- (c) is based on the assumption in Eq. (50).

- (d) follows (Metelli et al., 2021, Lemma B.3), where we treat $r = -\tilde{c}$ and note that $V_{\widehat{\mathcal{M}}\cup(-\tilde{c})}^\pi = -V_{\widehat{\mathcal{M}}\cup\tilde{c}}^\pi$, $Q_{\widehat{\mathcal{M}}\cup(-\tilde{c})}^\pi = -Q_{\widehat{\mathcal{M}}\cup\tilde{c}}^\pi$ and $A_{\widehat{\mathcal{M}}\cup(-\tilde{c})}^\pi = -A_{\widehat{\mathcal{M}}\cup\tilde{c}}^\pi$.

- (e) results from Lemma C.16 and $\gamma < 1$.

$\square$

## C.14 OPTIMIZATION PROBLEM AND THE TWO-TIMESCALE STOCHASTIC APPROXIMATION

We can now formulate the optimization problem.

$$
\begin{aligned}
\varepsilon_{k+1} = \sup_{\substack{\mu_0\in\Delta^\mathcal{S} \\ \pi\in\Pi_k}} &\mu_0^T(I_{\mathcal{S}\times\mathcal{A}}-\gamma P_\mathcal{T}\pi)\mathcal{C}_{k+1} \\
\text{s.t.}\quad \Pi_k &= \Pi_k^c \cap \Pi_k^r \\
\Pi_k^c &= \left\{ \pi\in\Delta_\mathcal{S}^\mathcal{A} : \sup_{\mu_0\in\Delta^\mathcal{S}} \mu_0^T(V_{\widehat{\mathcal{M}}\cup\widehat{c}_k}^{c,\pi}-V_{\widehat{\mathcal{M}}\cup\widehat{c}_k}^{c,*}) \leq 4\varepsilon_k+2\epsilon \right\} \\
\Pi_k^r &= \left\{ \pi\in\Delta_\mathcal{S}^\mathcal{A} : \inf_{\mu_0\in\Delta^\mathcal{S}} \mu_0^T\left(V_{\widehat{\mathcal{M}}}^{r,\pi}-V_{\widehat{\mathcal{M}}}^{r,\widehat{\pi}^*}\right) \geq \mathfrak{R}_k \right\}
\end{aligned}
\tag{54}
$$

where $\mathfrak{R}_k = \frac{2\gamma R_{\max}}{(1-\gamma)^2}\|P_\mathcal{T}-\widehat{P_\mathcal{T}}\|_\infty + \frac{\gamma R_{\max}}{(1-\gamma)^2}\|(\pi^*-\widehat{\pi}^*)\|_\infty$.

Recall that the discounted normalized occupancy measure is defined by

$$\rho_{\mathcal{M}}^{\pi}(s, a) = (1 - \gamma) \sum_{t=0}^{\infty} \gamma^t \mathbb{P}_{\mu_0}^{\pi}(s_t = s, a_t = a), \tag{55}$$

where the normalizer $(1 - \gamma)$ makes $\rho_{\mathcal{M}}^{\pi}(s, a)$ a probability measure, i.e., $\sum_{(s,a)} \rho_{\mathcal{M}}^{\pi}(s, a) = 1$.

The promised relationship between reward value function and occupancy measure is as follows:

$$
\begin{aligned}
(1 - \gamma) V_{\mathcal{M}}^{r,\pi} &\overset{(a)}{=} (1 - \gamma) \mathbb{E}_{\mu_0, \pi, P_{\mathcal{T}}} \Big[ \sum_{t=0}^{\infty} \gamma^t r(s_t, a_t) \Big] \\
&= (1 - \gamma) \sum_{t=0}^{\infty} \gamma^t \sum_{(s,a)} \mathbb{P}_{\mu_0}^{\pi}(s_t = s, a_t = a) r(s_t = s, a_t = a) \\
&\overset{(b)}{=} \sum_{(s,a)} \Big[ (1 - \gamma) \sum_{t=0}^{\infty} \gamma^t \mathbb{P}_{\mu_0}^{\pi}(s_t = s, a_t = a) \Big] \cdot \Big[ r(s_t = s, a_t = a) \Big] \\
&= \langle \rho_{\mathcal{M}}^{\pi}, r \rangle, 
\end{aligned}
\tag{56}
$$

where step (a) follows the definition of the reward state-value function, and step (b) exchanges the order of two summations.

Similarly, concerning the cost function, the relationship between the cost value function and (the same) occupancy measure is as follows:

$$
\begin{aligned}
(1 - \gamma) V_{\mathcal{M}}^{c,\pi} &= (1 - \gamma) \mathbb{E}_{\pi, P_{\mathcal{T}}} \Big[ \sum_{t=0}^{\infty} \gamma^t c(s_t, a_t) \Big] \\
&= (1 - \gamma) \sum_{t=0}^{\infty} \gamma^t \sum_{(s,a)} \mathbb{P}_{\mu_0}^{\pi}(s_t = s, a_t = a) c(s_t = s, a_t = a) \\
&= \sum_{(s,a)} \Big[ (1 - \gamma) \sum_{t=0}^{\infty} \gamma^t \mathbb{P}_{\mu_0}^{\pi}(s_t = s, a_t = a) \Big] \cdot [c(s_t = s, a_t = a)] \\
&= \langle \rho_{\mathcal{M}}^{\pi}, c \rangle.
\end{aligned}
\tag{57}
$$

For simplicity, denote the occupancy measure vector $\rho_{\mathcal{M}}^{\pi}$ as vector $x$. As a result, the optimization problem (54) can be recast as a linear program.

$$
\begin{aligned}
\min_{x} \quad & - \langle x, \mathcal{C}_{k+1} \rangle \\
\text{s.t.} \quad & - (1 - \gamma)(V_{\widehat{\mathcal{M}} \cup \widehat{c_k}}^{c,*} + 4\varepsilon_k + 2\epsilon) + \langle x, \widehat{c_k} \rangle \le 0 \\
& (1 - \gamma)(V_{\widehat{\mathcal{M}}}^{r,\widehat{\pi}^*} + \mathfrak{R}_k) - \langle x, r \rangle \le 0
\end{aligned}
\tag{58}
$$

To solve this linear program, we introduce the Lagrangian function and calculate its saddle points by solving the dual problem. The Lagrangian of this primal problem is defined as:

$$
\begin{aligned}
L(x, \lambda) = & - \langle x, \mathcal{C}_{k+1} \rangle + \lambda_1 \Big( -(1 - \gamma)(V_{\widehat{\mathcal{M}} \cup \widehat{c_k}}^{c,*} + 4\varepsilon_k + 2\epsilon) + \langle x, \widehat{c_k} \rangle \Big) \\
& + \lambda_2 \Big( (1 - \gamma)(V_{\widehat{\mathcal{M}}}^{r,\widehat{\pi}^*} + \mathfrak{R}_k) - \langle x, r \rangle \Big),
\end{aligned}
\tag{59}
$$

where $\lambda = [\lambda_1, \lambda_2]^T$ is a nonnegative real vector, composed of so-called Lagrangian multipliers. The dual problem is defined as:

$$\min_{x} \max_{\lambda \ge 0} L(x, \lambda). \tag{60}$$

To solve this dual problem, we follow a gradient-based approach, known as the two-timescale stochastic approximation (Szepesvári, 2021), . At time step $k$, the following updates are conducted,

$$x_{k+1} - x_k = -a_k(L_x'(x_k, \lambda_k) + W_k), \tag{61}$$

$$\lambda_{k+1} - \lambda_k = b_k(L_\lambda'(x_k, \lambda_k) + U_k), \tag{62}$$

where the two coefficients $a_k \ll b_k$, satisfying $\sum_k a_k = \sum b_k = \infty$, $\sum a_k^2 < \infty$ and $\sum b_k^2 < \infty$. Under this condition, the convergence is guaranteed in the limit. As an option, we can set $a_k = c/k$, $b_k = c/k^{0.5+\kappa}$, with $c$ being a constant and $0 < \kappa < 0.5$. $W_k$ and $U_k$ are two zero-mean noise sequences. The two gradients are:

$$L'_x(x_k, \lambda_k) = -\mathcal{C}_{k+1} + \lambda_1 \widehat{c}_k - \lambda_2 r, \tag{63}$$

$$L'_\lambda(x_k, \lambda_k) = \begin{bmatrix} L'_{\lambda_1}(x_k, \lambda_k) \\ L'_{\lambda_2}(x_k, \lambda_k) \end{bmatrix} = \begin{bmatrix} -(1-\gamma)(V^{c,*}_{\widehat{\mathcal{M}} \cup \widehat{c}_k} + 4\varepsilon_k + 2\epsilon) + \langle x, \widehat{c}_k \rangle \\ (1-\gamma)(V^{r,\widehat{\pi}*}_{\widehat{\mathcal{M}}} + \mathfrak{R}_k) - \langle x, r \rangle \end{bmatrix}. \tag{64}$$

At each time step $k$, the exploration policy can be calculated as,

$$\pi_k(a|s) = \frac{x_k(s,a)}{\sum_a x_k(s,a)}. \tag{65}$$

## D    EXPERIMENTAL DETAILS

We ran experiments on a desktop computer with Intel(R) Core(TM) i5-14400F and NVIDIA GeForce RTX 2080 Ti.

### D.1    DISCRETE ENVIRONMENT

**More details about Gridworld.** In this paper, we create a map with dimensions of $7 \times 7$ units and define four distinct settings, as illustrated in Figure 2. We use two coordinates to represent the location, where the first coordinate corresponds to the vertical axis, and the second coordinate corresponds to the horizontal axis. The agent aims to navigate from a starting location to a target location, while avoiding the given constraints. The agent starts in the lower left cell $(0,0)$, and it has 8 actions which corresponds to 8 adjacent directions, including four cardinal directions (up, down, left, right) as well as the four diagonal directions (upper-left, lower-left, upper-right, lower-right). The reward and target location are the same, which locates in the upper right cell $(6,6)$ for the first, second and fourth Gridworld environment or locates in the upper left cell $(6,0)$ for the third Gridworld environment. If the agent takes an action, then with probability $0.05$ this action fails and the agent moves in any viable random direction (including the direction this action leads to) with uniform probabilities. The reward in the reward state cell is $1$, while all other cells have a $0$ reward. The cost in a constrained location is also $1$. The game continues until a maximum time step of $50$ is reached.

**Comparison Methods.** The upper confidence bound (UCB) exploration strategy is derived from the UCB algorithm, which selects an action with the highest upper bound. The maximum-entropy strategy selects an action on a state with the maximum entropy given previous choices of actions. The random strategy uniformly randomly selects a viable action on a state $s$. The $\epsilon$-greedy strategy selects an action based on the $\epsilon$-greedy algorithm, balancing exploration and exploitation with the exploration parameter $\epsilon = 1/\sqrt{k}$.

**More details about Figure 3.** In Figure 3, we plot the mean and $68\%$ confidence interval (1-sigma error bar) computed with 5 random seeds ($123456, 123, 1234, 36, 34$) and exploration episodes $n_e = 1$. The six exploration strategies compared in Figure 3 include: upper confidence bound (UCB), maximum-entropy, random, BEAR, $\epsilon$-greedy and PCSE. Meanwhile, we utilize the running score to make the training process more resilient to environmental stochasticity: $running\_score = 0.2 * running\_score + 0.8 * iteration\_rewards$ (or $iteration\_costs$) (Luo et al., 2022).

### D.2    WEIGHTED GENERALIZED INTERSECTION OVER UNION (WGIOU)

In this section, we present the methodology for designing the metric that assesses the similarity between the estimated and ground-truth cost functions, which we refer to as WGIoU. We commence our discussion by explaining IoU, followed by GIoU, and ultimately introduce the novel concept of WGIoU for ICRL.

Intersection Over Union (IoU) score is a commonly used metric in the field of object detection, which measures how similar two sets are. The IoU score is bounded in $[0, 1]$ (0 being no overlap between

two sets and 1 being complete overlap). Suppose there are two sets $X$ and $Y$,

$$\text{IoU} = \frac{|X \cap Y|}{|X \cup Y|}.$$

Note that IoU equals to zero for all two sets with no overlap, which is a rough metric and incurs the problem of vanishing gradients. To further measure the difference between two sets with no overlap, Signed IoU (SIoU) (Simonelli et al., 2019) and Generalized IoU (GIoU) (Rezatofighi et al., 2019) are proposed. Both SIoU and GIoU are bounded in $[-1, 1]$. However, SIoU is constrained to rectangular bounding box, which is not the case for cost function. By contrast, GIoU is not limited to rectangular box. Thus, GIoU is more suitable for comparing the distance between the estimated cost function and the ground-truth cost function.

$$\text{GIoU} = \text{IoU} - \frac{|Z \backslash (X \cup Y)|}{|Z|},$$

where set $Z$ is the minimal enclosing convex set that contains both $X$ and $Y$. Taking cost function into account, the difference between $\widehat{c}_k$ the estimated cost function at iteration $k$ and $c$ the ground-truth cost function is calculated as,

$$\text{GIoU} = \frac{|c \cap \widehat{c}_k|}{|c \cup \widehat{c}_k|} - \frac{|(c \oplus \widehat{c}_k) \backslash (c \cup \widehat{c}_k)|}{|c \oplus \widehat{c}_k|},$$

where $\widehat{c}_k \oplus c$ denotes the enclosing convex matrix of $c$ and $\widehat{c}_k$.

Note that the estimated cost function $\widehat{c}_k$ could have different values, but GIoU only reflects spatial relationship and is unable to represent weight features. To accommodate our settings, weighted GIoU (WGIoU) is proposed, where we measure the distance between a weighted estimated cost function and a uniformly valued (or weighted) ground-truth cost function. WGIoU is also bounded in $[-1, 1]$. To calculate WGIoU, first, remap the cost function to $(\{0\} \cup [1, +\infty))^{\mathcal{S} \times \mathcal{A}}$,

$$\widehat{c}_k^{\circledast}(s, a) = \frac{\widehat{c}_k(s, a)}{\min \left\{ \min_{(s,a) \in \mathcal{S} \times \mathcal{A}}^{+} \widehat{c}_k(s, a), \min_{(s,a) \in \mathcal{S} \times \mathcal{A}}^{+} c(s, a) \right\}}, \tag{66}$$

$$c^{\circledast}(s, a) = \frac{c(s, a)}{\min \left\{ \min_{(s,a) \in \mathcal{S} \times \mathcal{A}}^{+} \widehat{c}_k(s, a), \min_{(s,a) \in \mathcal{S} \times \mathcal{A}}^{+} c(s, a) \right\}}. \tag{67}$$

where $\min_{(s,a) \in \mathcal{S} \times \mathcal{A}}^{+}$ returns the minimum positive value of $\widehat{c}_k$ or $c$ over all $(s, a)$ pairs. Note that $c$ must exceed 0 at certain $(s, a)$, otherwise the cost function are all zeros, indicating an absence of constraint at anywhere. Also note that if $\widehat{c}_k$ are all zeros, let $\widehat{c}_k^{\circledast}(s, a) = 0$ and $c^{\circledast}(s, a) = c(s, a)/\min_{(s,a) \in \mathcal{S} \times \mathcal{A}}^{+} c(s, a)$. Besides the two trivial situations, the above two equations (66 and 67) can be applied naturally.

Then, WGIoU is defined as:

$$\text{WGIoU} = \frac{\langle \widehat{c}_k^{\circledast}, c^{\circledast} \rangle}{\langle \mathbf{1}, \max\{\widehat{c}_k^{\circledast}, c^{\circledast}, \langle \widehat{c}_k^{\circledast}, c^{\circledast} \rangle\} \rangle} + \left( e^{-\langle \mathbf{1}, \max\{\widehat{c}_k^{\circledast}, c^{\circledast}\} \rangle} - 1 \right) \mathbb{1} \left\{ \langle \widehat{c}_k^{\circledast}, c^{\circledast} \rangle = 0 \right\},$$

where $\mathbf{1}$ denotes the vector with appropriate length whose elements are all 1s. The rationale here can be understood by distinguishing two cases. For the first case, there is overlap between $\widehat{c}_k$ and $c$, so the second term in WGIoU is 0. For the first term, for some $(s, a)$, 1) if both $\widehat{c}_k^{\circledast}(s, a) \geq 1$ and $c^{\circledast}(s, a) \geq 1$, WGIoU approaches 1; 2) if either $\widehat{c}_k^{\circledast}(s, a) = 0$ or $c^{\circledast}(s, a) = 0$, WGIoU approaches 0. For the second case, so there is no overlap between $\widehat{c}_k$ and $(s, a)$, the first term in WGIoU is 0. The second term is always below 0 and approaches $-1$ if the estimated and ground-truth cost functions contain large values.

### D.3 CONTINUOUS ENVIRONMENT

**Density model.** Recall that in Definition 5.3 , the concept of pseudo-counts is introduced to analyze the uncertainty of the transition dynamics without a generative model. Here, we abuse the concept of pseudo-counts for generalizing count-based exploration algorithms to the non-tabular settings

(Bellemare et al., 2016). Let $\rho$ be a density model on a finite space $\mathcal{X}$, and $\rho_n(x)$ the probability assigned by the model to $x$ after being trained on a sequence of states $x_1, \ldots, x_n$. Assume $\rho_n(x) > 0$ for all $x, n$. The recoding probability $\rho'_n(x)$ is then the probability the model would assign to $x$ if it was trained on that same $x$ one more time. We call $\rho$ *learning-positive* if $\rho'_n(x) \geq \rho_n(x)$ for all $x_1, \ldots, x_n, x \in \mathcal{X}$. A learning-positive $\rho$ implies $\mathrm{PG}_n(x) \geq 0$ for all $x \in \mathcal{X}$. For learning-positive $\rho$, we define the *pseudo-count* as $\hat{\mathrm{N}}_n(x) = \rho_n(x) \cdot n$, where $n$ is the total count. The pseudo-count generalizes the usual state visitation count function $\mathrm{N}_n(x)$, also called the empirical count function or simply empirical count, which equals to the number of occurrences of a state in the sequence.

**Methods.** We first train a Deep Q Network (DQN) in advance that stores the Q values of the constrained Point Maze environment. This DQN induces the expert policy at any given state. We also train a density model that accounts for calculating the pseudo-count of any given state-action pairs. The agent then collects samples from an unconstrained Point Maze environment where it could violate constraints. For algorithm BEAR, Proximal Policy Optimization (PPO) is utilized to obtain the exploration policy $\pi_k$. For algorithm PCSE, we rank 8 permissible actions for the exploration policy, the action that has a high estimated cost or a high reward is assigned with more probability to choose from. After a rollout of this exploration policy, the density model and accuracy are updated for the selection of the next exploration policy. Multiple rounds of iterations are conducted until the target accuracy is achieved.

**Point Maze.** In this environment, we create a map of $5\mathrm{m} \times 5\mathrm{m}$, where the area of each cell is $1\mathrm{m} \times 1\mathrm{m}$. The center of the map is the original point, i.e. $(0,0)$. The constraint is initially set at the cell centered at $(-1, 0)$. The agent is a 2-DoF ball, force-actuated in the cartesian directions x and y. The reward obtained by the agent depends on where the agent reaches a target goal in a closed maze. The ball is considered to have reached the goal if the Euclidean distance between the ball and the goal is smaller than $0.5\mathrm{m}$. The reward in the reward state cell is $1$, while all other cells have a $0$ reward. The cost in a constrained location is also $1$. The game terminates when a maximum time step of $500$ is reached. The state space dimension is continuous and consists of $4$ dimensions (two as x and y coordinates of the agent and two as the linear velocity in the x and y direction). The action space is discrete and at each state there are $8$ permissible actions ($8$ directions to add a linear force), similar to the action space of Gridworld environment. The environment has certain degree of stochasticity because there is a sampled noise from a uniform distribution to the cell's $(x, y)$ coordinates.

## E    MORE EXPERIMENTAL RESULTS

### E.1    GRIDWORLD ENVIRONMENTS

Figure 7, 8, 9 and 10 show the constraint learning process of six exploration strategies in four Gridworld environments, i.e. Gridwworld-1, 2, 3 and 4. Note that in Figure 8 (Gridworld-2) and Figure 10 (Gridworld-4) only a fraction of ground-truth constraint is learned. This is attributed to ICRL's emphasis on identifying the minimum set of constraints necessary to explain expert behavior. Venturing into unidentified part of ground-truth constraints will not yield any advantages for cumulative rewards.

### E.2    POINT MAZE ENVIRONMENT

Figure 6 shows the constraint learning process of PCSE in the Point Maze environment.

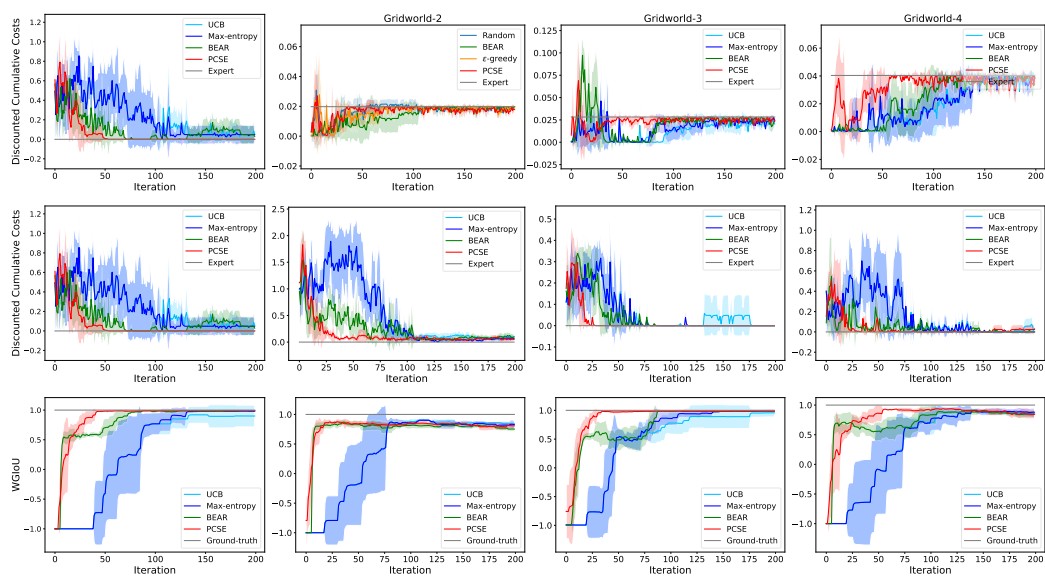

Figure 5: Training curves of discounted cumulative rewards (top), costs (middle), and WGIoU (bottom) for two other exploration strategies in four Gridworld environments.

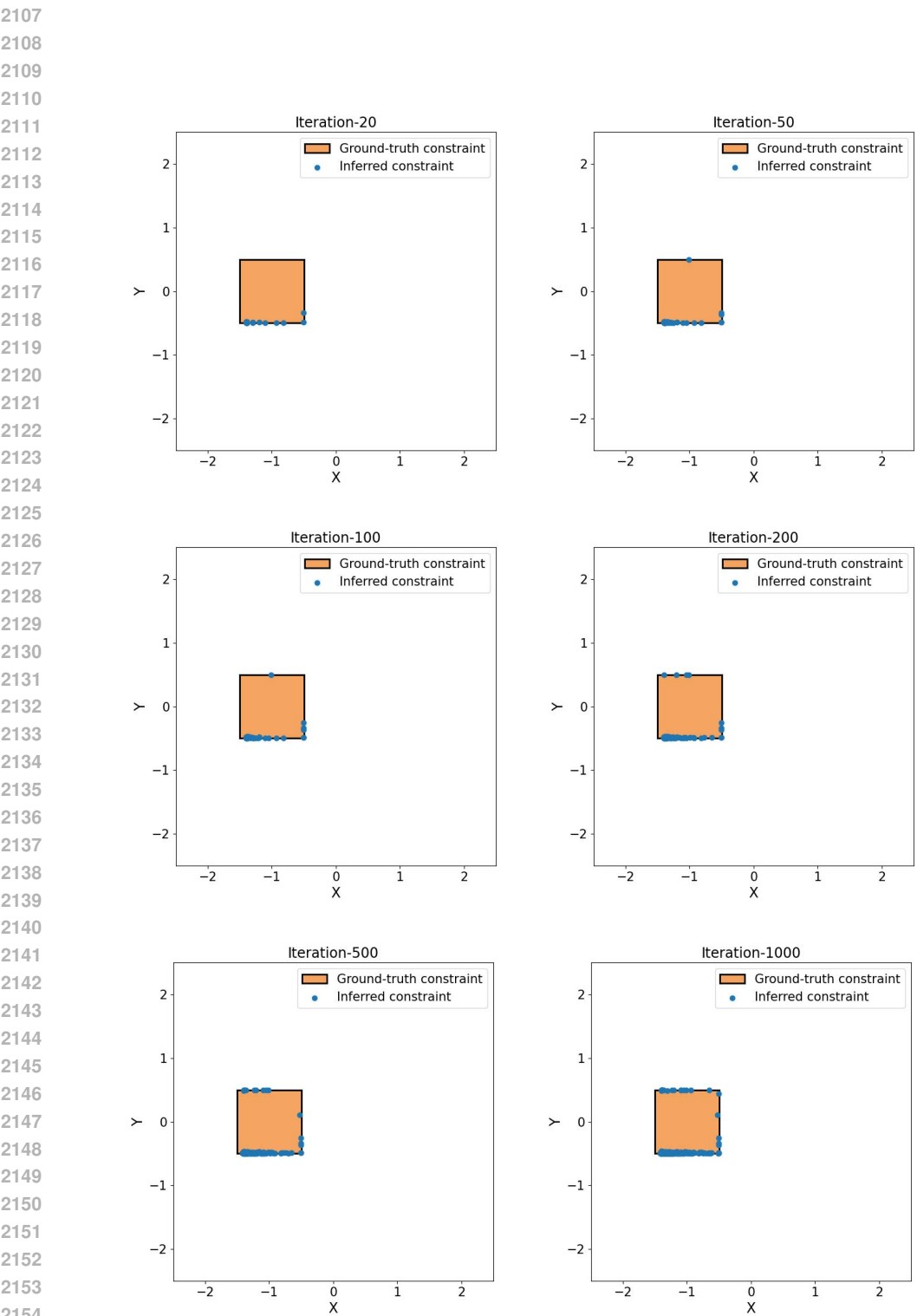

Figure 6: Constraint learning performance of PCSE for ICRL in the Point Maze environment.

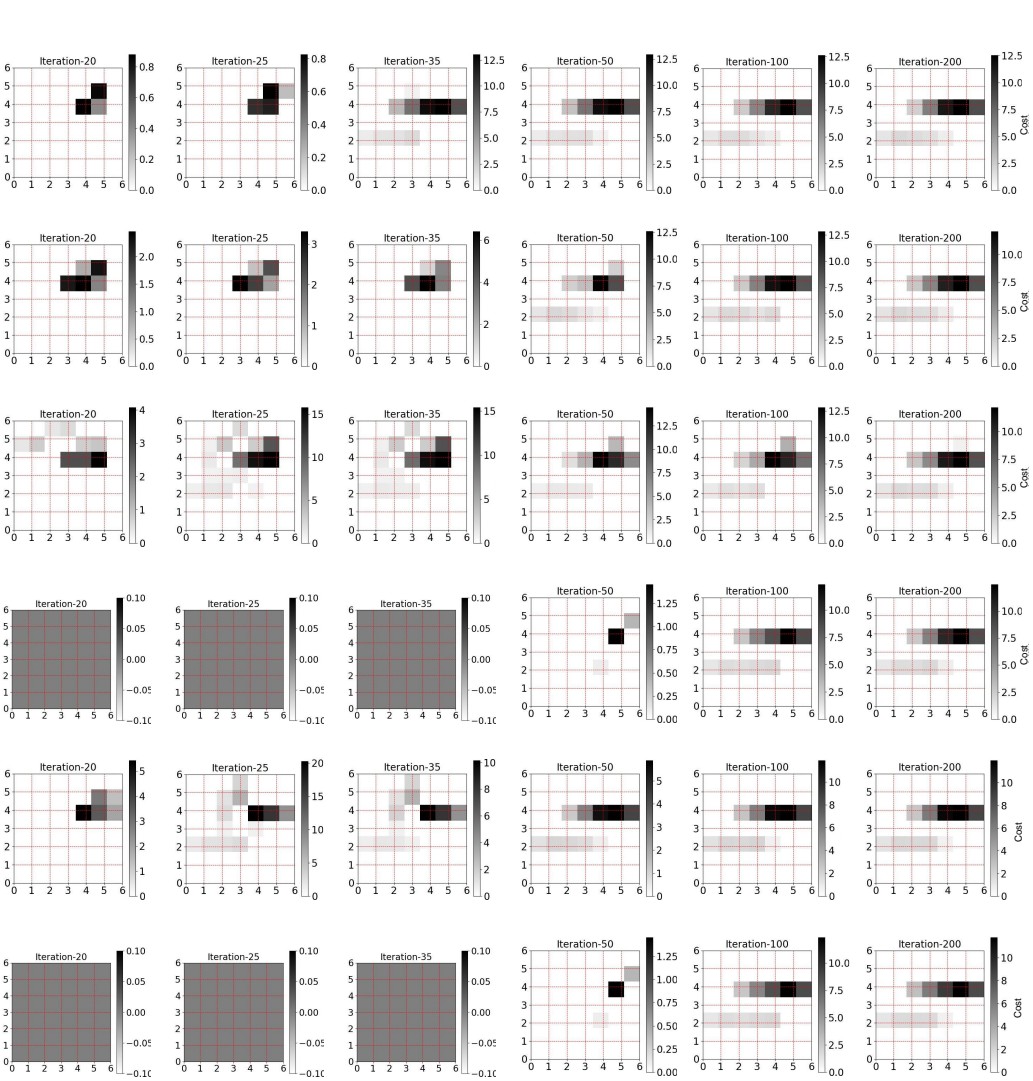

Figure 7: Constraint learning performance of six exploration strategies for ICRL in Gridworld-1. PCSE (1st row), BEAR strategy (2nd row), $\epsilon$-greedy exploration strategy (3rd row), Maximum-entropy exploration strategy (4th row), Random exploration strategy (5th row), Upper confidence bound (UCB) exploration strategy (bottom row).

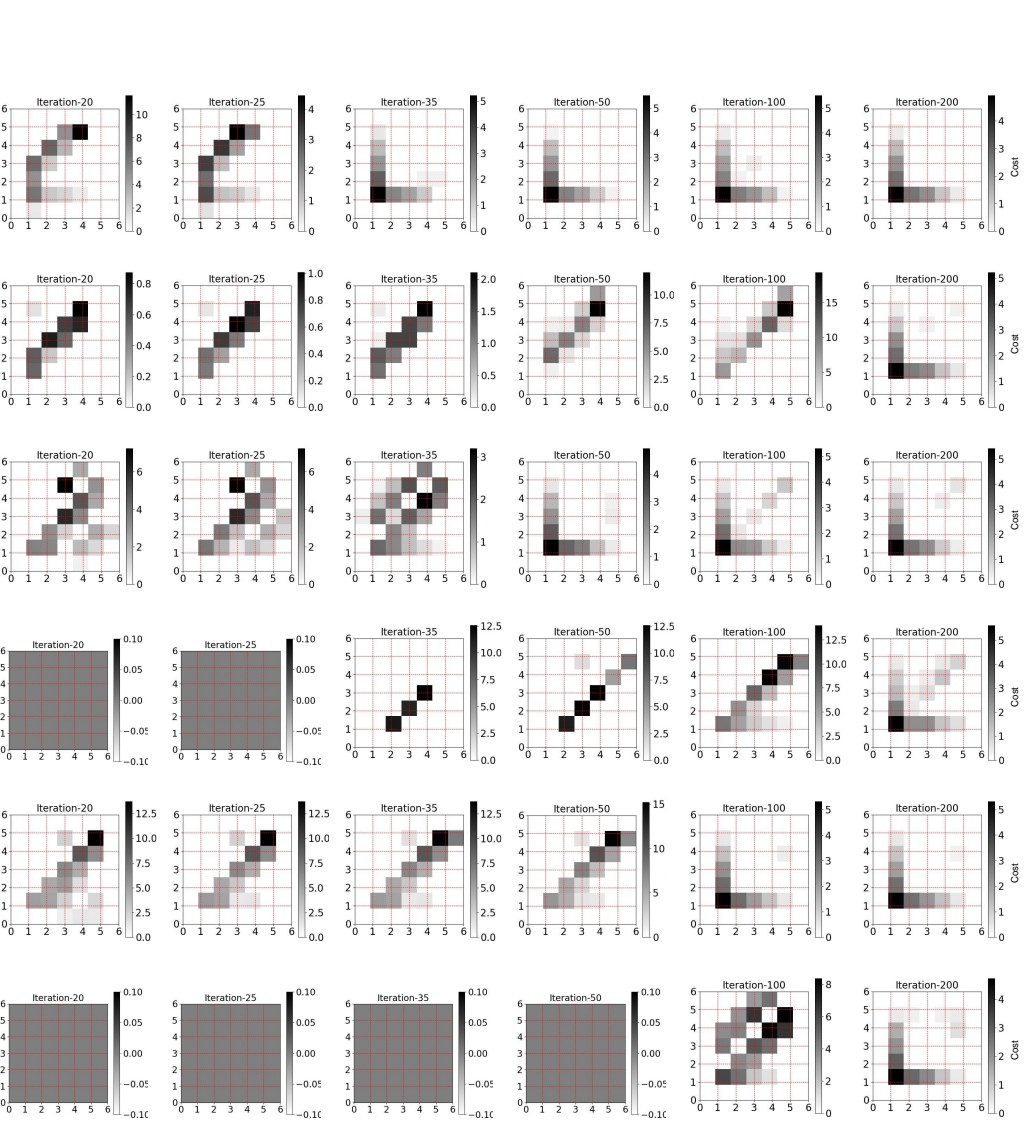

Figure 8: Constraint learning performance of six exploration strategies for ICRL in Gridworld-2. PCSE (1st row), BEAR strategy (2nd row), $\epsilon$-greedy exploration strategy (3rd row), Maximum-entropy exploration strategy (4th row), Random exploration strategy (5th row), Upper confidence bound (UCB) exploration strategy (bottom row).

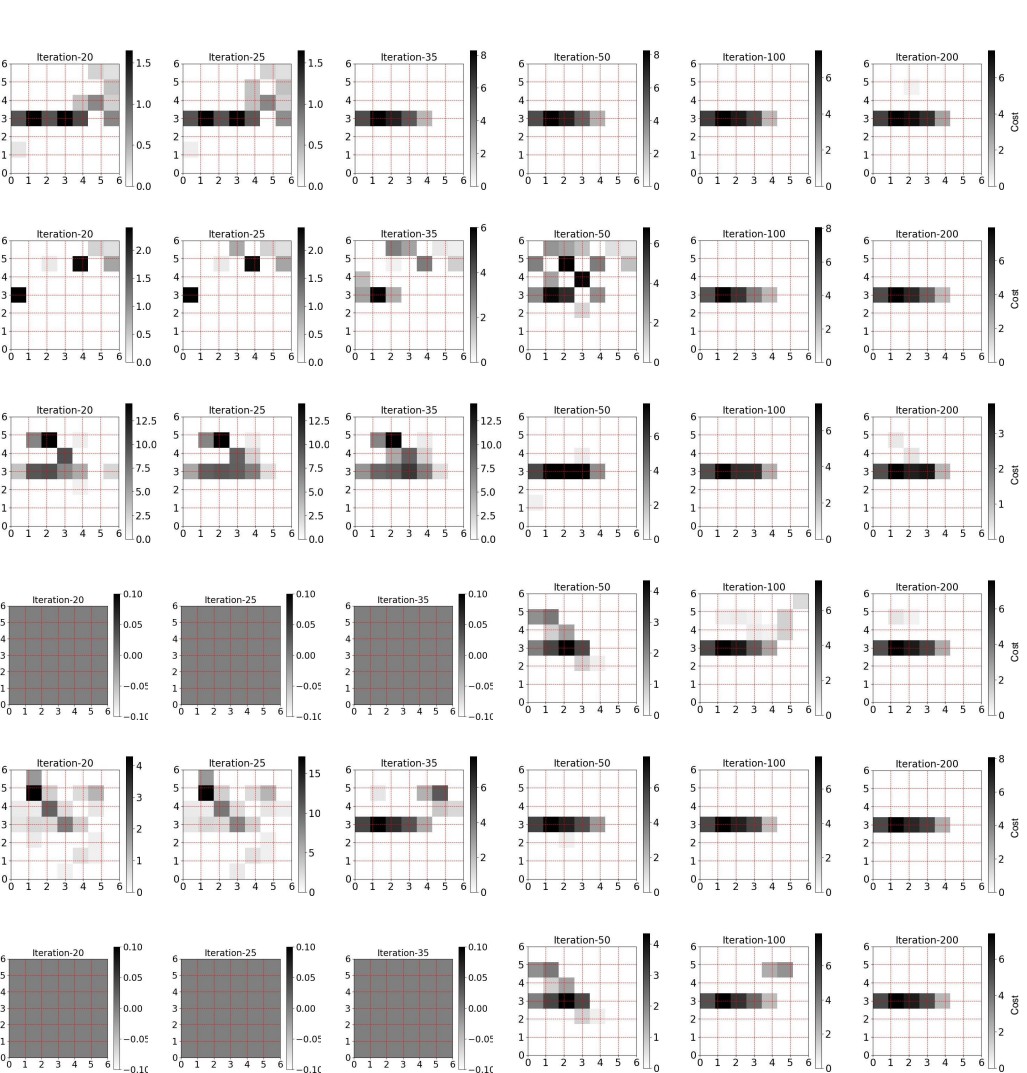

Figure 9: Constraint learning performance of six exploration strategies for ICRL in Gridworld-3. PCSE (1st row), BEAR strategy (2nd row), $\epsilon$-greedy exploration strategy (3rd row), Maximum-entropy exploration strategy (4th row), Random exploration strategy (5th row), Upper confidence bound (UCB) exploration strategy (bottom row).

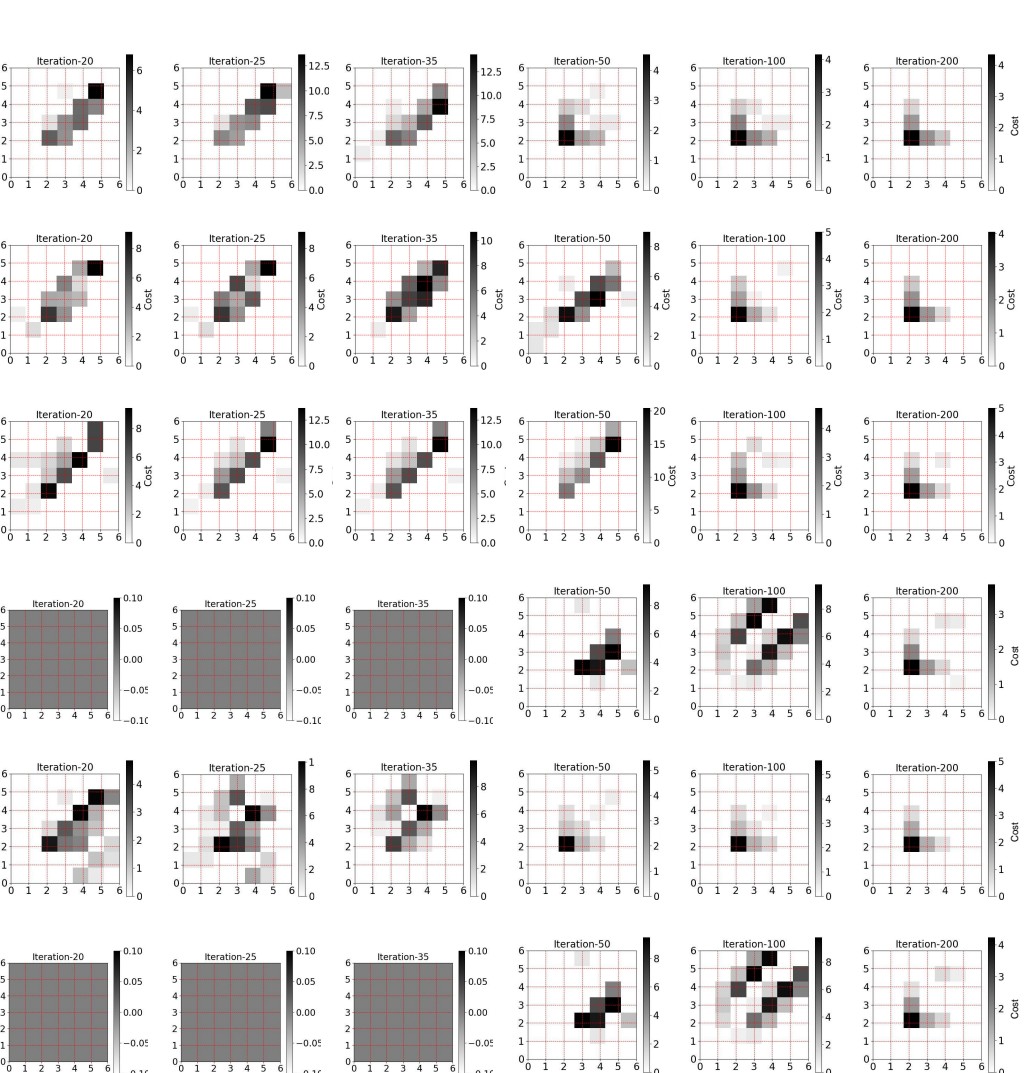

Figure 10: Constraint learning performance of six exploration strategies for ICRL in Gridworld-4. PCSE (1st row), BEAR strategy (2nd row), $\epsilon$-greedy exploration strategy (3rd row), Maximum-entropy exploration strategy (4th row), Random exploration strategy (5th row), Upper confidence bound (UCB) exploration strategy (bottom row).

## F    DISCUSSION ON SCALING TO PRACTICAL ENVIRONMENTS

Sample complexity analysis has primarily focused on discrete state-action spaces (Agarwal et al., 2019). Existing algorithms for learning feasible sets (Metelli et al., 2023; Zhao et al., 2023; Lazzati et al., 2024a) struggle to scale effectively to problems with large or continuous state spaces. This limitation arises because their sample complexity depends directly on the size of the state space, and real-world problems frequently involve large or continuous spaces. Scaling feasible set learning to practical problems with large state spaces remains a pressing challenge in the field (Lazzati et al., 2024b). One key difficulty is the estimation of the ground-truth expert policy, which is hard to obtain in an online setting. A potential solution involves extracting the expert policy from offline datasets of expert demonstrations. However, these datasets often contain a mix of optimal and sub-optimal demonstrations, leading to sub-optimal expert policies. Addressing this issue could involve: 1) treating the dataset as noisy and applying robust learning algorithms designed to handle noisy demonstrations, or 2) combining offline demonstrations with online fine-tuning, where feasible, to refine the learned policy. Finally, the scalability of learning in continuous spaces is frequently hindered by the curse of dimensionality. Dimensionality reduction techniques can mitigate this challenge by simplifying state and action representations while retaining the features essential for effective policy learning.

To enable our complexity analyses scalable to practical environments, linear Markov Decision Processes (MDPs) (Jin et al., 2020; Yang & Wang, 2019) offer a straightforward yet robust framework by assuming that the reward function and transition dynamics can be represented as linear combinations of predefined features. This assumption allows for theoretical exploration of sample complexity. In future work, we plan to leverage the Linear MDP framework and its extensions (Jin et al., 2021; Wang et al., 2020; Du et al., 2021) as a foundation to design scalable methods for inferring feasible cost sets within the ICRL framework.

