# OpenReview forum: "Strategic Exploration for Inverse Constraint Inference with Efficiency Guarantee"
_ICLR.cc/2025/Conference — Submitted to ICLR 2025_

### Official Review · Reviewer_Dxj7 · 2024-10-17

**Soundness:** 2
**Presentation:** 1
**Contribution:** 2
**Rating:** 3
**Confidence:** 5

**Summary:**

This paper analyses the problem of learning the constraints underlying some expert demonstrations in a provably efficient manner. Given that multiple constraint functions are compatible with the observed behavior, the learning target is the set of cost functions compatible with them.

**Strengths:**

The problem setting of designing provably-efficient algorithms for ICRL is really interesting in my opinion.

**Weaknesses:**

The paper is based on Definition 3.1, which defines the notion of feasible cost set. However, because of the typos and of the absence of explanation provided, it is impossible to understand what is the feasible cost set in a formal manner. Without this formal definition, the following results, and in particular Lemma 4.3, cannot be understood. Since all the theorems proved are based on Lemma 4.3, then it is no possible to understand whether the results are correct or not.

TYPOS:
- 103: what is a space?
- 107: there is an additional dot . not needed
- 137: $r$ should not be present in the tuple
- 138: no need to write "CMDP without knowing the cost", because this notation has already been defined earlier
- 139: the cost should be defined as bounded
- 139: simbol $\Pi^*$ never defined
- 141,143: bad definition of set
- ...: definitely too many typos. Have the authors read the paper after having written it? Why me and the other reviewers have to waste time in reading your paper if not even you have read it?

**Questions:**

None

---

> ### Author Response · Authors · 2024-11-25
> **Author Response to Reviewer Dxj7**
>
> Dear Reviewer Dxj7:
>
> It is a pity that you did not find the paper valuable. However, we hope that the positive feedback from other reviewers indicates that the work contains content that may be informative and beneficial to others.
>
> We have made a list to respond to your comments. Specifically, we find comments 1,3,4,5,6 are not typos at all. As for comment 2, thanks for your correction, we have deleted the dot. As for comment 7, we find 141 not a typo and we have polished 143 in the revised manuscript.
>
> We are more than willing to engage in further discussions to refine our work.
>
> | **No.** | **Comment**  | **Response** |
> |----------|----------|----------|
> | 1    | 103: what is a space?     | Not a typo. Vector space, such as $\mathbb{R}$.   |
> | 2    | 107: there is an additional dot . not needed     | Minor typo corrected in revision.     |
> | 3    | 137: $r$ should not be present in the tuple     | Not a typo. ICRL utilizes known reward signals. Refer to existing prior works [1-6].     |
> | 4    | 138: no need to write "CMDP without knowing the cost", because this notation has already been defined earlier     | Not a typo. Simply want to recall the notation.     |
> | 5    | 139: the cost should be defined as bounded     | Not a typo. Already defined as bounded in line 114.     |
> | 6    | 139: simbol $\Pi^*$ never defined     | Not a typo. $\Pi^*$ is a widely used and standard notation of the set of optimal policies.   |
> | 7    | 141,143: bad definition of set     | Not a typo. 141. Reorganized and clarified it in lines 176-185 for revision. 143. We wonder why  $\mathcal{Q}= \\{  (s,a) \mid c(s,a) > 0  \\} $ is a bad definition of set?     |
>
>
>
> References
>
> [1] Malik, S., Anwar, U., Aghasi, A., and Ahmed, A. Inverse constrained reinforcement learning. In International Conference on Machine Learning (ICML), pp. 7390–7399, 2021.
>
> [2] Scobee, D. R. R. and Sastry, S. S. Maximum likelihood constraint inference for inverse reinforcement learning. In International Conference on Learning Representations (ICLR), 2020.
>
> [3] Liu, G., Luo, Y., Gaurav, A., Rezaee, K., and Poupart, P. Benchmarking constraint inference in inverse reinforcement learning. In International Conference on Learning Representations (ICLR), 2023.
>
> [4] Gaurav, A., Rezaee, K., Liu, G., and Poupart, P. Learning soft constraints from constrained expert demonstrations. In International Conference on Learning Representations (ICLR), 2023.
>
> [5] Papadimitriou, D., Anwar, U. and Brown, D. S. Bayesian methods for constraint inference in reinforcement learning. Transactions on Machine Learning Research (TMLR), 2023.
>
> [6] Liu, G., Xu, S., Liu, S., Gaurav, A., Subramanian, S. G., \& Poupart, P. (2024). A Comprehensive Survey on Inverse Constrained Reinforcement Learning: Definitions, Progress and Challenges. arXiv preprint arXiv:2409.07569.

---

> ### Comment · Reviewer_Dxj7 · 2024-11-25
>
> Dear authors, as I said in the review, I think that the topic faced by the paper is interesting, and I would really appreciate if someone published a paper containing provably-efficient algorithms for ICRL. However, I believe that **this paper, as is written at the time of submission, must be rejected**. The reason is the following: the paper is a theory paper, thus the theoretical analysis of the algorithms plays a great role. To do the proofs, you need a precise notation, but if the notation is very imprecise or even missing, then no one can verify nor reproduce your proofs. During my review, I tried to read the proofs, but, as explained in the review, the (very) bad notation does not allow me to understand them. If I cannot understand them, then I cannot verify if the results are correct, and as such the paper must be rejected. As I see from the confidence of other reviewers (small than 5), I am the only Reviewer who read the proofs, thus probably I am the only one who noticed this issue. I strongly suggest the authors to improve the notation and the presentation of the paper in the future.
>
> Anyway, I would like to say that the answer of the Authors is very rude, and this is an unacceptable behavior. **All the typos I have mentioned are typos**, and the fact that you deny it is very rude. In the original version:
> 1. $\mathcal{Y}$ is a set, and not a vector space. A vector space is defined by a set and an operation and a scalar field, none of which is present. Moreover, $\mathbb{R}$ is commonly considered a scalar field, much different from a vector space.
> 2. Minor typo, but typo.
> 3. This is a typo. I know that ICRL utilizes a known reward signal, but your notation is wrong. You define $\mathcal{M}$ as already containing the reward $r$, thus when you define $\mathfrak{B}=(\mathcal{M},\pi^E,r)$, you are using two reward functions. Is this what you want? Definitely not.
> 4. As I said, there is no need, because already defined above.
> 5. The definition as bounded in line 114 refers to a different cost. At 139 you define a new function $c\in\mathbb{R}^{\mathcal{S}\times\mathcal{A}}$ which is unbounded by definition. You should explicitly write it as bounded.
> 6. Symbol $\Pi^*$ is definitely not standard. In a paper, you should introduce all these symbols to allow the readers understand the content.
> 7. Line 141 *is* a typo, because set $\mathcal{Q}$ does not seem to depend on the cost $c$, and you should add the dependence. But the thing that makes me think the most is that you say to me that Line 143 is not a typo, but you say to Reviewer cgQr that it indeed is.
>
> In conclusion, I will keep my score and confidence, because **I strongly believe the paper should be rejected**. In addition, **I will tell to the Area Chair your dishonest behavior in trying to mislead me**. I believe that some measures should be taken.

---

> > ### Comment · Reviewer_Dxj7 · 2024-11-27
> >
> > Dear Authors, even though you have been rude with me, I have read the revised version of the paper. Thanks to the correct definition of feasible cost set, I have been able to read the proofs and check their validity. In particular, I find the definition of  feasible cost set interesting. I would like to increase the score to 5, but there are some issues that I would like you to correct:
> > - All the typos I told you in the previous comment (some of them you already fixed).
> > - In Lemma 4.5 you cannot use value $\zeta=0$ with a positive $A>0$, so you should fix it.
> > - In Lemma 4.6 you should check that $\widehat{c}$ for those choices of $V,A$ might lie outside interval $[0,C_{\max}]$, and thus it might not be bounded. For this reason, when you apply Lemma 4.6 to find the cost $c$ in the feasible set close to each $\widehat{c}$ (see Lemma 5.1 to bound the *accuracy*, as you call it in Definition 4.9) you make a choice of cost outside the set, and this is wrong. Note that this error was present also in "Provably Efficient Learning of Transferable Rewards" of Metelli et al., and it was fixed by "Towards Theoretical Understanding of Inverse Reinforcement Learning" of Metelli et al. through the choice of *normalized* $V,A$ (see their Lemma B.1 and Theorem 3.2).
> > - There are so many other typos and things that you could present better, like: Line 179 you should say explicitly what is $\pi^E$ (optimal in CMDP), and not in the text, Line 180 action $a'$ is useless, Line 196 you could simplify notation $\mathcal{C}\_{\mathfrak{B}}=\text{arg}\min\_{c:...}|...|$, and so on.
> > - You should improve the presentation.
> >
> > If you fix these things, I will be more than happy to raise my score to 5. In the meanwhile, I give you 3.

---

> > > ### Author Response · Authors · 2024-11-28
> > > **Author Response to Reviewer Dxj7**
> > >
> > > Thank you once again for your valuable comments and for pointing out the aspects of the paper that require careful refinement.
> > >
> > > - Thank you for highlighting these points. We have addressed the issues you mentioned in your previous comment:
> > >   - 1. We have modified this in line 104
> > >   - 2. The additional dot has been deleted.
> > >   - 3. $r$ is deleted from $\mathfrak{P}$ in line 185 and the remainder of the paper.
> > >   - 4. We have deleted it from line 186 and we have defined it in line 113.
> > >   - 5. We have modified $c$ to be bounded in line 186 and the remainder of the paper.
> > >   - 6. We have explicitly defined $\Pi^*$ in line 114.
> > >   - 7. We have reorganized this in lines 181-190.
> > >
> > > - Thank you for this comment. We have corrected the definition of $\zeta$ in Lemma 4.5.
> > >
> > > - We appreciate you pointing this out and providing the relevant reference. We have modified Lemma 4.6 to make sure $\widehat{c}$ also lies in the bounded region $\\|\widehat{c}\\|_ \infty\leq\mathcal{C}_ {\max}$. We shall further prove that $\widehat{c}\geq0$ which has not been done yet due to limited time.
> > >
> > > - Thank you for your insightful comments.
> > >   - We have explicitly defined $\pi^E$ in line 115.
> > >   - We have deleted $a^\prime$ in Lemma 4.1.
> > >   - We have improved the presentation of Definition 4.2.
> > >
> > > - Thank you for your helpful advice. We have made efforts to improve the presentation of the paper. A revised manuscript has been uploaded.
> > >
> > > We sincerely appreciate your constructive feedback and your generosity in helping us refine the paper. Thank you very much.

---

> > ### Author Response · Authors · 2024-11-28
> > **Author Response to Reviewer Dxj7**
> >
> > Dear Reviewer Dxj7,
> >
> > We sincerely appreciate the time and effort you have dedicated to reviewing our paper, and we would like to address any potential misunderstandings that may have arisen during the process. Please rest assured that we have no intention of being disrespectful in any way.
> >
> > When we mentioned that a particular point was not a typo, our aim was just to clarify our understanding of whether the content in question was indeed a typo. If we believed it was not, we provided our reasoning, which we acknowledge could have been incorrect. We did not intend any further implications, such as appearing rude or misleading.
> >
> > Once again, thank you for your constructive feedback and valuable assistance in improving the quality of our paper.
> >
> > Also, we wish you a very happy Thanksgiving!

---

> ### Comment · Reviewer_Dxj7 · 2024-11-28
>
> Thank you for the updates!
>
> I do not think that this approach can guarantee that $\widehat{c}>0$. Could you fix it?

---

> ### Author Response · Authors · 2024-11-29
>
> Thank you for engaging in the discussion and keep driving us forward. We can now guarantee that $\widehat{c}>0$. We provide a detailed analysis as follows.
>
> From Lemma 4.5, $\forall (s,a)\in\mathcal{S}\times\mathcal{A}$, we can express the cost functions belonging to $\mathcal{C}_ {\mathfrak{P}}$ as:
> $
>     c(s,a) = A^{r,\pi^{E}}_ {\mathcal{M}}\zeta(s,a)+(E-\gamma P_ \mathcal{T})V^{c}(s,a).
> $
> Regarding $\widehat{\pi}^E$ and $\widehat{P_\mathcal{T}}$, we can express the estimated cost function belonging to $\mathcal{C}_ {\widehat{\mathfrak{P}}}$ as:
> $
>     \widehat{c}(s,a) = A^{r,\widehat{\pi}^{E}}_ {\widehat{\mathcal{M}}}\widehat{\zeta}(s,a)+(E-\gamma \widehat{P_\mathcal{T}})\widehat{V}^{c}(s,a)
> $
>
> What we need to do first is to provide a specific choice of $\widehat{\zeta}$ and $\widehat{V}$ under which $\widehat{c}\in[0,C_{\max}]^{\mathcal{S}\times\mathcal{A}}$.
>
> We construct
>
> $
>     \widetilde{c}(s,a) = A^{r,\widehat{\pi}^{E}}_ {\widehat{\mathcal{M}}}\zeta(s,a)+(E-\gamma \widehat{P_\mathcal{T}}) V^{c}(s,a).
> $
>
> We now define the absolute difference between $\widetilde{c}(s,a)$ and $c(s,a)$ as
>
> $\chi(s,a)=|\widetilde{c}(s,a)-c(s,a)|=\gamma\left| (P_ \mathcal{T}-\widehat{P_ \mathcal{T}}){V}^{c}\right|(s,a)+\left|A^{r,\pi^{E}}_ {\mathcal{M}}-A^{r,\widehat{\pi}^{E}}_ {\widehat{\mathcal{M}}}\right|\zeta(s,a),$
>
> $\chi=\max_{(s,a)\in\mathcal{S}\times\mathcal{A}}\chi(s,a).$
>
> $\forall (s,a)\in\mathcal{S}\times\mathcal{A}$, since $c(s,a)\in[0,C_{\max}]$ and $\widetilde{c}(s,a) - c(s,a)\in[-\chi,\chi]$, we have:
>
> $
>         \widetilde{c}(s,a) = c(s,a) + (\widetilde{c}(s,a) - c(s,a))\in[-\chi,C_{\max}+\chi]
> $
>
> Therefore, there is always a state-action pair $(s^\prime,a^\prime)$ such that
>
> $
>         \min_ {(s,a)\in\mathcal{S}\times\mathcal{A}}\widetilde{c}(s,a) = \widetilde{c}(s^\prime,a^\prime)=A^{r,\widehat{\pi}^{E}}_ {\widehat{\mathcal{M}}}\zeta(s^\prime,a^\prime)+(E-\gamma P_ \mathcal{T}) V^{c}(s^\prime,a^\prime)\geq-\chi.
> $
>
> By subtracting $\widetilde{c}(s^\prime,a^\prime)$ from all $\widetilde{c}(s,a)$, we have
>
> \begin{align}
>         \bar{c}(s,a)
>         &=\widetilde{c}(s,a)-\widetilde{c}(s^\prime,a^\prime) \\\\
> &= A^{r,\widehat{\pi}^{E}}_ {\widehat{\mathcal{M}}}\zeta(s,a)+(E-\gamma \widehat{P_\mathcal{T}}) V^{c}(s,a)-A^{r,\widehat{\pi}^{E}}_  {\widehat{\mathcal{M}}}\zeta(s^\prime,a^\prime)-(E-\gamma \widehat{P_\mathcal{T}}) V^{c}(s^\prime,a^\prime)\\\\
>         &= A^{r,\widehat{\pi}^{E}}_ {\widehat{\mathcal{M}}}[\zeta(s,a)-\frac{A^{r,\widehat{\pi}^{E}}_ {\widehat{\mathcal{M}}}(s^\prime,a^\prime)}{A^{r,\widehat{\pi}^{E}}_{\widehat{\mathcal{M}}}(s,a)}\zeta(s^\prime,a^\prime)]+(E-\gamma \widehat{P _\mathcal{T}}) [V^{c}(s,a)-V^{c}(s^{\prime},a^{\prime})]\\\\
>         &\geq 0
> \end{align}
>
> Also, note that $\forall (s,a)\in\mathcal{S}\times\mathcal{A}$, we have:
>
> $
>         |\bar{c}(s,a)|
>         \leq|\widetilde{c}(s,a)-\widetilde{c}(s^\prime,a^\prime)|\leq|\widetilde{c}(s,a)|+|\widetilde{c}(s^\prime,a^\prime)|\leq C_ {\rm{max}}+2\chi
> $
>
> Hence, $\forall(s,a)\in\mathcal{S}\times\mathcal{A},\bar{c}(s,a)\in[0,C_ {\rm{max}}+2\chi]$.
>
> Because we are looking for the existence of $\widehat{c}(s,a)\in\mathcal{C}_ {\widehat{\mathfrak{P}}}$ satisfying $\widehat{c}\in[0,C_ {\max}]^{\mathcal{S}\times\mathcal{A}}$, we can now provide a specific choice of $\widehat{\zeta}$ and $\widehat{V}$ under which $\widehat{c}\in[0,C_ {\max}]^{\mathcal{S}\times\mathcal{A}}$:
>
> $
> \widehat{\zeta}(s,a)=\frac{\zeta(s,a)-\frac{A^{r,\widehat{\pi}^{E}}_ {\widehat{\mathcal{M}}}(s^\prime,a^\prime)}{A^{r,\widehat{\pi}^{E}}_ {\widehat{\mathcal{M}}}(s,a)}\zeta(s^\prime,a^\prime)}{1 + 2\chi/C_ {\max}}, \widehat{V}^c(s,a)=\frac{V^{c}(s,a)-V^c(s^\prime,a^\prime)}{1 + 2\chi/C_ {\max}}.
> $
>
> We then quantify the estimation error between $\widehat{c}(s,a)$ and $c(s,a)$.
>
> \begin{align}
>         |c(s,a)-\widehat{c}(s,a)|
>         & = \left|c(s,a)-\frac{\bar{c}(s,a)}{1 + 2\chi/C _{\max}}\right| \\\\
>         & = \frac{1}{1 + 2\chi/C _{\max}}\big[|c(s,a)-\bar{c}(s,a)|+(2\chi/C _{\max})|c(s,a)|\big] \\\\
>         & \leq \frac{1}{1 + 2\chi/C _{\max}}\big[|c(s,a)-\widetilde{c}(s,a)|+|\widetilde{c}(s,a)-\bar{c}(s,a)|+(2\chi/C _{\max})|c(s,a)|\big]\\\\
>         & \leq \frac{\chi+\chi+(2\chi/C _{\max})C _{\max}}{1 + 2\chi/C _{\max}}\\\\
>         & \leq \frac{4\chi}{2 + \chi/C _{\max}}
> \end{align}
>
> May you a happy weekend by the way!

---

> > ### Comment · Reviewer_Dxj7 · 2024-11-29
> >
> > Thank you! I guess there is a mistake in your derivation. Specifically, in the last but one passage, you bound $|\widetilde{c}(s,a)-\overline{c}(s,a)|=|\widetilde{c}(s',a')|\le X$, but actually from earlier passages we know that it should be bounded by $X+C_{\max}$, which invalidates the proof since $C_{\max}$ does not depend on samples.
> >
> > Please, just answer yes or no and be sincere. If no, please explain me why. If yes, please do not submit to me any other long attempt because I will not read it, but at most a very short fix (if it exists).
> >
> > Thank you!

---

> > > ### Author Response · Authors · 2024-11-30
> > >
> > > Thank you!
> > > Yes, it is a mistake.
> > > We can fix it by distinguishing two cases: 1) $\widetilde{c}(s^\prime,a^\prime)<0$ and 2) $\widetilde{c}(s^\prime,a^\prime)\geq0$.
> > >
> > > - In case one, $|\widetilde{c}(s^\prime,a^\prime)|\leq\chi$, so the above derivation applies.
> > > - In case two, we can directly let $\widehat{c}(s,a)=\frac{\widetilde{c}(s,a)}{1 + \chi/C_{\rm{max}}}$ such that $|c(s,a)-\widetilde{c}(s,a)|\leq\chi$ which does not depend on $C_{\max}$.
> > >
> > > Thank you!

---

> > > > ### Comment · Reviewer_Dxj7 · 2024-12-01
> > > >
> > > > Thank you, now it works.
> > > >
> > > > Dear Authors, I would like to thank you for all the efforts in updating the presentation of the paper as well as fixing some typos and the errors I spotted. I am aware that I told you that I might have increased my score to 5 if you rearranged these issues, however:
> > > > - There have been huge modifications to the paper, that require time to be adjusted accurately.
> > > > - Viewing again the revised version of the paper that you uploaded, I have found another potential issue. In definition 4.9, you do not mention which $(s,a)$ pair you mean (typo). When I have checked the proof of Theorem 5.5 to understand what is the correct definition you used, I have found out a maximum over all $(s,a)\in\mathcal{S}\times\mathcal{A}$, which can be problematic if you do not use a generative model for collecting samples, since not all states might be connected (note that this issue forced the authors of "Active Exploration for Inverse Reinforcement Learning" to change their definition from the maximum over $(s,a)\in\mathcal{S}\times\mathcal{A}$ to a new definition that keeps into account the coverage of the space, as you can see from arxiv).
> > > >
> > > > In summary, because of the many adjustments that have been applied to the paper, and because of this issue that should be checked carefully, I think that the paper requires additional efforts and time to adjust all the details and be ready for publication. For this reason, I will keep my score.

---

> > > > > ### Author Response · Authors · 2024-12-02
> > > > >
> > > > > Thank you for your invaluable support in improving our paper throughout the rebuttal and discussion phases. We will incorporate these modifications and address this potential issue, along with any other underlying concerns.

---

### Official Review · Reviewer_cgQr · 2024-10-29

**Soundness:** 2
**Presentation:** 2
**Contribution:** 2
**Rating:** 5
**Confidence:** 3

**Summary:**

This paper addresses inverse constrained reinforcement learning, a problem in which we aim to infer a cost constraint by looking at expert's behaviour only. The specific setting works as follows: We can deploy a policy in the MDP to get a rollout of state transitions and expert's actions, but we cannot see the cost. We aim to infer a set of costs compatible with the expert's actions, which are assumed to be optimal, while minimizing the samples taken from the environment. The paper proposes two algorithmic solutions for this setting, a bonus-based exploration strategy called BEAR and a further refined version called PCSE, together with the analysis of their corresponding sample complexity and a brief empirical evaluation.

**Strengths:**

- The paper tackles an interesting problem setting that may have practical upside for relevant applications;
- The paper addresses the strategic exploration problem in ICRL, which it has been previously studied in settings with known dynamics or a generative model;
- The paper provides two algorithmic solutions and corresponding sample complexity results;
- The paper includes a numerical validation, which is not that common in purely theoretical RL papers.

**Weaknesses:**

I summarize below some of my main concerns about the work. More detailed comments are below.
- The paper dives into a theoretical analysis of a rather peculiar setting without providing proper motivations;
- The paper seems to have some presentation issues: I understood (most of) the interaction protocol at the start of Section 5, but some details are still obscure, whereas the notation does not look always sharp and detailed. The sample complexity results include some terms for which the dependence with $S, A, \gamma$ is not obvious;
- The paper lacks a in-depth discussion of the results, e.g., how they relate with prior works on IRL or ICRL with generative models, the considered assumptions (especially deterministic policies), computational complexity of the algorithms, the technical novelty of the presented analysis;
- The numerical validation does not seem to be conclusive. Most of the curves are not separated with statistical significance.

**COMMENTS**

MOTIVATION. The formulation of the setting could be more clearly motivated. While it is roughly clear the kind of applications that are target, it is less clear why some choices are made.
- Is the discounted setting more interesting than finite-horizon for ICRL?
- In which kind of applications we can expect to have unconstrained access to the MDP online, even though the expert acts under constraints? Do the authors have any example of an application with cost constraints that allows for unconstrained exploration?
- Also the PAC requirement is somewhat questionable: Under approximation errors of the MDP and expert's policy we are not guaranteed that the optimal policy for the costs in the feasible set is "safe" to use in the true MDP (i.e., it would respect the true constraint). This is common in other inverse RL paper, but while some sub-optimality can be acceptable in unconstrained RL, some violations of the constraints are less acceptable in a constrained setting.

PRESENTATION. The presentation is not always sharp in the paper. I am listing below some questions, suggestions on how I think it could be improved.
- Some broader context could be added to first sentence of the abstract, e.g., that we want to optimize an objective function under cost constraint(s);
- The equation at l. 143 likely includes a typo. The Definition 3.1 would also benefit from more context and a more formal introduction to the notation (e.g., what do the value functions mean exactly?). It requires quite a lot of time to be processed;
- I could not fully comprehend Eq. 2. Are $\zeta$ and $E$ defined somewhere? If that is the case, perhaps it is worth recalling their meaning here;
- The interaction setting shall be introduced earlier than Sec. 5.1 and some aspects are still not clear then. How is the reward accessed/estimated?
- Sec. 5.5: "The above exploration strategy has limitations, as it explores to minimize uncertainty across all policies, which is not aligned with our primary focus of reducing uncertainty for potentially optimal
policies." This is not fully clear and would benefit from further explanation. I thought the goal was to achieve the PAC requirement with minimal sample complexity. In general, the description of PCSE is not easy to process.

TECHNICAL NOVELTY. BEAR looks like a rather standard bonus-based exploration approach, in which the main novelty seems to come from adapting the bonuses expression to the ICRL setting. Can the authors describe if other uncommon technical challenges arise from the specificity of the setting (especially w.r.t. prior works) and how are they addressed? I am not very familiar with the related literature in solving ICRL with a generative model, but in theoretical RL is sometimes straightforward to get a "strategic exploration" result from a "generative model" result.

DETERMINISTIC POLICY. Assuming the expert's policy to be deterministic in MDPs is reasonable, a little less in CMDP. Can the authors discuss this point? It looks like they think determinism is necessary. Can they prove that formally?

COMPARISON WITH PRIOR WORK. The paper shall discuss how the presented sample complexity results compare with prior works in IRL, reward-free exploration, and, especially, ICRL with a generative model. Is the PAC requirement significantly different from prior works? Moreover, the $\sigma$ terms in the sample complexity may have hidden dependencies in $S, A, \gamma$...

OTHER COMMENTS
- ll. 175-181. Those considerations look informal if not incorrect. One can easily imagine optimal trajectories that do not fully overlap with the expert's one in the given example, whereas not all of the sub-optimal trajectories are necessarily satisfying constraints!
- How can the C_k bonus be computed in BEAR? It seems to include the advantage, but the estimation of the reward is not mentioned anywhere;
- Are the described approaches computationally tractable?
- Another alternative yet interesting setting is the one in which the cost is also collected from the environment, but the constraint (threshold) is not known;
- Some additional related works on misspecification in IRL https://ojs.aaai.org/index.php/AAAI/article/view/26766, https://arxiv.org/pdf/2403.06854 and sample efficient IRL https://arxiv.org/pdf/2402.15392, https://arxiv.org/abs/2409.17355, https://arxiv.org/pdf/2406.03812 could also be mentioned.

**EVALUATION**

The addressed setting looks technically challenging and of practical interest. I am currently providing a slightly negative evaluation to account for my confusion over some aspects of the paper, which the authors may resolve with their response.

**Questions:**

I reported some of my questions in the comments above.

---

> ### Author Response · Authors · 2024-11-25
> **Author Response to Reviewer cgQr - (1/3)**
>
> Dear Reviewer cgQr,
>
> We sincerely appreciate your constructive feedback. In response, we have carefully revised the manuscript, highlighting all changes in orange for discrepancies. We have carefully considered your suggestions, and we hope that the following response can address your concerns:
>
> > Comment 1: Is the discounted setting more interesting than finite-horizon for ICRL?
>
> **Response 1: There are several advantages for discounted settings over finite-horizon settings.
> The discounted setting encourages reasoning over potentially infinite time horizons, which is useful for problems where long-term behavior matters. Many constrained real-world scenarios do not have natural endpoints, making the discounted setting more suitable. For example, 1) autonomous driving where autonomous vehicles operate continuously, navigating roads, interpreting traffic signals, and responding to dynamic conditions without a predetermined endpoint, 2) industrial process control
> where manufacturing plants and chemical processing facilities run continuously, requiring constant monitoring and adjustments to maintain optimal performance and product quality, 3) home service robots where robots performing repetitive household tasks—such as vacuuming, dishwashing, or lawn mowing—operate on a continuous basis to maintain cleanliness and order in the home.
> Also, discounted settings are often more analytically tractable because they lead to stationary policies and stationary value functions.
>
> ---
>
> > Comment 2: In which kind of applications we can expect to have unconstrained access to the MDP online, even though the expert acts under constraints? Do the authors have any example of an application with cost constraints that allow for unconstrained exploration?
>
> **Response 2:** We appreciate your concern regarding this important issue in ICRL. Indeed, we had in-depth discussions with our industrial partner and clarified this issue with our collaborators. We have identified two key points.
>
> First, learning constraints necessitate sub-optimal behaviors to effectively train the discriminators. Relying solely on positive, or 'correct' samples is insufficient, and sub-optimal behaviors are essential, even if they result in unsafe actions during the control phase.
>
> Second, in an industrial context, constraint violations during the learning phase have limited consequences. This is because control algorithms are primarily trained in simulated environments rather than real-world settings. While constraint violations encounter a simulation-to-reality gap, they do not cause significant real-world losses. Bridging this gap is a major focus in the field, with ongoing efforts to develop accurate world models that adapt to real-world complexities.
>
> ---
>
> > Comment 3: Also the PAC requirement is somewhat questionable: Under approximation errors of the MDP and expert's policy we are not guaranteed that the optimal policy for the costs in the feasible set is "safe" to use in the true MDP (i.e., it would respect the true constraint). This is common in other inverse RL papers, but while some sub-optimality can be acceptable in unconstrained RL, some violations of the constraints are less acceptable in a constrained setting.
>
> **Response 3:** We want to clarify two points. 1) This approximation error can be controlled by altering the significance $\delta$ and the target accuracy $\epsilon$. 2) The estimated cost function will be first tested in simulated environments rather than real-world settings, causing no significant real-world losses.
>
> ---
>
> > Comment 4: Some broader context could be added to the first sentence of the abstract, e.g., that we want to optimize an objective function under cost constraint(s);
>
> **Response 4:** Thanks for your suggestion. We have modified the first sentence as 'Optimizing objective functions under cost constraints is a fundamental problem in many real-world applications. However, the constraints are often not explicitly provided and must be inferred from the observed behavior of expert agents.'

---

> ### Author Response · Authors · 2024-11-25
> **Author Response to Reviewer cgQr - (2/3)**
>
> > Comment 5: The equation at l. 143 likely includes a typo. The Definition 3.1 would also benefit from more context and a more formal introduction to the notation (e.g., what do the value functions mean exactly?). It requires quite a lot of time to be processed;
>
> **Response 5:** 1) Yes, we have corrected this typo. It should be $\mathcal{Q}_ c=\{(s,a)|c(s,a) > 0\}$. 2) We have reorganized the content. In the revised manuscript, we first present the intuitive example of Figure 1, then introduces $\mathcal{Q}_ c=\{(s,a) \vert Q^{c,\pi^{E}}_ {\mathcal{M}}(s,a)-V^{c,\pi^{E}}_ {\mathcal{M}}(s) > 0\}$ to characterize the ICRL's choice of minimal set of cost functions, and finally formalize the ICRL problem in Definition 4.1. The cost value function under the expert policy actually represents the expert's used budget for the current state. If there exists an action yielding greater rewards than the expert action, the expert policy’s cumulative costs must reach the threshold (use up all the budget) so that this action is banned from feasible regions. We have put Lemma 4.1 stating the above idea before defining the ICRL problem for better clarification in the revised manuscript.
>
> ---
>
> > Comment 6: I could not fully comprehend Eq. 2. Are $\zeta$ and $E$ defined somewhere? If that is the case, perhaps it is worth recalling their meaning here;
>
> **Response 6:** Sorry for this confusion. $\zeta\in\mathbb{R}_{\geq 0}^{\mathcal{S}\times\mathcal{A}}$ and $E$ is the expansion operator satisfying $( Ef) ( s, a) = f( s)$. We previously defined them in the notation part. We have now added contexts to recall their meanings in the revised version.
>
> ---
>
> > Comment 7: The interaction setting shall be introduced earlier than Sec. 5.1 and some aspects are still not clear then. How is the reward accessed/estimated?
>
> **Response 7:** Reward is given in the setting of ICRL problems, as stated in Def. 4.2 in $\mathfrak{P}$. Most existing ICRL literature [1,2,3,4,5,6] generally assumes the availability of a nominal reward function.
>
> ---
>
> > Comment 8: Sec. 5.5: "The above exploration strategy has limitations, as it explores to minimize uncertainty across all policies, which is not aligned with our primary focus of reducing uncertainty for potentially optimal policies." This is not fully clear and would benefit from further explanation. I thought the goal was to achieve the PAC requirement with minimal sample complexity. In general, the description of PCSE is not easy to process.
>
> **Response 8:** Eq. (7) states that the exploration algorithm converges (satisfies Definition 4.9) when either (i) or (ii) is satisfied. If we aim to converge the exploration algorithm by (i) via BEAR, we minimize uncertainty across all policies, because the cumulative term takes its upper bound as $\frac{1}{1-\gamma}$. (ii) is a more relaxed version than (i). To converge the exploration algorithm by (ii), we can identify which $\pi$ leads to the maximum LHS of (ii).
>
> ---
>
> > Comment 9: TECHNICAL NOVELTY. BEAR looks like a rather standard bonus-based exploration approach, in which the main novelty seems to come from adapting the bonus expression to the ICRL setting. Can the authors describe if other uncommon technical challenges arise from the specificity of the setting (especially w.r.t. prior works) and how are they addressed? I am not very familiar with the related literature in solving ICRL with a generative model, but in theoretical RL is sometimes straightforward to get a "strategic exploration" result from a "generative model" result.
>
> **Response 9:** We discussed ICRL from a generative model in Appendix C.8. The key takeaway for BEAR is that it does not rely on a generative model for collecting samples. Instead, it determines which states require more frequent visits and how to traverse to them, starting from
> the initial state distribution.
>
> ---
>
> > Comment 10: DETERMINISTIC POLICY. Assuming the expert's policy to be deterministic in MDPs is reasonable, a little less in CMDP. Can the authors discuss this point? It looks like they think determinism is necessary. Can they prove that formally?
>
> **Response 10:** Yes. In Assumption 4.3, we assume the expert policy is deterministic in terms of soft constraints (the expert policy can be stochastic in terms of hard constraints).  The rationale is that when the expert policy $\pi^E$ is stochastic at state $s$, we only know $\mathbb{E}_ {a^\prime\sim\pi^E}[Q^{c,\pi^E}_ {\mathcal{M}\cup c}(s,a^\prime)]=V^{c,\pi^E}_ {\mathcal{M}\cup c}(s)\geq 0$. In order to determine the value of $Q^{c,\pi^E}_ {\mathcal{M}\cup c}(s,a)$ for a specific expert action $a$ (so that the feasible cost set can be defined), additional information is required, such as whether the budget is used up and reward signals of other expert actions.

---

> ### Author Response · Authors · 2024-11-25
> **Author Response to Reviewer cgQr - (3/3)**
>
> > Comment 11: COMPARISON WITH PRIOR WORK. The paper shall discuss how the presented sample complexity results compare with prior works in IRL, reward-free exploration, and, especially, ICRL with a generative model. Is the PAC requirement significantly different from prior works? Moreover, the $\sigma$ terms in the sample complexity may have hidden dependencies in $\mathcal{S}$, $\mathcal{A}$, $\gamma$...
>
> **Response 11:** The PAC requirement considers CMDP settings while prior works consider regular MDP settings. $\sigma$ only relies on $\gamma$, others are only constants once the environment is given.
>
> ---
>
> > Comment 12: ll. 175-181. Those considerations look informal if not incorrect. One can easily imagine optimal trajectories that do not fully overlap with the expert's one in the given example, whereas not all of the sub-optimal trajectories are necessarily satisfying constraints!
>
> **Response 12:** We agree with the reviewer. We have removed these informal contexts. We originally wanted to describe such a scenario, where the expert policy is a distribution on all optimal policies, i.e., the expert policy has a chance to be any optimal policy. In this sense, if there is a policy that does not match any policy the expert may choose, it must be constraint-violating. However, this adds a very strong assumption on the expert policy, so we choose to remove it to avoid any confusion.
>
> ---
>
> > Comment 13: How can the $C_k$ bonus be computed in BEAR? It seems to include the advantage, but the estimation of the reward is not mentioned anywhere;
>
> **Response 13:** Reward is a known prior in the setting of ICRL problems, as stated in Def. 4.2 in $\mathfrak{P}$. Most existing ICRL literature [1,2,3,4,5,6] generally assumes the availability of a nominal reward function. Given $r$, the advantage function $\min^+\big|A^{r,\pi^{E}}_{\mathcal{M}}\big|$ is actually a constant number related to the ground-truth environment.
>
> ---
>
> > Comment 14: Are the described approaches computationally tractable?
>
> **Response 14:** Yes, they are. Sample complexity for BEAR and PCSE are derived.
>
> ---
>
> > Comment 15: Another alternative yet interesting setting is the one in which the cost is also collected from the environment, but the constraint (threshold) is not known;
>
> **Response 15:** If the expert policy is known, the threshold can be estimated by collecting multiple rollouts of the expert policy.
>
> ---
>
> > Comment 16: Some additional related works on misspecification in IRL and sample efficient IRL could also be mentioned.
>
> **Response 16:** Thanks for your suggestion. We have discussed related works on sample efficient IRL in Main text Sec 2 and related works on misspecification in IRL in Appendix B due to the page limit.
>
> ---
>
> References
>
> [1] Malik, S., Anwar, U., Aghasi, A., and Ahmed, A. Inverse constrained reinforcement learning. In International Conference on Machine Learning (ICML), pp. 7390–7399, 2021.
>
> [2] Scobee, D. R. R. and Sastry, S. S. Maximum likelihood constraint inference for inverse reinforcement learning. In International Conference on Learning Representations (ICLR), 2020.
>
> [3] Liu, G., Luo, Y., Gaurav, A., Rezaee, K., and Poupart, P. Benchmarking constraint inference in inverse reinforcement learning. In International Conference on Learning Representations (ICLR), 2023.
>
> [4] Gaurav, A., Rezaee, K., Liu, G., and Poupart, P. Learning soft constraints from constrained expert demonstrations. In International Conference on Learning Representations (ICLR), 2023.
>
> [5] Qiao G., Liu G., Poupart P., and Xu Z. Multi-modal inverse constrained reinforcement learning from a mixture of demonstrations. In Advances in Neural Information Processing Systems (NeurIPS), 2023.
>
> [6] Papadimitriou, D., Anwar, U. and Brown, D. S. Bayesian methods for constraint inference in reinforcement learning. Transactions on Machine Learning Research (TMLR), 2023.

---

> > ### Comment · Reviewer_cgQr · 2024-11-28
> >
> > Dear Authors,
> >
> > Thanks for addressing my comments in your thorough responses and for integrating reviewers' suggestions in the updated manuscript.
> >
> > I have some follow-up comments to share. If the authors could give their perspective on them too, that would be very helpful for my evaluation.
> >
> > **2) Unconstrained exploration.** It is totally understandable that sub-optimal actions must be taken in order to learn the constraints. My concern was mostly related to when this is acceptable in practice: I think it would go a long way to mention in the paper one or two use cases like those in your responses (also at a high level if the problem is to keep the identity of the industrial partner confidential).
> >
> > This problem of violating constraints during exploration is common in the constrained RL literature too, where people came up with algorithms to minimize violations while learning. If that could be done in ICRL too, I think it would make for a much greater practical upside.
> >
> > **3) Approximation error.** Of course the error can be controlled with $\epsilon$ and $\delta$, what I was thinking of is whether we can be conservative and make sure that the approximation will fall on the "safe side" w.h.p. (i.e., with limited samples constraints will be stricter than needed, but never more forgiving).
> >
> > **9) Technical novelty.** Let me clarify my comment here. I understand that addressing the problem with strategic exploration is harder than with a generative model. What I would like to know is whether going from generative model results to strategic exploration results in ICRL is technically different than RL, because there is large body of literature on translating generative -> strategic in the latter setting.
> >
> > **10) Deterministic policies.** I see, but it is still a little underwhelming. Perhaps the deterministic policy assumption could be strengthen with a formal impossibility result (when the policy is stochastic) and further highlighted in the abstract/introduction.
> >
> > **11) Comparison with prior work.** The ICRL setting is certainly peculiar, but I think some of the ideas could still be related to previous IRL works. Also, a comparison of the respective sample complexities would clarify to which extent CMDPs are harder than MDPs in the inverse problem.
> >
> > **14) Computational tractability.** What I meant is whether the algorithms are *computationally* efficient, not *sample* efficient. Do the presented algorithms run in polynomial time? Sorry if the computational complexity is already reported, I did not remember seeing it while going through the paper.
> >
> > **15) MINOR: Inferring the threshold.** Not sure I understand this: By taking rollouts from the expert we may estimate the cost incurred by the expert's policy, but this does not necessarily coincide with the threshold, right?

---

> ### Author Response · Authors · 2024-11-30
>
> Thank you for sharing these follow-up comments with us.
>
> *Comment on 2) Unconstrained exploration.*
>
> **Response:** The high sample complexity and challenges in exploration demonstrate that reinforcement learning (RL) algorithms struggle to efficiently acquire data online, particularly in safety-critical tasks. A common approach in this context is to conduct exploration and policy learning within simulators. In the field of embodied AI, robotic policies are typically trained in simulators that replicate real-world physical environments before being deployed on actual robots [1,2,3]. In this sense, sub-optimal actions that are potentially constraint-violating are acceptable because there's hardly any cost in a simulator such that the robot can do such unsafe actions repeatedly to acquire efficient constraint information.
>
> We recognize that minimizing violations during learning is an interesting direction to explore. However, if our goal is to make ICRL more practical, we believe the simulator-based approach is more effective than learning restrictively in real-world settings.
>
> References
>
> [1] DrEureka: Language Model Guided Sim-To-Real Transfer. https://eureka-research.github.io/dr-eureka/
>
> [2] RoboGSim: A Real2Sim2Real Robotic Gaussian Splatting Simulator. https://robogsim.github.io/
>
> [3] Bi-directional Domain Adaptation for Sim2Real Transfer of Embodied Navigation Agents. https://arxiv.org/pdf/2011.12421
>
> ---
>
> *Comments on 3) Approximation error.*
>
> **Response:** We agree with the reviewer that a more conservative exploration strategy in learning the constraint is helpful, but since there exists a simulator-based approach where constraint violation is allowed, it is better to explore more sufficiently without a safety concern.
>
> ---
>
> *Comments on 9) Technical novelty.*
>
> **Response:** For the BEAR algorithm, the reward is the upper bound for estimation error of costs, so the technical difficulty of it is the same as RL. The PCSE algorithm additionally restricts the policy with reward and cost limitations, so it basically is a constrained RL problem.
>
> ---
>
> *Comments on 10) Deterministic policies.*
>
> **Response:** We agree that a formal impossibility result provides a better clarification on deterministic policies in soft constraint scenarios. Here, we can further offer an intuitive example. Suppose the agent starts at state $s_0$ and can navigate to two other states $s_1$ and $s_2$. State $s_1$ has a reward of $20$ and a cost of $10$ while state $s_2$ has a reward of $4$ and a cost of $4$. If the threshold is $7$, the expert policy should be going to $s_1$ and $s_2$ with equal possibility (0.5). However, in this sense, the expert policy achieves a cost of $7$, but we do not know the exact cost of going to $s_1$ and $s_2$. More specifically, we only know $0.5c(s_1)+0.5c(s_2)=7$, but we can not solve the equation to obtain $c(s_1)$ and $c(s_2)$. We need additional information.
>
> ---
>
> *Comments on 11) Comparison with prior work.*
>
> **Response:** We agree with the reviewer on this point. To properly compare the sample complexity of IRL and ICRL, we must first establish a fair basis for comparison. We also find it intriguing to apply IRL to the CMDP problem, where the agent learns a reward correction term. This correction term, when combined with the original reward, can induce a safe-optimal policy within the CMDP framework.
>
> For an intuitive comparison between IRL and ICRL, the additional complexity of ICRL comes from the fact that we utilize the advantage function to learn a minimal cost set so that only necessary cost functions are introduced. Intuitively, banning all state-action pairs ensures the optimality of the expert, but it harms the generalizability of the inferred cost functions (transferring to an environment with different transition or reward signals).
>
> ---
>
> *Comments on 14) Computational tractability.*
>
> **Response:** Sorry, we misunderstood your point. We did not study the time complexity of ICRL algorithm in this paper. We totally agree with the reviewer that it is important for an algorithm to be computationally efficient, especially since we hope the algorithm has potential practical applications.
>
> Could you offer some references where time complexity is studied besides sample complexity in the literature? Thank you.
>
> ---
>
> *Comments on 15) MINOR: Inferring the threshold.*
>
> **Response:** There is an explicit relationship between the cost incurred by the expert policy and the threshold. In Lemma 4.1, we prove that if an action yields greater rewards than the expert action, the expert policy’s cumulative costs must reach the threshold. Hence, the threshold can be estimated by the cost incurred by the expert policy.
>
> Thank you again for engaging in the discussion and provide further feedbacks!

---

> > ### Comment · Reviewer_cgQr · 2024-11-30
> >
> > Thanks a lot for the further clarifications. My current evaluation is currently borderline as I am trying to weigh the strengths and weaknesses of the manuscript. I will discuss those with other reviewers before taking a final stance. Below I report my final comments on this thread.
> >
> > **Unconstrained exploration.** I understand your point on using simulators for the constraint inference. It makes a lot of sense, I think it shall be highlighted in the paper and the abstract especially. The question of motivation remains: If we are always working with simulators, is strategic exploration that important? I guess not all of the simulators are resettable like a generative model, but the set of applications for which the proposed solutions are necessary may be tinier than one would expect.
> >
> > **Approximation error.** Not sure my comment was clear enough here. What I meant is that it might be desirable to have in the approximate feasible cost set only costs that are at least as stringent than those of the true feasible cost set. Otherwise, if one picks a cost function at random in the learned set and optimize the resulting CMDP, there is a (small, depending on $\epsilon, \delta$) chance to obtain an "unsafe" policy (to be deployed in the true system).
> >
> > **Computational tractability.** Computational tractability is often at least commented in theoretical RL papers (one random example "Efficient Model-Free Exploration in Low-Rank MDPs" by Mhammedi et al., 2023). In principle, it is important to understand whether the best sample complexity rates can be obtained with "implementable" algorithms or not. In the literature of inverse RL, the paper "Offline Inverse RL: New Solution Concepts and Provably Efficient Algorithms" by Lazzati et al., 2024, provides tractable implementations for some of their algorithms.
> >
> > **Inferring the threshold.** This sounds rather counterintuitive, but again, it is not worth quibbling on this point as it is not included in the paper.
> >
> > Best wishes,
> >
> > Reviewer cgQr

---

> > > ### Author Response · Authors · 2024-11-30
> > >
> > > Thanks for your feedback and providing additional references.

---

### Official Review · Reviewer_UphH · 2024-11-02

**Soundness:** 3
**Presentation:** 2
**Contribution:** 2
**Rating:** 6
**Confidence:** 3

**Summary:**

The paper introduce two exploratory algorithms, BEAR and PCSE,  to solve Inverse Constrained RL problem setting, where constraint signals are learnt from expert policies. The approach recovers a set of feasible constraints that align with expert preferences. A theoretical analyses of these algorithms is provided including sample complexity bounds.

**Strengths:**

Concrete theoretical analysis that is well detailed, along with good empirical results for the provided environments.

**Weaknesses:**

Although some experiments were performed on a continuous setting, it is unknown (not even addressed) how the algorithms scales in both continuous state and action spaces. The current test case is a simple environment with discrete actions.

**Questions:**

1) Please discuss challenges that you anticipate in scaling your approach to environments were both state and action spaces are continuous, and potential solutions to the challenges.
2) Please include a discussion of how you might adapt your theoretical analysis and sample complexity bounds for high-dimensional continuous spaces in your future work.

---

> ### Author Response · Authors · 2024-11-25
> **Author Response to Reviewer UphH**
>
> Dear Reviewer UphH,
>
> We sincerely appreciate your constructive feedback. In response, we have carefully revised the manuscript, highlighting all changes in orange for discreparencies.
> We have carefully considered your suggestions, and we hope that the following response can address your concerns:
>
> > Comment 1: Although some experiments were performed on a continuous setting, it is unknown (not even addressed) how the algorithms scales in both continuous state and action spaces. The current test case is a simple environment with discrete actions.
>
> **Response 1:** Thank you for raising this concern.
> Please note that our main contributions are on the theoretical side. Extending such analyses to continuous spaces remains a significant challenge in the field [1].
>
> ---
>
> > Comment 2: Please discuss challenges that you anticipate in scaling your approach to environments where both state and action spaces are continuous, and potential solutions to the challenges.
>
> **Response 2:**
> The literature highlights significant challenges in learning feasible sets for large or continuous state spaces [2,3,4]. My approach also encounters several obstacles in addressing these issues.
>
> Scaling feasible set learning to practical problems with large state spaces remains a pressing challenge in the field [1]. One key difficulty is the estimation of the ground-truth expert policy, which is hard to obtain in an online setting. A potential solution involves extracting the expert policy from offline datasets of expert demonstrations. However, these datasets often contain a mix of optimal and sub-optimal demonstrations, leading to sub-optimal expert policies. Addressing this issue could involve: 1) treating the dataset as noisy and applying robust learning algorithms designed to handle noisy demonstrations, or 2) combining offline demonstrations with online fine-tuning, where feasible, to refine the learned policy.
> Finally, the scalability of learning in continuous spaces is frequently hindered by the curse of dimensionality. Dimensionality reduction techniques can mitigate this challenge by simplifying state and action representations while retaining the features essential for effective policy learning.
>
> We have included this discussion in the revised manuscript in Appendix F.
>
> ---
>
>
> > Comment 3: Please include a discussion of how you might adapt your theoretical analysis and sample complexity bounds for high-dimensional continuous spaces in your future work.
>
> **Response 3:**
> Sample complexity analysis has primarily focused on discrete state-action spaces [5]. Extending such analyses to continuous spaces remains a significant challenge in the field. Existing algorithms for learning feasible sets [2, 3, 4] face difficulties when scaling to problems with large or continuous state spaces. This is largely due to their sample complexity being directly tied to the size of the state space, which presents a substantial limitation since real-world problems often involve large or continuous spaces.
>
> To address this, function approximation plays a pivotal role in mitigating the curse of dimensionality and promoting generalization. Linear Markov Decision Processes (MDPs) [6, 7] offer a straightforward yet robust framework by assuming that the reward function and transition dynamics can be represented as linear combinations of predefined features. This assumption allows for theoretical exploration of sample complexity.
>
> In future work, we plan to leverage the Linear MDP framework as a foundation to design scalable methods for inferring feasible cost sets within the ICRL framework.
>
> We have included this discussion in the revised manuscript in Appendix F.
>
> ---
>
> References
>
> [1] Lazzati, F., Mutti, M., \& Metelli, A. M. How does Inverse RL Scale to Large State Spaces? A Provably Efficient Approach. In The Thirty-eighth Annual Conference on Neural Information Processing Systems.
>
> [2] Alberto Maria Metelli, Filippo Lazzati, and Marcello Restelli. Towards theoretical understanding of inverse reinforcement learning. ICML, 2023.
>
> [3] Lei Zhao, Mengdi Wang, and Yu Bai. Is inverse reinforcement learning harder than standard reinforcement learning? ICML, 2024.
>
> [4] Filippo Lazzati, Mirco Mutti, and Alberto Maria Metelli. Offline inverse rl: New solution concepts and provably efficient algorithms. ICML, 2024.
>
> [5]] Agarwal, Alekh, et al. "Reinforcement learning: Theory and algorithms." CS Dept., UW Seattle, Seattle, WA, USA, Tech. Rep 32 (2019): 96.
>
> [6] Chi Jin, Zhuoran Yang, Zhaoran Wang, and Michael I Jordan. Provably efficient reinforcement learning with linear function approximation. COLT, 2020.
>
> [7] Lin Yang and Mengdi Wang. Sample-optimal parametric q-learning using linearly additive features. In ICML, 2019.

---

> ### Comment · Reviewer_UphH · 2024-11-26
>
> Thanks for including discussions about adopting your analysis to continuous spaces. I appreciate the work done however my rating  still remains unchanged.

---

> ### Author Response · Authors · 2024-11-28
> **Author Response to Reviewer UphH**
>
> Dear Reviewer UphH,
>
> We are deeply grateful for the reviewer’s feedback and the significant time and effort invested in reviewing our manuscript. Your insightful comments have been invaluable in enhancing the clarity of our work. Thank you very much!

---

### Official Review · Reviewer_B2Ar · 2024-11-04

**Soundness:** 3
**Presentation:** 2
**Contribution:** 3
**Rating:** 6
**Confidence:** 3

**Summary:**

In applications such as robot learning, it is often the case that the learner (e.g. robot) must abide by certain safety constraints when learning to perform a task. Because such constraints can be hard to specify, methods of learning the constraints from demonstration have been proposed, an approach known as Inverse Constrained Reinforcement Learning (ICRL). Prior work has made one of the following assumptions: access to a known transition model or access to a generative transition model that can be queried at any state-action pair. The existing work that does not impose such assumptions has not examined efficiency and estimation errors. This paper proposes two algorithms for learning a set of feasible constraints that align with the expert preferences. Sample complexity bounds are presented for both algorithms. The algorithms are evaluated on Gridworld and Point Maze tasks.

**Strengths:**

The paper presents novel sample complexity guarantees on ICRL problems in the setting where the transition is unknown. While the paper presents substantial notation, Section 4 does a good job of describing the lemmas in understandable terms. Section 5, especially 5.1 and 5.2, would benefit from similar elaboration. The steps in the proofs are mostly explained well.

**Weaknesses:**

Several weaknesses are listed below, of varying importance.

I believe the paper would benefit from a broader discussion of the Related Works. More specifically, how does the paper and the setting it considers compare to the following works?
- Chou et al., “Learning constraints from demonstrations,” 2020
- Kim and Oh, “Efficient off-policy safe reinforcement learning using trust region conditional value at risk”, 2022.
- Moskovitz et al., “Reload: Reinforcement learning with optimistic ascent-descent for last-iterate
convergence in constrained mdps,” 2023.
- Lindner et al., “Learning safety constraints from demonstrations with unknown rewards,” 2024
- Kim et al., “Learning Shared Safety Constraints from Multi-task Demonstrations,” 2024

Nearly all of the results in the Empirical Evaluation section (Sec. 6) are visually difficult to parse. For example, the UCB results are almost entirely hidden in Figure 3’s top and middle rows (rewards and costs, respectively). While including numerous baseline comparisons is beneficial, considering including different plots in the appendix to make the comparison more interpretable. In addition to an unclear comparison to the baselines in terms of discounted cumulative rewards and discounted cumulative costs, neither BEAR nor PCSE appear to beat the Random algorithm in terms of WGloU score. Overall, it is unclear to me what the takeaways from the empirical results are.

The paper assumes finite state and action spaces, as well as an infinite horizon. In the experiments, these assumptions do not always hold (e.g. Point Maze is a continuous environment). There is a brief mention in Appendix D.3 about the density model in the continuous space, but overall, the discussion of how the theoretical assumptions translate into the practical settings considered is lacking.

In Theorem 5.6, the sample complexity of PCSE is given by the minimum of the sample complexity of BEAR and a term dependent on the minimum cost advantage function. In the proof of Theorem 5.6, the paper states that the sample complexity of BEAR (Theorem 5.5) applies to PCSE because it is optimizing a tighter bound. The justification is Corollary C.6. Examining the proof of Corollary C.6, it is not clear how one is a tighter bound than the other.

The paper would benefit from further discussion of the optimality criterion. The first constraint, with the Q difference for completeness, “tracks every potential true cost function.” The second constraint, focused on accuracy, expresses that the learned cost function must be close to a true cost function. How does it “[prevent] an unnecessarily large recovered feasible set?” At a higher level, the paper would benefit from more motivation/discussion of why the estimation should be included in the optimization problem. In other words, why can we not naively solve the problem as though we had perfect estimates, and then handle the estimation errors exclusively in the analysis? As discussed above, Section 5 (especially 5.1 and 5.2) would benefit from non-technical elaboration in the style of Section 4.

In Equation 9, which defines the optimization problem of PCSE, the supremum is over distributions over the state space, rather than the state and action space. More specifically, it is

$\Pi^r = {\pi \in \Delta: \inf_{\mu_0 \in \Delta^S} \mu_0^T (V^{r, \pi} - V^{r, \hat{\pi}^*}) \geq \mathcal{R}_k} $

rather than

$\Pi^r = {\pi \in \Delta: \inf_{\mu_0 \in \Delta^{S \times A}} \mu_0^T (V^{r, \pi} - V^{r, \hat{\pi}^*}) \geq \mathcal{R}_k} $

**Questions:**

Some questions are included in the weakness section.

The PCSE approach, described in Algorithm 1, obtains an exploration policy pi_k by solving the optimization problem in Equation 9. In Equation 9, $\Pi^r$ (rewards, not costs) is defined as

$$\Pi^r = \{ \pi \in \Delta: \inf_{\mu_0} \mu_0^T (V^{r, \pi} - V^{r, \hat{\pi}^*}) \geq \mathcal{R}_k \}$$

Because these are the value function of rewards, I am confused why the difference should not be flipped, such that:

$$\Pi^r = \{\pi \in \Delta: \inf_{\mu_0} \mu_0^T (V^{r, \hat{\pi}^*} - V^{r, \pi}) \geq \mathcal{R}_k\}.$$

In other words, why should the order of the two value functions not be flipped, given it is an infimum.

---

> ### Author Response · Authors · 2024-11-25
> **Author Response to Reviewer B2Ar - (1/3)**
>
> Dear Reviewer B2Ar,
>
> We sincerely value your time and effort in evaluating our work. In response, we have carefully revised the manuscript, highlighting all changes in orange for discrepancies. We have prepared comprehensive responses and clarifications to address each point you raised. We hope these responses can resolve your concerns.
>
> > Comment 1: I believe the paper would benefit from a broader discussion of the Related Works. More specifically, how does the paper and the setting it considers compare to the following works?
>
> **Response 1:** Thanks for mentioning these related papers. Our approach infers a feasible cost set encompassing all cost functions consistent with the provided demonstrations, eliminating reliance on additional information to address the inherent ill-posedness of inverse problems (multiple solutions to expert demonstrations). In contrast, prior works either require multiple demonstrations across diverse environments or rely on additional settings to ensure the uniqueness of the recovered constraints.
> This feasible set approach can focus on analyzing the intrinsic complexity of the ICRL problem only, without being obfuscated by other factors, resulting in solid theoretical guarantees [1].
>
> In the revised manuscript, we have included these additional related works and the above discussion on comparison with mentioned works in Appendix B.
>
> [1] Lazzati, F., Mutti, M., \& Metelli, A. M. How does Inverse RL Scale to Large State Spaces? A Provably Efficient Approach. In The Thirty-eighth Annual Conference on Neural Information Processing Systems.
>
> ---
>
> > Comment 2: Nearly all of the results in the Empirical Evaluation section (Sec. 6) are visually difficult to parse. For example, the UCB results are almost entirely hidden in Figure 3’s top and middle rows (rewards and costs, respectively). While including numerous baseline comparisons is beneficial, considering including different plots in the appendix makes the comparison more interpretable. In addition to an unclear comparison to the baselines in terms of discounted cumulative rewards and discounted cumulative costs, neither BEAR nor PCSE appear to beat the Random algorithm in terms of WGloU score. Overall, it is unclear to me what the takeaways from the empirical results are.
>
> **Response 2:** Thanks for your suggestion. We have updated Figure 3 to include two baselines (random and
> $\epsilon$-greedy) and moved the plots for the other two baselines (max-entropy and UCB) to Appendix Figure 5.
>
> We want to clarify that PCSE beats the Random algorithm in terms of WGIoU and cumulative rewards and costs. We can see that the red curve (representing PCSE) converges more quickly than the dark blue curve (representing Random algorithm). The key takeaway here is that PCSE is sample-efficient which validates the theoretical side.
>
> ---
>
> > Comment 3: The paper assumes finite state and action spaces, as well as an infinite horizon. In the experiments, these assumptions do not always hold (e.g. Point Maze is a continuous environment). There is a brief mention in Appendix D.3 about the density model in the continuous space, but overall, the discussion of how the theoretical assumptions translate into the practical settings considered is lacking.
>
> **Response 3:** For the assumption of finite state and action spaces to the continuous environment, we have included an additional paragraph for methods regarding how we utilize the density model for scaling to the continuous environment in Appendix D.3 in the revised manuscript. Having the max-length episode setting in Gridworld does not defy the infinite horizon assumption in our theory, because the Gridworld environment (with a terminal location) has finite states, and the optimal solution of a CRL phase in ICRL should have limited length. We want to eliminate scenarios like the cyclic circumstances for better time efficiency where the agent traverses in a cycle without an endpoint.

---

> > ### Comment · Reviewer_B2Ar · 2024-11-25
> > **Reviewer response**
> >
> > Thank you for your thorough response. Regarding "Response 2," I am still confused about how we can conclude that PCSE is more sample-efficient than the other methods, when it seems that in nearly all plots, the shaded regions overlap. Should I be only focused on the means? (Also, the y-axis titles for rows 1 and 2 in Figure 3 are the same.)

---

> > > ### Author Response · Authors · 2024-11-28
> > > **Author Response to Reviewer B2Ar**
> > >
> > > Dear Reviewer B2Ar:
> > >
> > > Thank you for engaging in discussion with us!
> > > We apologize that we previously wrongly uploaded the figures in the left-upright corner. We have now fixed it in the latest version. We think that you should focus on the mean curve. PCSE is more sample-efficient because the violation rate decreases fastest and WGIoU converges fastest.

---

> ### Author Response · Authors · 2024-11-25
> **Author Response to Reviewer B2Ar - (2/3)**
>
> > Comment 4: In Theorem 5.6, the sample complexity of PCSE is given by the minimum of the sample complexity of BEAR and a term dependent on the minimum cost advantage function. In the proof of Theorem 5.6, the paper states that the sample complexity of BEAR (Theorem 5.5) applies to PCSE because it is optimizing a tighter bound. The justification is Corollary C.6. Examining the proof of Corollary C.6, it is not clear how one is a tighter bound than the other.
>
> **Response 4:** This is because the LHS of Corollary C.6. case 1 is the upper bound of the LHS of case 2, i.e.,
> $\max\limits_{\pi\in\Pi^\dagger}\max\limits_ {\mu_0\in \Delta^{\mathcal{S}}}|\mu_0^T(I_ {\mathcal{S}\times\mathcal{A}}-\gamma P_\mathcal{T}\pi)^{-1}\mathcal{C}_ k|\leq\frac{1}{1-\gamma}\max\limits_ {(s,a)\in\mathcal{S}\times\mathcal{A}}\mathcal{C}_ k(s,a)$, which results from
>
> - matrix infinity norm inequalities that $\|AB\|_\infty\leq\|A\|_\infty\|B\|_\infty$;
>
> - $\|\mu_0\|_\infty\leq 1$;
>
> - $\|(I_{\mathcal{S}\times\mathcal{A}}-\gamma\pi P_\mathcal{T})^{-1}\|_{\infty}\leq \frac{1}{1-\gamma}$.
>
> Thus, when BEAR converges (satisfies case 1 in Corollary C.6), PCSE definitely converges (satisfies case 2 in Corollary C.6).
>
> ---
>
> > Comment 5: The paper would benefit from further discussion of the optimality criterion. The first constraint, with the Q difference for completeness, “tracks every potential true cost function.” The second constraint, focused on accuracy, expresses that the learned cost function must be close to a true cost function. How does it “[prevent] an unnecessarily large recovered feasible set?” At a higher level, the paper would benefit from more motivation/discussion of why the estimation should be included in the optimization problem. In other words, why can we not naively solve the problem as though we had perfect estimates, and then handle the estimation errors exclusively in the analysis? As discussed above, Section 5 (especially 5.1 and 5.2) would benefit from non-technical elaboration in the style of Section 4.
>
> **Response 5:**
> The first condition of Def. 4.9 states that for every cost within the exact feasible set, the best estimated cost within the estimated feasible set should exhibit a low error under all optimal policies across all instances. Consider the case where estimated costs exist everywhere (the estimated feasible cost set is a universal set), the first constraint is still satisfied. Hence, we need the second condition to get rid of such undesirable cases ([prevent] an unnecessarily large
> estimated feasible set). The second condition does this by requiring that there exists a ground-truth cost function with a low error for every estimated cost function. A universal estimated set (or any other unnecessarily large
> estimated feasible set) under the second condition can not have a low error for every estimated cost function.
>
> We don't quite understand what the reviewer means by 'as though we had perfect estimates'. In fact, if the transition model and the expert policy are perfectly estimated, the exact feasible cost set can be recovered and the ICRL problem is solved.
>
> ---
>
> > Comment 6: In Equation 9, which defines the optimization problem of PCSE, the supremum is over distributions over the state space, rather than the state and action space. More specifically, it is
> $\Pi_k^r=\{\pi\in\Delta_ {\mathcal{S}}^{\mathcal{A}}:\inf_ {\mu_0\in\Delta^{\mathcal{S}}}\mu_0^{T}\Big(V^{r,\pi}_ {\widehat{\mathcal{M}}_  k}-V^{r,\widehat{\pi}^*_ k}_ {\widehat{\mathcal{M}}_ k}\Big)\geq \mathfrak{R}_ k\}$
> rather than
> $\Pi_ k^r=\{\pi\in\Delta_ {\mathcal{S}}^{\mathcal{A}}:\inf_ {\mu_0\in\Delta^{\mathcal{S}\times\mathcal{A}}}\mu_0^{T}\Big(V^{r,\pi}_ {\widehat{\mathcal{M}}_ k}-V^{r,\widehat{\pi}^*_ k}_ {\widehat{\mathcal{M}}_ k}\Big)\geq \mathfrak{R}_ k\}$
>
> **Response 6:** Could the reviewer offer further explanations, because we do not see the problem with this. $\mu_0$ is the initial distribution, as defined in the notation. Directly applying the current policy at states where $\mu_0(s)>0$ can generate the initial distribution on state-action pairs.

---

> ### Author Response · Authors · 2024-11-25
> **Author Response to Reviewer B2Ar - (3/3)**
>
> > Comment 7: The PCSE approach, described in Algorithm 1, obtains an exploration policy $\pi_k$ by solving the optimization problem in Equation 9. In Equation 9, $\Pi^r$ (rewards, not costs) is defined as
>
> $\Pi_ k^r=\{\pi\in\Delta_ {\mathcal{S}}^{\mathcal{A}}:\inf_ {\mu_0\in\Delta^{\mathcal{S}}}\mu_0^{T}\Big(V^{r,\pi}_ {\widehat{\mathcal{M}}_ k}-V^{r,\widehat{\pi}^*_ k}_ {\widehat{\mathcal{M}}_ k}\Big)\geq \mathfrak{R}_ k\}$
>
> Because these are the value functions of rewards, I am confused as to why the difference should not be flipped, such that:
>
> $\Pi_ k^r=\{\pi\in\Delta_{\mathcal{S}}^{\mathcal{A}}:\inf_ {\mu_0\in\Delta^{\mathcal{S}}}\mu_0^{T}\Big(V^{r,\widehat{\pi}^*_ k}_ {\widehat{\mathcal{M}}_ k}-V^{r,\pi}_ {\widehat{\mathcal{M}}_ k}\Big)\geq \mathfrak{R}_ k\}$
>
> In other words, why should the order of the two value functions not be flipped, given it is an infimum?
>
> **Response 7:** $\Pi_k^r$ states that exploration policies should focus on states
> with potentially higher cumulative rewards, where possible constraints lie. This is reasonable because constraints exist in places where a policy achieves higher rewards than the expert. If the difference is flipped, $\pi$ may achieve lower rewards than the optimal policy on the estimated transition model ($\widehat{\pi}_k^*$), these places are not of critical importance for exploration. We have proven in lemma C.15 that the optimal policy $\pi^*$ exists in $\Pi_k^r$ (not flipped).

---

### Meta-Review · Area_Chair_nRbB · 2024-12-16

**Metareview:**

This paper presents algorithm more efficient exploration algorithms for inverse reinforcement learning with constraints. The reviewers found the setting novel and worthwhile, and were positive about the theoretical guarantees. Ultimately, reviewers had three main criticisms. First, they felt that the submission provided an incomplete perspective on prior work. Second, reviewers were disappointed that the constraint satisfaction was only approximate, and feared that this may deter the algorithm from succeeding in practical settings. Finally, one of the authors found multiple technical errors. Though some were resolved during the discussion, others emerged still, and the reviewer could not be certain of the correctness. Given the middling reviews, and the positive of legitimate mathematical errors, I recommend this submission be rejected.

**Additional Comments On Reviewer Discussion:**

Discussions focused primarily on the lack of applicability to practical scenarios due to strong assumptions and approximate constraint satisfaction, among others, as well as prevalence of mathematical errors. The positive reviewers were unwilling to champion the paper in response.

---

### Decision · Program_Chairs · 2025-01-22

Reject